# Preparing high-concentration individualized carbon nanotubes for industrial separation of multiple single-chirality species

Dehua Yang [1,2,3], Linhai Li [1,2,4], Xiao Li[1,2,4], Wei Xi[1,2], Yuejuan Zhang[1,2,4], Yumin Liu[1], Xiaojun Wei[1,2,4,5], Weiya Zhou [1,2,4,5], Fei Wei[6], Sishen Xie[1,2,4,5] & Huaping Liu [1,2,4,5] ✉

Industrial production of single-chirality carbon nanotubes is critical for their applications in high-speed and low-power nanoelectronic devices, but both their growth and separation have been major challenges. Here, we report a method for industrial separation of single-chirality carbon nanotubes from a variety of raw materials with gel chromatography by increasing the concentration of carbon nanotube solution. The high-concentration individualized carbon nanotube solution is prepared by ultrasonic dispersion followed by centrifugation and ultrasonic redispersion. With this technique, the concentration of the as-prepared individualized carbon nanotubes is increased from about 0.19 mg/mL to approximately 1 mg/mL, and the separation yield of multiple single-chirality species is increased by approximately six times to the milligram scale in one separation run with gel chromatography. When the dispersion technique is applied to an inexpensive hybrid of graphene and carbon nanotubes with a wide diameter range of 0.8–2.0 nm, and the separation yield of single-chirality species is increased by more than an order of magnitude to the sub-milligram scale. Moreover, with present separation technique, the environmental impact and cost of producing single-chirality species are greatly reduced. We anticipate that this method promotes industrial production and practical applications of single-chirality carbon nanotubes in carbon-based integration circuits.

Single-wall carbon nanotubes (SWCNTs) are considered ideal electronic and photoelectronic materials in the post-Moore era due to their extremely high carrier mobility, structure-tunable bandgap, and nanoscale body[1–5]. However, a slight difference in the atomic arrangement between different SWCNTs induces large changes in their optical and electrical properties. The diverse properties of an as-synthesized SWCNT mixture will definitely hinder their application. To overcome these challenges, the industrial production of single-chirality SWCNTs with identical properties has long been a goal in the field of carbon nanomaterials[6,7]. In recent decades, impressive progress in the structural separation of SWCNTs has been made. Various liquid-phase separation techniques, including ultracentrifugation[8,9], conjugated polymer[10,11], DNA-wrapped ion exchange chromatography[12,13], aqueous two-phase extraction (ATPE)[14–16], and gel chromatography[17–25], have

[1]Beijing National Laboratory for Condensed Matter Physics, Institute of Physics, Chinese Academy of Sciences, Beijing 100190, China. [2]Center of Materials Science and Optoelectronics Engineering, and School of Physical Sciences, University of Chinese Academy of Sciences, Beijing 100049, China. [3]Advanced Passivation Technology Lab, College of Physics Science and Technology, Hebei University, Baoding 071002, China. [4]Beijing Key Laboratory for Advanced Functional Materials and Structure Research, Beijing 100190, China. [5]Songshan Lake Materials Laboratory, Dongguan, Guangdong 523808, China. [6]Department of Chemical Engineering, Tsinghua University, Beijing 10084, China. ✉e-mail: liuhuaping@iphy.ac.cn

been established for high-quality raw SWCNTs with narrow diameter distributions, such as CoMoCAT, HiPco-SWCNTs, and arc-discharge SWCNTs[26,27]. With these techniques, dozens of single-chirality SWCNTs and even double-wall semiconducting carbon nanotubes have been separated[28,29]. In particular, some of them have recently been produced on the sub-milligram scale by gel chromatography due to the simplicity, high efficiency, and low cost of this technique[21]. However, such a separation yield is still far from the requirements of practical applications in carbon-based electronic and optoelectronic devices. Moreover, the extreme growth conditions of high-quality SWCNTs limit their production yield and increase their production cost, hindering the subsequent industrial separation of single-chirality SWCNTs. Therefore, developing a generalizable method for the efficient separation of single-chirality SWCNTs from different raw materials, especially those industrially produced at low cost is of great significance for the industrial production of various single-chirality species.

The chirality separation of SWCNTs by gel chromatography is based on their selective adsorption onto and desorption from the gel medium[20,21]. Although the separation efficiency of single-chirality species was improved greatly by overloading[17], temperature control[18], mixed surfactants[19] and even their combination[20–22], further scaling up separation yield of (n, m) is still difficult due to insufficient resolution and relatively low efficiency[17–25]. To improve the separation yield of single-chirality SWCNTs, the preparation of high-concentration individualized SWCNTs is expected to be an effective method. Based on molecular adsorption kinetics[30–32], increasing the SWCNT concentration could promote their mass transfer and binding rate to the gel surface and result in an increase in the adsorption of the absolute amount of each (n, m) semiconducting species in solution, thus improving the separation efficiency and yield of single-chirality SWCNTs. Notably, the preparation of individualized SWCNT solution is a critical step in the separation of single-chirality SWCNTs that determines the structural purity of the separated SWCNTs. At present, ultrasonic dispersion is widely used to prepare individualized SWCNT solutions[8–25]. However, with this technique, high concentrations and high dispersity are difficult to reconcile. Ultrasonic dispersion detaches SWCNT bundles into individuals or thin bundles through shear forces generated by acoustic cavitation[33]. With an increase in the initial SWCNT concentration, the viscosity and density of the dispersed SWCNT solution inevitably increased, frustrating the ultrasonic propagation in the dispersion and degrading acoustic energy into heat via viscous friction. According to classical acoustic theories, $P_x = P_0\,e^{-\alpha x}$, where $P_0$ and $P_x$ represent the sound pressure from the source and that at distance x from the source, respectively[33,34]. The attenuation coefficient $\alpha$ is roughly proportional to the viscosity of the medium. The reduction in the sound pressure increased the threshold of the cavitation effect to prevent the formation and implosion of cavitation bubbles[34]. As a result, it is difficult to detach single SWCNT from their bundles and thus prepare individualized SWCNT solutions. Although several groups have reported high-concentration SWCNT solutions dispersed by molecules of cresols[35], polymers[36], and even surfactants[37] for the preparation of films, the dispersion states of SWCNTs have not been verified, and the dispersant molecules are incompatible with the separation process of SWCNTs. The highest concentration of individualized SWCNT solution for the separation of single-chirality SWCNTs has been reported to be approximately 0.25 mg/mL[22]. Such a low concentration makes it difficult to achieve industrial separation of SWCNTs.

Here, we report a strategy for dispersing a highly concentrated individualized SWCNT solution by redispersion, in which the SWCNT solution was first ultrasonically dispersed, followed by ultracentrifugation and reultrasonic dispersion. With this technique, the dispersible initial concentration of HiPco-SWCNTs increased from 1 to 8 mg/mL, and the corresponding concentration of the resulting individualized SWCNT solution increased from 0.19 to ~1.02 mg/mL. The achievement of high-concentration SWCNT dispersion is attributed to the removal of large SWCNT bundles and impurities through the first centrifugation which is difficult to disperse, which greatly reduces the viscosity of SWCNT solution and thus enhances the ultrasonic dispersion efficiency of SWCNTs. The separation yield of single-chirality SWCNTs increased by approximately six times with gel chromatography by using the high-concentration SWCNT mixture as the parent solution. More than 8 types of single-chirality species, such as (6, 4), (6, 5), (11, 1), (7, 5), (7, 6), (8, 3), (8, 4) and (9, 1), were separated on a milligram scale from HiPco-SWCNTs in one run of separation by increasing the volume of gel column and the amount of SWCNT dispersion. Moreover, this dispersion technique was shown to be applicable for the low-cost and commercial hybrid of graphene and SWCNTs (G-SWCNTs) with a wide diameter range of 0.8–2.0 nm[38–40]. By increasing the initial dispersible concentration of the G-SWCNT raw materials from the typical 1 to 4 mg/mL, the separation yield of single-chirality SWCNTs was increased by more than one order of magnitude. In particular, nine types of single-chirality SWCNTs, namely, (6, 4), (6,5), (7, 3), (7, 5), (7, 6), (8, 4), (9, 1), (9, 4) and (10, 3), were prepared on the submilligram scale. In addition, the diameter separation of large-diameter semiconducting SWCNTs with diameters ranging from 1.2 to 1.7 nm was also achieved on a milligram scale. The distinct improvement in the separation yield of SWCNTs by increasing the concentration of SWCNT solution is mainly ascribed to the enhanced transfer of SWCNTs from bulk solution to the gel surface and thus their adsorption onto a gel, which reduces the proportions of unadsorbed and irreversibly adsorbed SWCNTs. By life techno-economic and life cycle assessments, the mass separation of single-chirality species showed distinct advantages in efficiency, energy consumption, and cost compared with our previous methods[18,20] by increasing the concentration of SWCNTs. Our present dispersion and separation strategy provides a method for the industrial separation of single-chirality SWCNTs over a wide diameter range. Given that sonication and centrifugation processes have been industrially applied[41–43], we believe that the current method can further scale up the separation of single-chirality SWCNTs, and reduce environmental impact, energy consumption, and separation costs by employing industrial homogenizer, and centrifugation, and gel column system.

## Results

### Dispersion of high-concentration SWCNT solution

To prepare a high-concentration individualized SWCNT solution, we proposed a strategy consisting of sonication, centrifugation, and resonication. An illustration of this process is shown in Fig. 1a. First, the raw SWCNT materials were added to an aqueous solution of 2 wt% sodium dodecyl sulfate (SDS) and ultrasonically dispersed into smaller bundles. The dispersion time was set as a proportional function of the initial SWCNT concentration (see the experimental details in Supplementary Note 1), so as to sufficiently disperse SWCNTs to increase the concentration of the resulting SWCNT dispersion (Supplementary Note 2 and Supplementary Figs. 1 and 2). To optimize the dispersion time of the SWCNT solutions with different initial concentrations, we studied the variation of their viscosity and concentration with time during the dispersion process (Supplementary Fig. 2). At the beginning, the viscosity of SWCNT solution increased rapidly and then reached the maximum value with increasing sonication duration. With continuously increasing sonication time, the viscosity of SWCNT solution decreased dramatically and finally reached an approximately constant value. We can imagine that at the beginning, SWCNT powders were crushed into particles composed of large bundles and impurities under ultrasonic dispersion. Due to the low dispersity and increasing number of suspended particles, the friction between these particles increased the viscosity of solutions. As the sonication time increased, SWCNTs were continuously stripped from the bundles. The sidewalls of SWCNTs exposed to the solution were readily coated with

surfactant molecules, decreasing the friction due to the high lubrication effect of surfactants[44]. Meanwhile, denser surfactant coatings around SWCNTs provided a repulsive region and thus excellent fluidity[45]. Therefore, when the viscosity of a SWCNT solution dropped to a stable value after ultrasonic dispersion, it should be sufficiently dispersed, which was evidenced by its concentration evolution over sonication time (Supplementary Note 2 and Supplementary Fig. 2).

During the dispersion process, the viscosity of the SWCNT solution with an initial concentration of 4 mg/mL was much higher than that of the SWCNT solution with a concentration of 1 mg/mL (Supplementary Note 2 and Supplementary Fig. 2). As shown in Fig. 1b, the resulting viscosity of the dispersed SWCNT solution increased with increasing the initial concentrations. In particular, at concentrations higher than 4 mg/mL, the viscosity increased faster. After centrifugation at 210000 × g for 60 min, the large bundles and impurities precipitated, and the viscosity of the supernatant solutions decreased. The higher the initial concentration was, the more viscosity was reduced. As a result, the resulting viscosity differed slightly over the wide concentration range of 1–8 mg/mL. Clearly, the high viscosity at higher concentrations was mainly derived from large bundles and impurities, which not only consumed the ultrasonic dispersion power but also hindered ultrasonic propagation, degrading the dispersion efficiency. Thus, a high-concentration individualized SWCNT solution has rarely been achieved by simply increasing the time of sonication or ultracentrifugation (Supplementary Notes 1 and 3, Supplementary Fig. 3), which was verified by the chiral purity of the (6, 4) SWCNTs separated at low temperature[18]. To achieve the high dispersity of the SWCNT solution, the supernatant was collected for further ultrasonic redispersion. In this step, the dispersion time depends on the initial SWCNT concentration, which increases by 20 minutes for every 1-mg/mL increase in SWCNT concentration for 100 mL solution. Thus, the remaining small bundles were dispersed into individual SWCNTs, which was demonstrated by the separation of single-chirality (6, 4) species and their atomic force microscope images (Supplementary Note 3 and Supplementary Fig. 3). Before separation, the remaining impurities, such as carbon impurities, and the metallic particles

generated from the ultrasonic tips were removed by a second centrifugation at 210000 × g for 15 min. It should be noted that although the high dispersity of SWCNT solution was verified by the separation of high-purity single-chirality SWCNTs (Supplementary Note 3 and Supplementary Fig. 3), it was not possible for the two-step dispersion and centrifugation method to achieve a bundle-free solution, but to minimize bundles in solution.

Figure 1c shows the optical absorption spectra of SWCNT solutions prepared from different initial concentrations of HiPco-SWCNTs. With an increase in the initial concentration, the optical absorbance of the obtained SWCNT solution increased. The SWCNT concentration was evaluated based on the optical absorbance at 273 nm[46], and the relationship between the initial raw SWCNT concentration and the resulting concentration of the individualized SWCNT solution is plotted in Fig. 1d. It is clear that the concentration of the as-prepared individualized SWCNT solution increased almost linearly from 0.19 to 0.83 mg/ml with an increase in the concentration of the initial SWCNT solution from 1 to 4 mg/mL. When the initial concentration further increased, the concentration of the resulting SWCNT dispersion started to increase slowly. When the initial SWCNT concentration increased to 8 mg/mL, the concentration of the obtained SWCNTs reached ~1.02 mg/mL. Higher concentrations of raw SWCNTs resulted in the formation of gelation after a short period of ultrasonic dispersion, and the ultrasonic output power decreased rapidly to zero, rendering further dispersion impossible. The dispersion efficiency of SWCNTs was evaluated in terms of the ratio between the resulting concentrations of individualized SWCNTs and their initial concentrations. As shown in Fig. 1d, the dispersion efficiency remained at approximately 20% when the initial concentration of SWCNTs was lower than 4 mg/mL. As the initial concentration of SWCNTs further increased, the dispersion efficiency decreased rapidly. This result was consistent with the dramatic decrease in the viscosity of the higher initial concentration SWCNT solution after the first centrifugation (Fig. 1b). To improve the dispersion efficiency of high-concentration SWCNT solution, we tried to disperse the SWCNT solution with the initial concentration of 8 mg/mL by three iterative dispersions.

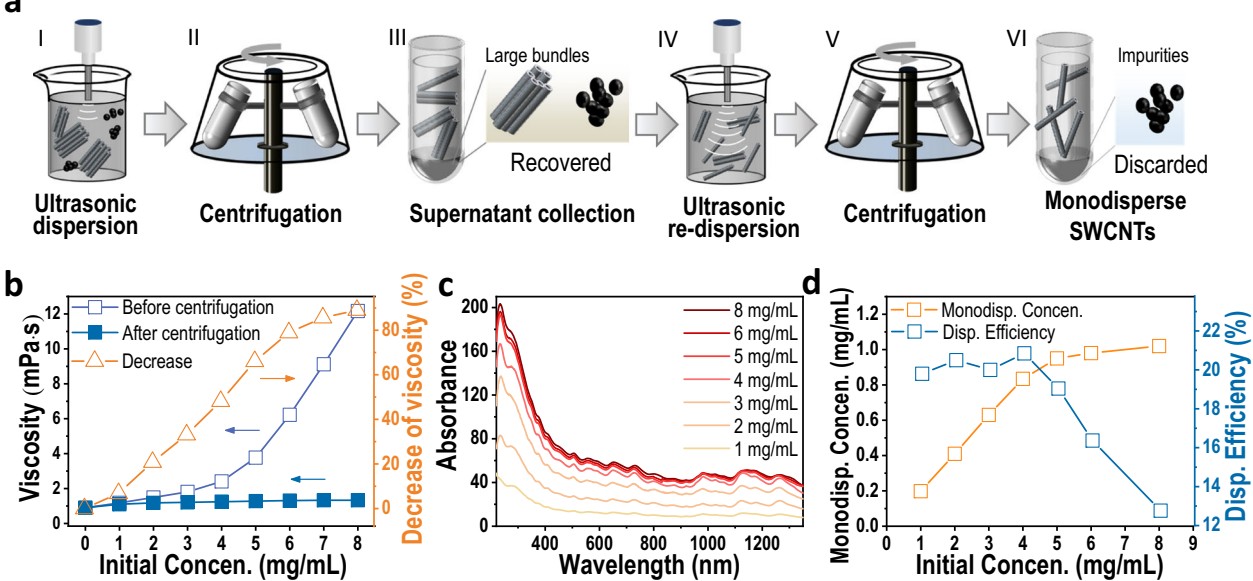

**Fig. 1 | The preparation of a high-concentration individualized SWCNT solution. a** Schematic diagram of the preparation of high-concentration individualized SWCNT solution; **b** The relationship between the shear viscosity and initial concentration of SWCNT solution before and after the first ultracentrifugation; **c** Optical absorption spectra of the as-prepared individualized SWCNTs with different initial concentrations; **d** The relationship between the initial SWCNT concentration and the concentration of the as-prepared SWCNT solution. "Disp" and "Concen" are the abbreviation of the words "Dispersion" and "Concentration". The dispersion efficiency of SWCNTs was evaluated by the ratio between their resulting and initial concentrations. The SWCNT concentration in the as-prepared dispersions was calculated based on their absorbance at 273 nm[46].

However, after the first ultrasonic dispersion for 11 h and subsequent centrifugation for 1 hour, the concentration of supernatant obtained was only 1.17 mg/mL (Supplementary Fig. 1), which was close to the concentration of SWCNTs obtained by re-dispersion (Fig. 1c, d). Therefore, even after the second and third dispersions, the highest SWCNT concentration obtained would not exceed 1.17 mg/mL. Due to the high initial concentration, it was difficult for SWCNTs to disperse sufficiently in a short time, resulting in a significant decrease in their concentration after the first centrifugation. The original concentration of 4 mg/mL could be the optimal concentration from the perspective of the full utilization of raw materials and subsequent separation efficiency.

## Milligram-scale separation of single-chirality species

To explore the effect of SWCNT concentrations on the separation yield of single-chirality species, 10 mL of as-prepared SWCNT dispersions with different concentrations, which were prepared by different initial concentrations of HiPco-SWCNT solution, were loaded into a 40 mL gel column for the separation of single-chirality SWCNTs by selective adsorption into and subsequent desorption of SWCNTs from the gel column, as shown in Fig. 2a. Specifically, in the selective adsorption stage, SWCNTs such as (6, 4), (6, 5), (7, 5) and (7, 6) with larger chiral angles were selectively adsorbed into gel columns in a single surfactant of the SDS system at different temperatures, while SWCNTs with small chiral angles were selectively adsorbed from the unadsorbed solutions in the binary surfactant SDS/sodium cholate (SC) system at low temperatures[21]. After that, the SWCNTs adsorbed in the gel column were stepwise eluted by gradually increasing the concentration of sodium deoxycholate (DOC) in the cosurfactant systems of SDS/SC/DOC to separate single-chirality SWCNTs[47]. The separated (n, m) species were tuned to the same volume for comparison. The detailed separation process is described in the experimental section (Supplementary Note 1 and Supplementary Fig. 4). Compared with previous works[17–25], the current separation strategy showed higher resolution on SWCNT structures and higher separation efficiency (Supplementary Note 4 and Supplementary Fig. 5). The optical absorption spectra indicated that the yield of the separated (6, 4) SWCNTs increased substantially with an increase in the concentration of individualized SWCNTs (Fig. 2b and Supplementary Fig. 6a). Furthermore, their relationship is plotted in Fig. 2c (blue line). The yield of the (6, 4) SWCNTs clearly increased linearly with an increase in the concentration of individualized SWCNT solution. The yield of (6, 4) species separated from the SWCNT dispersion with a concentration of 1.02 mg/mL (corresponding to the initial concentration of 8 mg/mL) was approximately six times that of the (6, 4) species separated from the SWCNT dispersion of 0.19 mg/mL. The yields of other (n, m) SWCNTs exhibited a similar increasing trend. Mass separation of 14 types of (n, m) species was achieved (Supplementary Fig. 7). The lowest chiral purity of them was evaluated to be higher than 70% (Supplementary Fig. 8), which was referred to as "single-chirality SWCNTs" due to high enrichment in chiral structure distribution. Notably, to improve the separation yield of single-chirality species by increasing the concentration of SWCNT dispersion, the size of the gel column is also important. The capacity of small gel columns to carry SWCNTs is limited. With increasing SWNCT concertation, the gel column would be saturated by SWCNTs, limiting an increase in the separation yield of SWCNTs. To confirm this, we employed a 10-mL gel column for the separation of SWCNTs. Similarly, the volume of the loaded SWCNT solution was fixed at 10 mL. The optical absorption spectra of the separated (6, 4) SWCNTs under different SWCNT concentrations are presented in Supplementary Fig. 6b. The relationship between the yield of (6, 4) SWCNTs and the concentration of the loaded SWCNT solution is described in Fig. 2c (red symbol). It can be clearly seen that with an increase in concentration, the separation yield of (6, 4) SWCNTs gradually reaches a plateau.

To demonstrate the potential of scalability, the separation yield of single-chirality SWCNTs was linearly expanded by increasing the volume of the gel column and the loaded amount of SWCNTs. As shown in Fig. 2d, e, eight single-chirality species, (6, 4), (8, 3), (11, 1) (7, 5), (7, 6), (6, 5), (9, 1) and (8, 4), were produced on a milligram scale by loading 360 mL of HiPco-SWCNTs with the SWCNT concentration of 0.83 mg/mL into a gel column of 900 mL using an automatic chromatography system (Avant 150 GE Healthcare) in one separation run. Their purities were confirmed to be above 90% by spectral evaluation (Supplementary Fig. 8). As reported previously[4], the optimal linear density of SWCNT films for high-performance carbon-based field-effect transistors is 100 to 200 tubes/μm. Approximately 0.003 mg of SWCNTs is required to prepare a 4-inch monolayer SWCNT film with a linear density of 200 tubes/μm (Supplementary Note 5). Therefore, the total yield (-12 mg) of single-chirality species in one separation run is sufficient for the preparation of about four thousand wafer-scale monolayer SWCNT films. Notably, although the dispersion time increased with an increase in the initial concentration of dispersible SWCNTs, the increase in ultrasonic time did not significantly decrease the length of SWCNTs (Supplementary Fig. 9). Another important feature of the current method is that the gel columns could be recycled for 20 times, which reduced the separation cost of SWCNTs. Specifically, the separation yield of single-chirality SWCNTs increased by 8–15% gradually with increasing cycle number within the first 5 cycles. Beyond 5 cycles, the separation yield started to decline, possibly because the growing SWCNT bundles and carbon impurities trapped in gel columns degraded the adsorption capacity of the gel column. After 20 cycles, the separation yield decreased by approximately 30% compared with the new gel. Thus, on average, the separation yield of single-chirality species did not decrease significantly. The detailed experimental process and discussion were presented in Supplementary Note 6 and Supplementary Fig. 10. We can expect that a further liner amplification of the yield of single-chirality species could be achieved when a larger chromatography system is used that can couple with larger gel columns. These results sufficiently confirm that increasing the nanotube concentration is an effective method to industrially produce single-chirality species.

## Dispersion and separation of high-concentration G-SWCNTs

Another key to the industrial production of single-chirality SWCNTs is to establish a high-throughput method to separate single-chirality SWCNTs from various raw SWCNTs including those with nonselective structures due to their high growth yield and low cost[38–40]. Because of the wide diameter distribution, more kinds of SWCNTs could be separated. However, both the absolute and relative contents of each type of SWCNT were relatively low in such SWCNT solutions when prepared by the traditional dispersion method, preventing mass separation of single-chirality SWCNTs. Neither the separation of single-chirality SWCNTs from such raw materials nor their mass production has been reported. Our present preparation technique of a high-concentration individualized SWCNT solution enables an increase in the absolute concentration of each kind of SWCNT in the dispersion solution and thus provides a possible method to separate single-chirality SWCNTs at a large scale from such materials.

To verify our hypothesis, we separated single-chirality SWCNTs by employing a low-cost hybrid of graphene and SWCNTs (G-SWCNTs), which were industrially produced by facile catalytic growth on a layered double hydroxide method, as the raw material[38–40]. The synthesis of SWCNTs by this method is supposed to be more sustainable and greener, compared with HiPco-SWCNTs[48]. The optical absorption spectra (Fig. 3a) exhibit no distinct absorption peaks in the range of 200 to 1400 nm, except for that at 257 nm originating from the plasma absorption of the π electrons of SWCNTs and graphene, indicating that the nanotubes have a uniform distribution over a wide diameter range because of the overlapping optical absorption peaks of

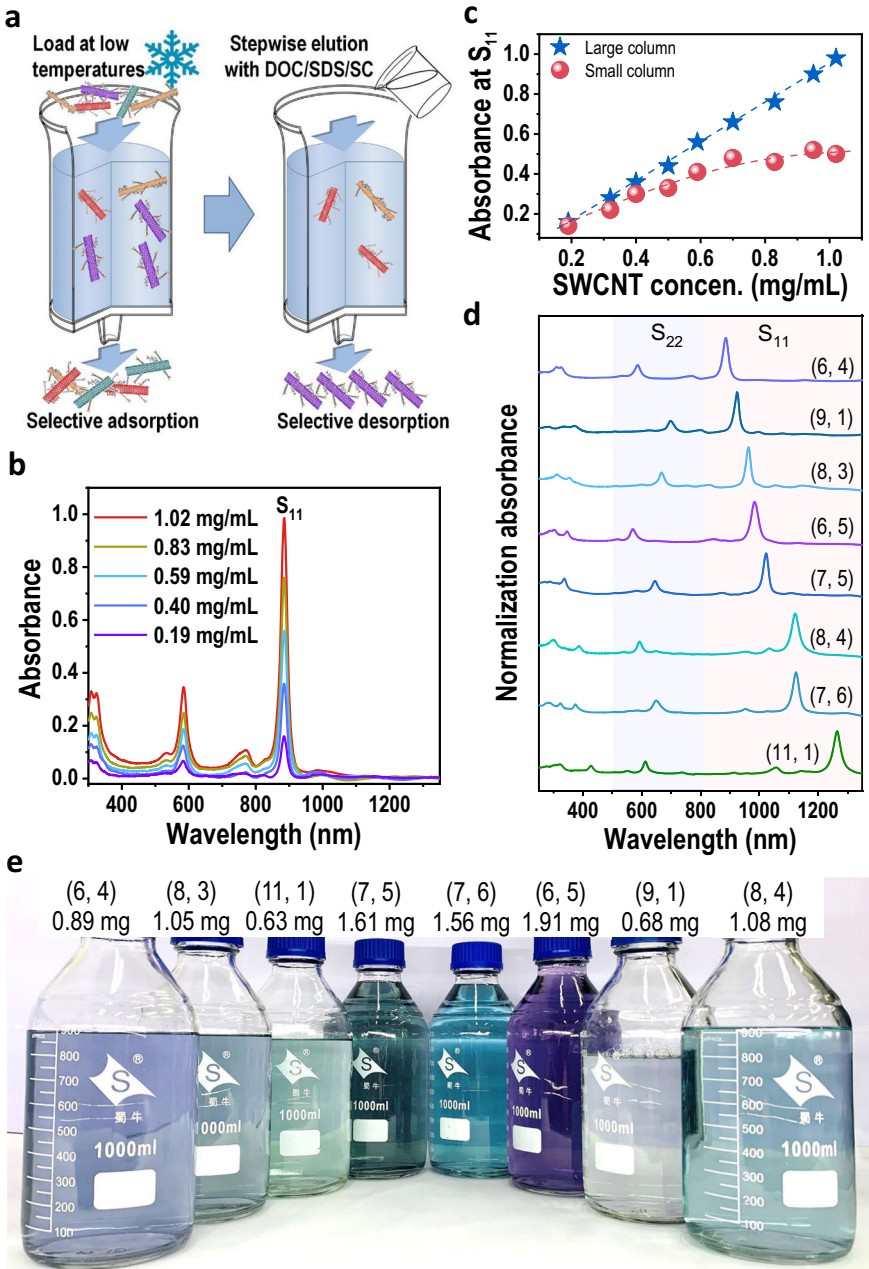

**Fig. 2 | Milligram-scale separation of single-chirality SWCNTs from a high-concentration individualized SWCNT solution. a** Schematic diagram of the separation of single-chirality SWCNTs. DOC, SC, and SDS represent sodium cholate, sodium deoxycholate, and sodium dodecyl sulfate. **b** Optical absorption spectra of single-chirality (6, 4) SWCNTs from SWCNT solutions with different concentrations using a 40-mL gel. The separated (6, 4) solutions were diluted to 20 mL for comparison. **c** The yield of single-chirality (6, 4) that separated by large (40 mL) and small (10 mL) gel columns as a function of the concentration of individualized SWCNT solution. **d** Optical absorption spectra of separated (n, m) species on the milligram scale. **e** Photograph of the solution of single-chirality species separated from high-concentration SWCNT solution. $S_{11}$ and $S_{22}$ are the optical transitions of the first and second subbands, respectively.

nanotubes of various diameters[49,50]. Raman and TEM characterization (Fig. 3b, c and Supplementary Fig. 11) further show that the raw materials contain graphene flakes and SWCNTs with a wide diameter range of 0.8 to 2 nm[49].

Individualized G-SWCNT solutions were prepared by the current redispersion technique. The initial concentration was varied from 1 to 5 mg/mL (Supplementary Note 1). The exact concentrations of the as-dispersed SWCNT solution could not be determined by optical absorbance due to the residual graphene (Fig. 3c and Supplementary Fig. 11), which was represented by the optical absorbance at 257 nm. With an increase in the initial SWCNT concentration, the optical absorbance at 257 nm (Fig. 3a) increased dramatically. The relationship

between the concentration of the initial SWCNTs and that of the as-prepared individualized SWCNT solution is shown in Fig. 3d. The concentration of the as-prepared individualized SWCNT solution increased approximately linearly with an increase in the initial concentration. Similarly, the dispersity of the SWCNT solution was verified by the separation of single-chirality (6, 4) SWCNTs. The results show that the dispersity of the G-SWCNT solution with an initial concentration of 5 mg/mL was insufficient for single-chirality separation (Supplementary Fig. 12), although the viscosity of the dispersion decreased dramatically after the first ultracentrifugation.

Single-chirality species were separated by the process shown in Fig. 2a (Supplementary Note 1). Selective adsorption was

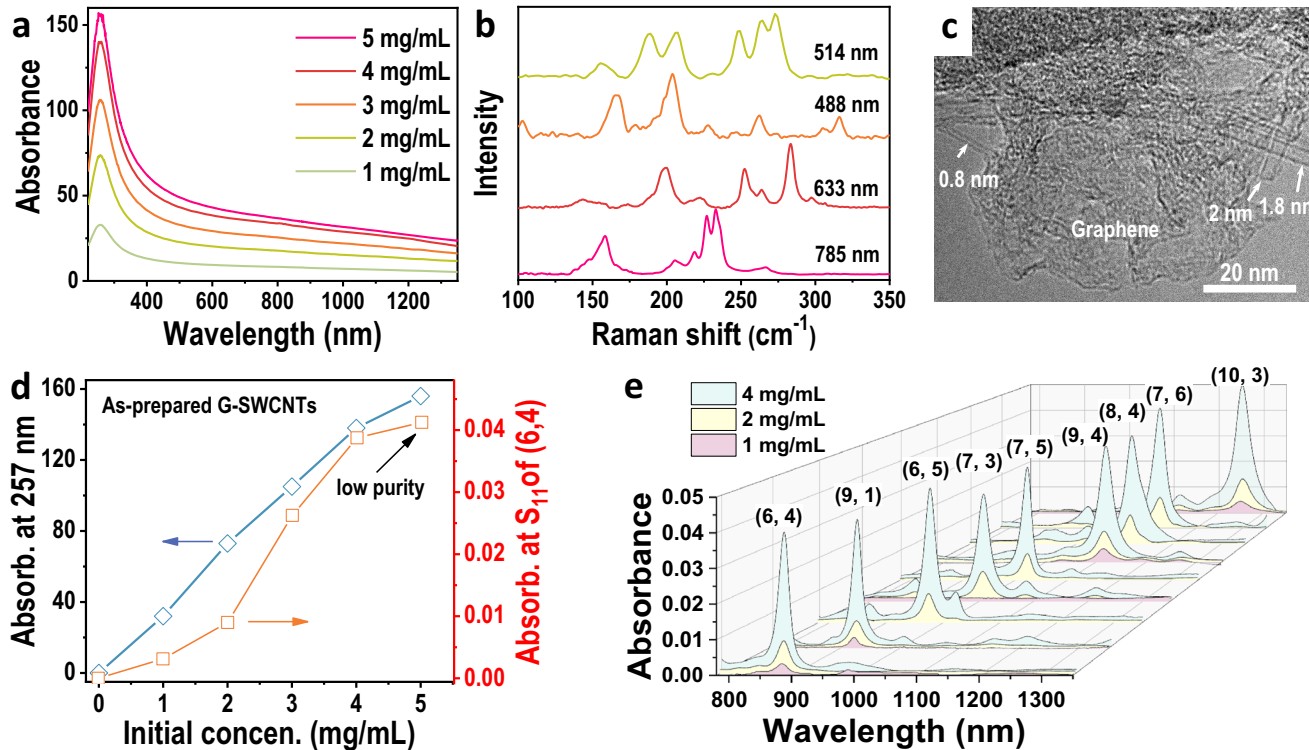

**Fig. 3 | Dispersion and structure separation of high-concentration G-SWCNTs.** **a** Optical absorption spectra of as-prepared G-SWCNTs with different concentrations. **b** Raman spectra of raw G-SWCNTs at excitation wavelengths of 514 nm, 488 nm, 633 nm, and 785 nm. **c** TEM image of raw G-SWCNTs containing graphene sheets and SWCNTs with different diameters. **d** Plot of the SWCNT concentration in the as-prepared dispersion (blue) and the corresponding yields of (6, 4) SWCNTs (orange) as a function of the initial concentration of G-SWCNTs. The resulting concentration of G-SWCNTs and the yield of (6,4) SWCNTs were represented by their optical absorbance. Absorb is the abbreviation of absorbance. **e** Optical absorption spectra of various single-chirality species separated from G-SWCNTs with different initial concentrations. For comparison, the volumes of the separated (*n, m*) species were tuned to 20 mL for the (9, 1), (6, 5), and (7, 3), and 40 mL for the (7, 5) and (10, 3), and 30 mL for the (6, 4), (8, 4), (9, 4) and (7, 6).

achieved in the SDS system at incrementally increased temperatures. Then, the adsorbed SWCNTs were selectively eluted with cosurfactants of SDS, SC, and DOC by gradually increasing the concentration of DOC. Specifically, 10-mL dispersions with different initial concentrations of 1, 2, and 4 mg/mL were loaded into 40-mL gel columns. The volumes of the separated (*n, m*) species were tuned to allow a comparison of their yields by their optical absorbance. As shown in Fig. 3e, a 12-fold increase in the separation yield of single-chirality (6, 4) SWCNTs between the 1- and 4-mg/mL samples was observed despite a 3.4-fold increase in the SWCNT concentration of the corresponding solutions. A similar result was also observed in the separation of other single-chirality species (Fig. 3e and Supplementary Table 1). Notably, in the case of the initial concentration of 1 mg/mL, although the purities of the separated (*n, m*) species were higher than 80% or even 90% (Supplementary Fig. 13), the yield was distinctly low, and even single-chirality (6, 5), (7, 5), (7, 6) and (8, 4) SWCNTs were not achieved, most likely due to their low concentration and low absolute content, and a large portion of them was lost as irreversible or unadsorbed species. A more detailed discussion was presented later. These results indicate that the preparation of a high-concentration and high-dispersity solution is a prerequisite for the separation of single-chirality SWCNTs from G-SWCNTs.

### Mass separation of single-chirality SWCNTs from G-SWCNTs
To achieve the scalable separation of single-chirality SWCNTs, 1600 mL of G-SWCNTs with an initial concentration of 4 mg/mL was separated with a 900-mL gel column using the temperature-assisted gel chromatography technique described above. Figure 4a shows the optical absorption spectra of the single-chirality species. Their chiral purities were comparable to those of the species separated from HiPco-SWCNTs (Fig. 4b and Supplementary Fig. 14). Submilligram-scale separation of multiple single-chirality species, including (8, 4), (6, 4), (7, 6), (9, 1), (7, 3), (7, 5), (9, 4), (6, 5) and (10, 3) species, was achieved (Fig. 4c). Possibly because of the different structural distributions in the raw materials (Supplementary Note 7 and Supplementary Figs. 15, 16), the separation yield of (8, 3) and (10, 2) was not scalable. Moreover, the separated single-chirality species did not contain graphene flakes, which directly flow through the gel columns as the unadsorbed products (Supplementary Fig. 17). Additionally, milligram-scale diameter separation of semiconducting SWCNTs with diameters larger than 1 nm has also been demonstrated using high-concentration G-SWCNTs (Supplementary Note 8 and Supplementary Fig. 18)[23], which covers the diameter ranges of commercial HiPco- (0.8–1.2 nm), plasma- (0.9–1.7 nm) and arc-discharge SWCNTs (1.2–2.0 nm). These fractions are productive materials for further separation and applications. To further demonstrate the universality of the present method, we further applied it to another commercial large-diameter SWCNT materials (OCSiAl's Tuball, 1.2–2.0 nm in diameter, Sigma-Aldrich). Since their diameter is larger than 1.2 nm, the separation of single-chirality SWCNTs cannot be achieved for the moment. We explored the separation of metallic/semiconducting SWCNTs from this material. The results also showed that, with the present redispersion technique, high-concentration individualized SWCNT solution was achieved, which greatly improved the separation yield of semiconducting SWCNTs (Supplementary Fig. 19).

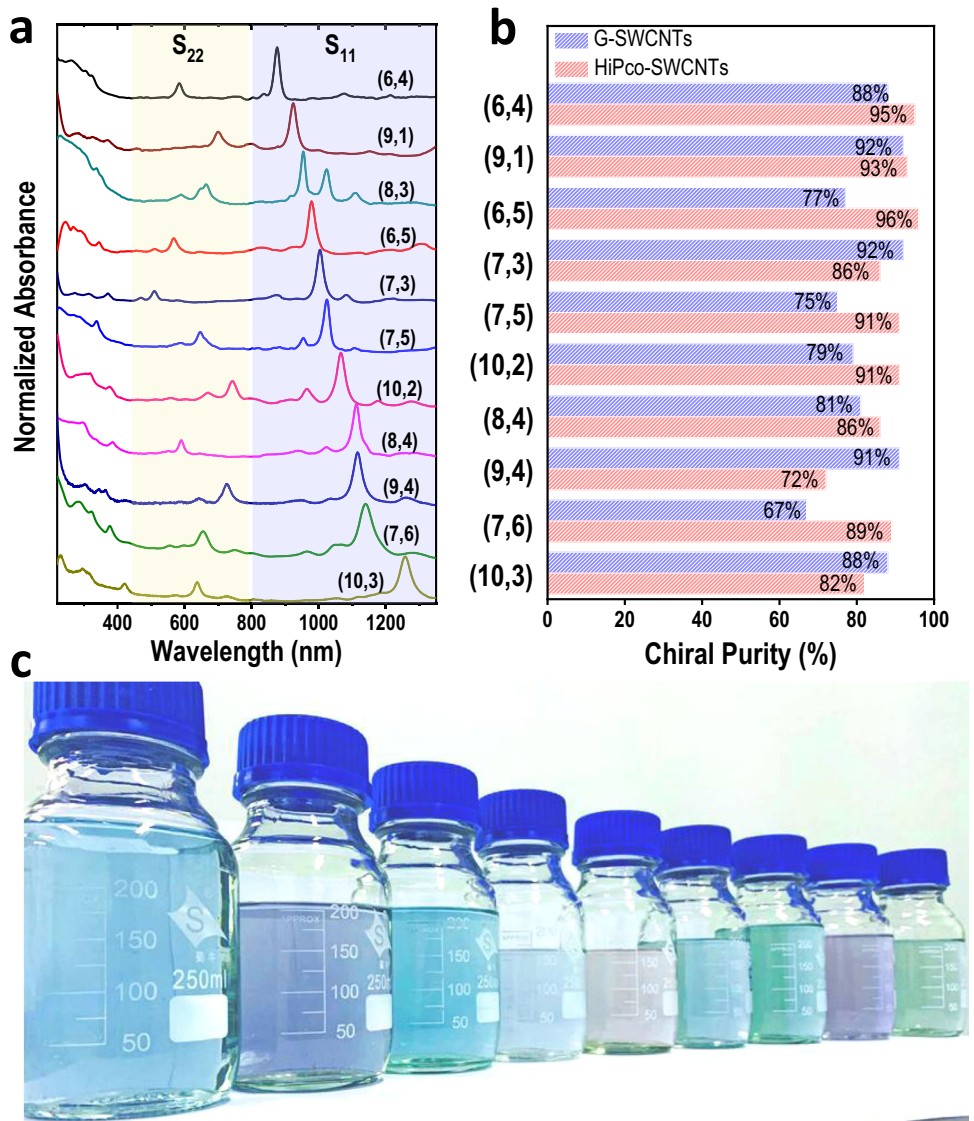

**Fig. 4 | Submilligram-scale separation of single-chirality species from high-concentration G-SWCNTs. a** Optical absorption spectra and **b** statistical purity distribution of the separated single-chirality SWCNTs. **c** Photograph of the solutions of high-purity single-chirality SWCNTs. From left to right, the solutions correspond to the (8, 4), (6, 4), (7, 6), (9, 1), (7, 3), (7, 5), (9, 4), (6, 5) and (10, 3) species.

## Discussion

To clarify the effect of the SWCNT concentration on the separation yield of single-chirality SWCNTs, we varied the concentrations of the loaded HiPco-SWCNT solution from 0.011 to 0.176 mg/mL and kept the volume fixed to systematically study the yield variation of (6, 4) SWCNTs. Specifically, 10 mL of as-prepared HiPco-SWCNT solutions with different concentrations were loaded onto 40 mL gel columns for the selective adsorption of (6, 4) SWCNTs at low temperatures.

The collected (6, 4) SWCNTs were diluted to 20 mL for the characterization of optical absorption spectra. As shown in Fig. 5a, the optical absorbance of the separated (6, 4) SWCNT solution with the same volume increased substantially with increasing concentration of loaded SWCNTs. Based on the optical absorbance at the $S_{11}$ peak of the separated (6, 4) SWCNTs, we plotted the relationship between the concentrations of the loaded SWCNTs and the yield of single-chirality species, as shown in Fig. 5b. To better describe the rate of increase in the yield of (6, 4) SWCNTs, we further calculated the first derivative of the relation curve between the concentration of the loaded raw materials and the yield of (6, 4) SWCNTs. The yield of (6, 4) SWCNTs

increased fastest as the concentration of the loaded SWCNTs increased from 0.011 to 0.044 mg/mL, and the corresponding optical absorbance at the $S_{11}$ peak of the separated (6, 4) SWCNTs increased by approximately an order of magnitude from 0.003 to 0.04. This trend coincides with the rapid increase in the separation yield of (6, 4) from G-SWCNTs, where the yield was increased by nearly a dozen fold as the initial concentration was increased from 1 to 4 mg/mL. The concentration of small diameter SWCNTs in the individualized G-SWCNT solution with an initial concentration of 1 mg/mL was further verified to be ~0.01 mg/mL by comparison with that of HiPco-SWCNTs (Supplementary Note 7 and Supplementary Fig. 15). After that, the rate of increase decreased, finally reaching a constant rate, i.e., a linear increase. Notably, increasing the concentration of the loaded SWCNT solution with a fixed volume increased both the concentration and the loading amount of SWCNTs. To understand the mechanism of high-concentration SWCNT solution improving the separation efficiency of single-chirality SWCNTs, it was necessary to study the influence mechanism of the concentration and the loading amount of SWCNT solution on the separation yield of single-chirality species, respectively.

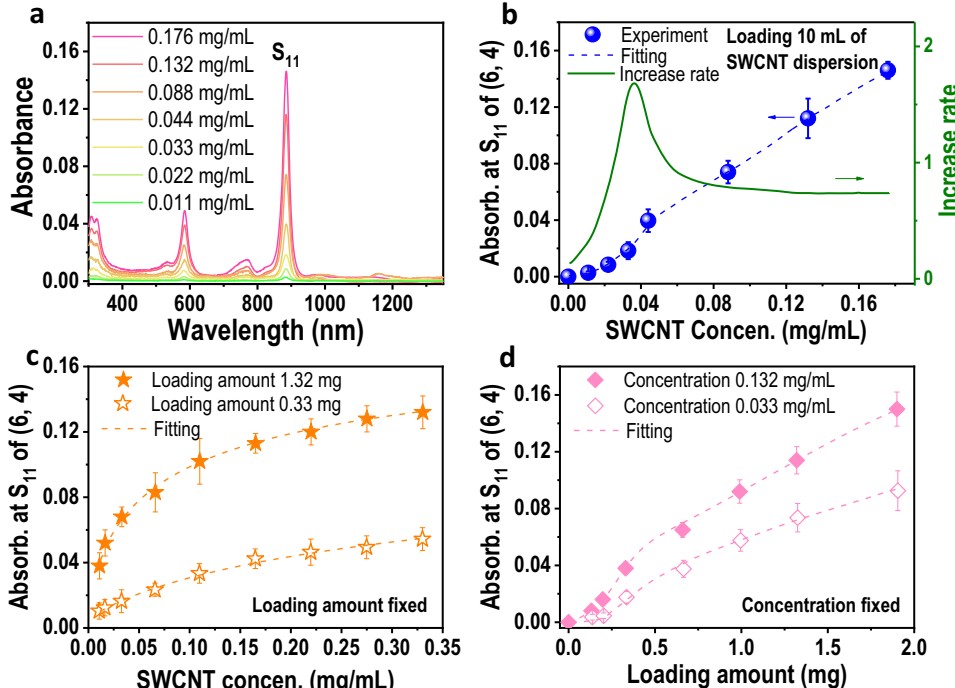

**Fig. 5 | Effect of SWCNT concentration on the separation yield of (6,4) single-chirality SWCNTs. a** Optical adsorption spectra of the separated (6, 4) species under different SWCNT concentrations with a fixed volume of 20 mL. **b** Plot of the absorbance at $S_{11}$ of (6, 4) as a function of SWCNT concentration. The green curve is the first derivative of the relation curve between the concentration of the loaded raw materials and the yield of (6, 4) SWCNTs. **c** Relationship between the concentration of individualized SWCNT concentration and the resulting yield of (6,4)

SWCNTs with a fixed loading amount. **d** Plot of loading amount versus absorbance intensity at $S_{11}$ of (6, 4) with a fixed concentration. "Absor." and "Concen." are the abbreviation of the words "Absorbance" and "Concentration". Notably, the data in **b**–**d** were piecewise fitted with polynomial functions. The green line in (**b**) represents the slope of the fitted curves. Each experiment was repeated three times. The error bars exhibit the standard deviation.

The effect of SWCNT concentration on the yield of single-chirality species was investigated by loading various HiPco-SWCNT concentrations with fixed loading amounts of 1.32 and 0.33 mg, which were diluted with surfactant solution to a variety of concentrations: 0.011, 0.017, 0.033, 0.067, 0.110, 0.165, 0.220, 0.275, and 0.33 mg/mL. Subsequently, all these dispersions were loaded into the same gel columns filled with 40 mL of gel for separating (6, 4) SWCNTs. The volumes of the separated (6, 4) SWCNTs were adjusted to 20 mL for comparison of their yields by their optical absorption spectra (Supplementary Fig. 20). The relationship between the (6, 4) yield and the concentration of the loaded HiPco-SWCNTs is presented in Fig. 5c. With an increase in concentration, the yield of (6, 4) single-chirality SWCNTs increased under both loading amounts. At low concentrations, the yield increased relatively quickly. In comparison, the effect of concentration was greater at higher loading amounts.

In SWCNT separation, the loaded SWCNTs are separated into unadsorbed SWCNTs that directly flow through the gel column, reversibly adsorbed SWCNTs that can be eluted, and irreversibly adsorbed SWCNTs. Among them, the reversibly adsorbed SWCNTs are regarded as the separable species, which determines the separation yield of single-chirality species. To clarify the effect of the concentration of loaded SWCNTs on the separation yield of single-chirality species, we systematically studied the proportion change of reversible adsorption, irreversible adsorption, and unadsorption by loading single-chirality (6, 5) SWCNTs with different concentrations into gel columns (Supplementary Note 9). The results show that a lower concentration of the loaded SWCNT solution with a fixed loading amount led to a higher proportion of unadsorbed SWCNTs (Supplementary Fig. 21). Similar to the separation of biomolecules in a chromatographic system[32,51], the concentration of SWCNT solution was a key parameter, which determined the mass transfer of SWCNTs from the

bulk solution to the boundary layer adjacent to gel surface and thus the SWCNT concentration in the reaction volume (Supplementary Note 9 and Supplementary Fig. 22). Based on the adsorption kinetics of molecules[30,31,52], a high SWCNT concentration in the reaction volume would promote the adsorption of SWCNT onto gel surface. At low concentrations, the mass transfer of SWCNTs to the gel surface was hindered due to the small SWCNT concentration gradient between the bulk solution and boundary layer and thus decreasing their adsorption onto gel surfaces. Most of SWCNTs remained in the bulk solution and flew through the gel column as unabsorbed species. In contrast, with increasing SWCNT concentration, the transfer of SWCNTs to the gel surface was enhanced, providing more SWCNTs for the binding reactions and thus reducing the proportion of unadsorbed SWCNTs. As the SWCNT concentration further increased, excessive SWCNT could be transferred to the gel surface and accumulated in the reaction volume within a short time, the adsorption kinetics turned to adsorption control. Therefore, further increasing the SWCNTs concentration, the adsorption amount of SWCNTs tended to saturate and the curve slopes in Fig. 5c decreased. A detailed discussion is presented in Supplementary Note 9.

The effect of the loading amount on the yield of the (6, 4) SWCNTs was explored by loading different amounts of HiPco-SWCNT solutions with a fixed concentration of 0.132 mg/mL into gel columns of 40 mL to separate (6, 4) SWCNTs. The volumes of the separated (6, 4) SWCNTs were adjusted to 20 mL for comparison of yield by optical absorbance at $S_{11}$ (Supplementary Fig. 23a). As shown in Fig. 5d, the yield of (6, 4) SWCNTs increased nonlinearly with an increase in the loading amount of SWCNTs. Specifically, under a loading amount of HiPco-SWCNTs less than 0.13 mg, the yield of the separated (6, 4) species increased slowly. When the loading amount increased from 0.13 to 0.33 mg, the yield of (6, 4) increased at the highest rate.

Although the loading amount increased by approximately two and a half times, the production of (6, 4) SWCNTs increased by approximately five times. With a further increase in the loading amount, the output of (6, 4) species increased linearly. At a fixed concentration of 0.033 mg/mL, the loading amount had a similar effect on the yield of (6, 4) SWCNTs (Supplementary Fig. 23b and Fig. 5d). The difference was that under the same loading amount, the yield of (6, 4) SWCNTs corresponding to a higher-concentration SWCNT solution was higher. As the loading amount increased, the yield difference between the two concentrations of samples increased.

To clarify the effect of the amount of loaded SWCNTs on the separation yield of single-chirality species, we systematically studied the proportion change of reversible adsorption, irreversible adsorption, and unadsorption by loading different volumes of (6, 5) SWCNTs with a fixed concentration into gel columns (Supplementary Note 10). The result showed that a low loading amount resulted in 70% irreversible SWCNTs and thus a low separation yield (Supplementary Note 10. and Supplementary Figs. 24 and 25). With an increase in the loading amount, the proportion of irreversibly adsorbed SWCNTs decreased rapidly from 70% to -15%, which originated from the quasirandom irreversible adsorption in gel chromatography[53]. Compared with reversible adsorption, irreversible adsorption sites exhibited stronger affinity with SWCNTs. When SWCNTs flew through gel column, they preferentially adsorbed at the irreversible adsorption sites. Conceptually, a gel column can be considered as being composed of many thin layers of gel[54]. When a small amount of SWCNT solution were loaded into a gel column, SWCNTs could be adsorbed by both reversible and irreversible adsorption sites in the upper layers of the gel column, leaving the irreversible adsorption sites in the lower layers unoccupied. During elution, the eluted SWCNTs from reversible adsorption sites may be captured again by the irreversible adsorption sites in the lower layers (Supplementary Fig. 25), leading to a significant decrease in the SWCNT concentration in the eluted solution and even no SWCNT were collected. With an increase in the loading amount of SWCNTs, more and more SWCNTs could be collected in the eluted solution because of the occupation of more irreversible adsorption sites, resulting in a dramatic decrease in the proportion of irreversible adsorption and thus a rapid increase in the separation yield of SWCNTs (Fig. 5d and Supplementary Fig. 25). Notably, as the loading amount continuously increased, more reversibly adsorbed SWCNTs were eluted while the probability of irreversible adsorption of SWCNTs also increased, leading to a constant proportion of irreversible adsorption. In this way, the separation yield began to increase linearly with an increase in the loading amount (Fig. 5d). A detailed discussion was presented in Supplementary Note 10.

These results sufficiently indicate that the rapid increase in the production of single-chirality SWCNTs caused by loading high-concentration SWCNT solution resulted from the combined effects of the increase in the concentration and loading amount of SWCNTs. In particular, for low concentrations and low loadings, the yield of single-chirality SWCNTs increased more obviously. At this stage, as the concentration and loading amount increased, the mass transfer and binding rate of SWCNTs were enhanced, and the proportion of reversible adsorption also increased, resulting in a rapid increase in the separation yield (Fig. 5b and Supplementary Notes 9 and 10). With further increasing the concentration and amount of the loaded SWCNTs, the increased rate of the adsorption of SWCNTs gradually slowed down to a constant value (Fig. 5b) because the adsorption of SWCNTs turned to adsorption control instead of mass transfer and the proportion of the irreversible adsorption reduced to a low constant proportion. These results fully illustrate the mechanism of improving the separation efficiency of single-chirality SWCNTs by using a high-concentration individualized SWCNT solution as raw materials in gel chromatography, especially for raw materials such as G-SWCNTs with a wide diameter distribution due to the low content of each type of

species. Thus, a high-concentration SWCNT solution is of great importance for industrial separation. These findings provide a reference for the improvement of the structural separation efficiency and yield with other liquid phase separation techniques because there is inevitable consumption in all separation processes. For example, the separated SWCNTs in the interphase boundary were inevitably lost due to difficult collection in aqueous two-phase extraction procedures[55].

Life cycle assessment (LCA) is very important for the industrial production of new material. Given that large-scale production data of single-chirality SWCNTs are lacking, LCA is essential for predicting the impacts of scaling up and assessing the technological readiness of the emerging methods[56,57]. Moreover, the LCA may provide a benchmark as well as a reference to evaluate the advantages and limitations of a separation method. Based on the full life cycle inventory (LCI) of the synthesis of raw SWCNTs (Supplementary Table 2)[48,56,57] and the inventory data of the separation process of single-chirality species (Supplementary Table 3), we performed an LCA on the industrial production of single-chirality SWCNTs with our present separation technique (Supplementary Fig. 26). Due to difference in the yield of various (n, m) species, the LCA analysis on different (n, m) species may vary greatly. For comparison, the average inventory data of producing 1 mg of single-chirality SWCNTs were calculated by dividing the sum of the various resources and energy consumed by the total yield of various single-chirality SWCNTs. The detailed process is presented in Supplementary Note 11. Figure 6a, b shows the life cycle greenhouse gas emissions and cumulative energy demand (CED) for producing 1 mg of single-chirality SWCNTs from HiPco- and G-SWCNTs with initial concentrations of 1 and 4 mg/ mL. It can be seen that, with increasing initial concentration from 1 to 4 mg/mL, both the greenhouse gas emissions and CED for HiPco-SWCNTs reduces by about 80% because they are dominantly contributed by electricity (Supplementary Table 3). The greenhouse gas emissions and CED for G-SWCNTs reduce even more by about 90%. These results indicate that the use of high-concentration SWCNTs for the separation of single-chirality species can greatly reduce the environmental impact. Due to the low separation efficiency from low-concentration SWCNT dispersion, it would take a longer time and more energy to separate the same amount of single-chirality SWCNTs. This is why, even when the initial concentration of G-SWCNTs was increased to 4 mg/mL, the greenhouse gas emissions and CED were still higher than those of HiPco-SWCNTs with an initial concentration of 1 mg/mL because the absolute concentration of (n, m) SWCNTs was still lower.

Considering the commercial application, we further analyzed the cost of separating 1 mg single-chirality species from HiPco- and G-SWCNTs with initial concentration of 4 mg/mL. The costs include those of raw SWCNT materials, surfactants, gel medium, deionized water, electricity, fixed assets, operating materials and labor (see details in Supplementary Note 11.3, Supplementary Tables 4–6, and Supplementary Fig. 27). The results are presented in Fig. 6c. It clearly shows that the production cost of single-chirality SWCNTs from high-concentration HiPco-SWCNTs and G-SWCNTs are $272/mg and $1337/mg, respectively. Compared with the initial concentration of 1 mg/mL, the costs reduce by about 77% and 92%, respectively. The separation cost of single-chirality species from high-concentration G-SWCNTs is comparable to those separated from 1-mg/mL HiPco-SWCNTs. As mentioned above, approximately 0.003 mg of SWCNTs is required to prepare a 4-inch monolayer SWCNT film (Supplementary Note 5)[4]. The separation costs of such an amount of single-chirality species from high-concentration HiPco- and G-SWCNTs are $0.8 and $3.8, respectively, which are comparable to that of a 4-inch silicon wafer. Overall, the separation efficiency using high-concentration SWCNT solution is distinctly higher compared with conventional 1-mg/mL samples, despite that the size of the gel column was increased, which reduced time, energy, and material consumption and thus the separation cost. Given that sonication and ultracentrifugation process

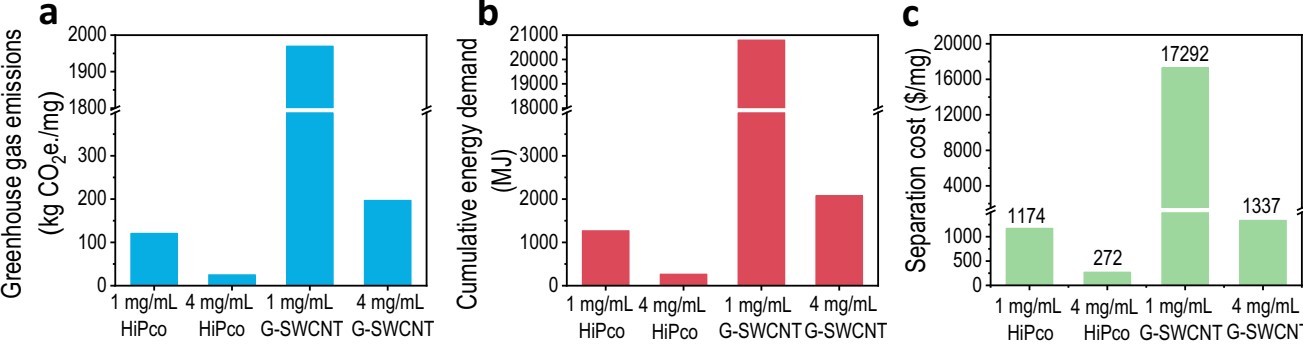

**Fig. 6 | Potential and challenge analysis for industrialization and commercialization of the current technique. a** Life cycle greenhouse gas emissions, **b** cumulative energy demand, and **c** cost of producing 1-mg single-chirality SWCNTs from HiPco- and G-SWCNTs under different concentrations. The detailed calculation is presented in Supplementary Note 11.3.

has been applied in the chemical synthesis industry[41,43] and also large volumes of SWCNT dispersion[42], we can expect that the separation efficiency of single-chirality species could be further increased while the separation cost, environmental impact, and energy consumption were further reduced by scaling up with industrial dispersion, centrifugation and gel column system (as discussed in Supplementary Note 11.3)[58]. These results demonstrate a high technology readiness level of our present technique, exhibiting a great promise of industrial preparation and commercial application of single-chirality SWCNTs.

It is notable that, although the cost of raw G-SWCNTs materials is only 1% of that of HiPco-SWCNTs (Supplementary Note 11 and Supplementary Fig. 27), the corresponding separation cost of single-chirality species is higher because of longer separation processes or more separation runs are required for separating the same amount of single-chirality species (Supplementary Table 3), which must require more energy, materials, and labor as well as waste output, and thus increases separation cost (Fig. 6c). Here, the unsorted large-diameter semiconducting SWCNTs in G-SWCNTs are not included in the cost. When more types of large-diameter single-chirality species could be separated, the separation cost could further reduce. Despite that, the low concentration of (n, m) species in G-SWCNTs increases the CED, greenhouse gas emissions, and cost of producing single-chirality species. Therefore, developing environmentally friendly and low-cost industrial synthesis of structurally enriched raw materials is important for the industrial separation and commercial application of single-chirality SWCNTs.

In summary, we have developed a method for the preparation of a high-concentration and individualized SWCNT solution with a concentration of up to 1 mg/mL by ultrasonic dispersion, centrifugation, and redispersion. By using an as-prepared high-concentration individualized solution of commercial HiPco-SWCNTs, the separation yield of single-chirality SWCNTs increased by nearly six times to reach the milligram scale. Moreover, the preparation of a high-concentration individualized hybrid of SWCNTs and graphene with a wide diameter distribution of 0.8–2.0 nm enabled us to separate single-chirality species on the submilligram scale and large-diameter semiconducting SWCNTs covering the diameter distribution of plasma-SWCNTs (0.9–1.7 nm) and arc-discharge SWCNTs (1.2–2.0 nm) on the milligram scale. The yield increase of the separated SWCNTs was mainly attributed to the increase in the absolute content of each species in the SWCNT solution, which reduced the proportion of irreversibly adsorbed and unadsorbed species. Based on the life cycle assessment and techno-economic assessment, our present technique makes distinct progress in efficiency, energy consumption, and cost, showing a high technology readiness level, which provides an approach for the industrial separation of single-chirality species.

## Data availability
The key data generated in this study are provided in the Supplementary Information. Source data are provided as a Source Data file and has also been deposited in Figshare under accession code https://doi.org/10.6084/m9.figshare.22579000. Source data are provided with this paper.

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

## Acknowledgements

This work was financially supported by the National Key Research and Development Program of China (grant nos. 2020YFA0714700 and 2018YFA0208402), the National Natural Science Foundation of China (grant nos. 51820105002, 51872320, 51472264, 11634014, and 52172060), the Strategic Priority Research Program of the Chinese Academy of Sciences (grant no. XDB33030100), the Key Research Program of Frontier Sciences, CAS (grant no. QYZDBSSW-SYS028), and the Youth Innovation Promotion Association of CAS (grant no. 2020005).

## Author contributions

H.L. proposed and supervised the project. D.Y. performed most of the experiments. L.L. and Y.L. contributed to the partial separation of SWCNTs. X.L., W.X., and Y.Z. assisted in the characterization of optical spectra, TEM, and AFM. F.W. provided G-SWCNTs. X.W., W.Z., and S.X. were involved in the discussions. H.L. and D.Y. co-wrote the manuscript. All authors analyzed and discussed the experimental data.

## Competing interests

The authors declare no competing interests.
