## [Peer Review File · Nature Communications]

Preparing high-concentration individualized carbon nanotubes for industrial separation of multiple single-chirality speciesREVIEWER COMMENTS

Reviewer #1 (Remarks to the Author):

This manuscript reports large-scale purification of single chirality SWCNT species using gel chromatograph method. By increasing the concentration of initial SWCNT dispersion loaded on gels, the authors obtained increased yield for single-chirality species. This result is certainly an improvement over what they reported before. However, there are a few concerns that need to be addressed before I can recommend its publication.

1. What is the new scientific finding this work offers? Increasing the concentration of initial SWCNT dispersion is expected to increase the purification yield. The authors should make it clear what is new.
2. What is the impact of higher dispersion concentration on gel column's life time and performance repeatability from one run to the next? These issues are important in "industrial" production.
3. The authors claim that they tested "a variety of raw materials", but the manuscript reported only two: Hipco and a synthetic mixture of graphene and SWCNTs. The latter is very interesting but not widely available. Is it similar to OCSiAl's Tuball material? Have you tested Tuball?

Reviewer #2 (Remarks to the Author):

The manuscript by Liu and co-workers describes the use of gel chromatography to purify highly concentrated solutions of single-walled carbon nanotubes into pure (single chirality) samples of numerous nanotube species. Using multiple rounds of ultrasonic dispersion, ultracentrifugation, and size-exclusion gel chromatography through Sephacryl S-200. While it is clear that the results obtained from this work are highly interesting to the community of researchers working on carbon nanotube purification, there are some concerns about the novelty and industrial applicability of the work. Some specific comments are provided below.

1. The authors claim that they are reporting a novel strategy for dispersing and purifying SWCNTs. However, there has been extensive literature on the purification of single-chirality SWCNTs by gel filtration through Sephacryl S-200 (references 17, 18, and 19). The authors use different elution temperatures to separate different SWCNTs, which has also been reported in reference 23. Some seminal reports of Sephacryl-based SWCNT and DWCNT purification are also missing from the references, including:

1. Flavel, B. S., Kappes, M. M., Krupke, R. & Hennrich, F. Separation of Single-Walled Carbon Nanotubes by 1-Dodecanol-Mediated Size-Exclusion Chromatography. *Organic Letters* 7, 3557–3564 (2013).
2. Moore, K. E. et al. Separation of Double-Walled Carbon Nanotubes by Size Exclusion Column Chromatography. *ACS Nano* 8, 6756–6764 (2014).
3. Flavel, B. S., Moore, K. E., Pfohl, M., Kappes, M. M. & Hennrich, F. Separation of Single-Walled Carbon Nanotubes with a Gel Permeation Chromatography System. *ACS Nano* 8, 1817–1826 (2014).

Although the authors did demonstrate that this separation method can be scaled up significantly relative to prior reports, the main difference here is an additional sonication and centrifugation step. This does not seem to be at the level of novelty required to warrant publication of the work in this journal.

2. The authors claim that their approach "provides a new method for the industrial separation of single-chirality SWCNTs". However, it is not clear that the extensive sonication and ultracentrifugation required to produce the dispersions, which are still on milligram scale, will be industrially relevant. In addition, the chromatographic gel used for this work is extremely expensive, limiting the feasibility of industrial scale-up.

3. The authors state that the yield of a particular SWCNT species (e.g., the (6,4) nanotubes) increases linearly with the concentration of the starting dispersion, which was loaded onto a 40 mL gel column. However, this must imply that the column does not get saturated by the SWCNTs, as

one would expect a plateau to be reached within the plot in Figure 2c. While one can appreciate that higher nanotube concentrations in the starting dispersion are difficult to achieve, it would have been helpful if the authors first showed that, with a smaller column volume, a plateau in nanotube separation efficiency is reached.

4. On page 14, the authors state that with a starting concentration of G-SWCNTs of 4 mg/mL, they achieved good separation of single-chirality species, but when the initial concentration was 1 mg/mL, single-chirality samples were not achieved. It is not clear why decreasing the concentration would decrease separation efficiency. One would expect the opposite. It would be helpful if the authors would propose an explanation for this result.

5. Overall, the authors show that as the concentration and volume of the starting nanotube dispersion increases, the size of the separation column must also increase. Additionally, due to some small amount of irreversible adsorption of nanotubes on the stationary phase, low initial nanotube concentrations will not lead to as high a recovery yield as high initial nanotube concentrations. Ultimately, many of the significant results are not surprising, and thus the novelty of the work is questionable. It does not seem to meet the standards required for this journal.

Reviewer #3 (Remarks to the Author):

Title: Preparing High-Concentration Monodisperse Carbon Nanotubes for Industrial Separation of Multiple Single-Chirality Species

Manuscript ID: NCOMMS-22-29646A-Z

Corresponding Author: Huaping Liu

This communication describes new achievements in gel-based purification of single walled carbon nanotubes (SWCNTs). The achievement of low-cost procurement of highly-purified "single-chirality" SWCNTs is a longstanding challenge of the carbon nanoscience community. Significant advancements in this area are highly likely to have a substantial immediate positive impact within a variety of areas that 1) study fundamental SWCNT properties; 2) presently employ SWCNT within devices/schemes but stand to benefit from the availability of higher purity materials; and 3) are limited in scale due to the high cost of purified SWCNT. Because of this potential impact coupled with the many scientific resources that have been invested in solving the problem of producing low-cost high purity SWCNT, the scope and scale of this work is appropriate for consideration in Nature Communications.

As an academic researcher who is choosing to sign my review letter, I would like to briefly highlight both my expertise and my naivete in the areas of this manuscript. Specifically, I have approximately 11 years of experience in SWCNT purification, nearly all of which has utilized gel-based methods. More recently, my research team at the University of Colorado at Colorado Springs has focused on understanding mechanistically the unique relationship between hydrogel surface, SWCNT structure/chirality, and surfactant nature/density that affords gel-based single chirality purification. Several of our recent publications have focused on the role of the gel medium in this dynamic system. Given this, I am intimately familiar with the challenges of creating high concentration SWNT suspensions and the general use of gel-based techniques to afford isolation of single chirality species (although my research team does not employ the temperature gradient method used in this specific work). In contrast, I am markedly less familiar with the Life Cycle Assessment (LCA) performed in this work, although I appreciate the value of such given the direct connection made by the authors between novel methodology and industrial scalability. All this is to say that I accepted the task of reviewing this work with both enthusiasm to learn about new findings in this area and with confidence that I am able to provide comments that are both informed and without conflicts of interest either in favor of or against acceptance for publication.

The manuscript reports data and analysis that appear to fit within five distinct themes (in order of appearance in the manuscript): 1) the novel method of iterative sonication/centrifugation to achieve high concentration suspensions of individualized SWCNT; 2) a demonstration of how such suspensions achieve scaled-up isolation of high-purity materials from a relatively high-cost source of unpurified SWCNT; 3) a demonstration of how such suspensions achieve scaled-up isolation of high-purity materials from a relatively low-cost source of unpurified SWCNT; 4) efforts to probe mechanistic effects of SWCNT concentration on gel-based purification; and 5) a Life Cycle Assessment estimating the costs of using the reported novel method to produce purified SWNT at scale. In general, the manuscript is well written and well connected with the data presented and the experimental methods are clear. Those methods employed are of appropriate scope and scale for this project and align well with standards in the field. The pure quantity of data collectively presented within the manuscript and the Supplementary Information (SI) demonstrates the investment that this team has made in this project, certainly a commendable achievement. It is my expectation that most SWNT purification researchers, upon reading this work, will be inspired to quickly test the effectiveness of relatively high-concentration SWCNT suspensions within their SWCNT purification schemes/workflows.

That said, there are some shortcomings of this manuscript that I would like to see addressed by the authors before this work is considered for publication. My concerns primarily fall into two categories: 1) a lack of control experiments to better demonstrate the novel effects of the iterative sonication/centrifugation method; and 2) a lack of a clear mechanism for concentration effects on both purification efficiency and efficacy (% purity of single-chirality species). Other relatively minor concerns are also listed below, as organized by the 5 themes of the manuscript listed above.

Theme 1 (pages 6-8): The novel method of iterative sonication/centrifugation to achieve high concentration suspensions of individualized SWCNT

- This section of the manuscript, specifically the method of iterative sonication/centrifugation, represents the foundation of the work and affords/justifies the remainder of the study. The authors report that "the dispersion time was set as a proportional function of the initial SWCNT concentration." I would expect that most of the energy dissipation from a tip-horn sonicator is transferred thermally (and not chemically), so to me it is unclear why scaling sonication duration with SWCNT concentration is a logical choice. Further, this choice convolutes initial SWNT concentration and duration of the first (of two) sonication periods. For example, if suspensions initially at 1, 4 and 8 mg/mL are subjected to identical sonication/centrifugation duration, how does the final SWNT concentration (after the second centrifugation) differ from a procedure that is scaled to initial SWCNT concentration? It would be useful for the authors to conduct and report this experiment, as it would provide insight into the necessity of prolonged sonication for high concentration samples as well as the linear scaling choice of sonication time vs. SWCNT concentration.

- The authors state that prolonged sonication does not achieve high concentrations of individualized SWCNT due to a limitation in viscosity. It would be interesting (and certainly related to the foundational achievement of this work) to know how SWCNT suspension viscosity increases over time for the two different SWCNT source materials used in this study. This question would be addressed in a singular plot showing sonication time on the horizontal axis and containing three vertical axes of 1) pre centrifugation viscosity; 2) total SWCNT dispersed following first centrifugation; and 3) total SWCNT dispersed following second sonication/centrifugation. This data at, say 1 and 4 mg/mL loading of raw SWNT, would clearly demonstrate the role of initial sonication time in eventual procurement of high concentration suspensions. Further, it would be a compelling addition to this work if the temporal increase in viscosity during sonication could be correlated with a model of system viscosity driven by a combination of rods (individualized SWNT), large diameter rods (bundled SWNT), and plates (graphitic impurities). The findings presented in this work clearly demonstrate that an improved understanding of suspension evolution and dynamics during tip horn sonication could greatly benefit the efficacy and efficiency of SWCNT purification, so any additional physical/mechanistic insights would be greatly appreciated by myself and would likely be appreciated by others in the SWCNT purification community.

- The authors claim that their two-step sonication/centrifugation method creates "monodisperse" SWCNT suspensions (I will use the word "individualized" here as "monodisperse" elicits, at least in my mind, a narrow colloidal size distribution that is necessarily larger than the smallest

fundamental unit of the colloid). In the methods section of the SI, they further state that while centrifuging for 1h in their two-step method leaves behind bundled SWCNT that can be individualized by further sonication, centrifuging for 2h removes all bundles from the suspension. It is expected that any SWCNT suspension that has been subjected to sonication and centrifugation in SDS contains some fraction of both individualized and bundled (of varying bundle sizes) SWCNT. While the authors have discovered a novel way to manipulate these fractions, it should be acknowledged somewhere in the manuscript that a truly "bundle free" sample is not likely attained, but rather, SWCNT bundles are minimized (not eliminated) by shortening the first centrifugation step and subjecting to further sonication/centrifugation.

- The authors claim that a 1h centrifugation following their initial sonication leaves some bundles in the suspension that can be individualized by further sonication. If this mechanism is accurate, the authors should be able to visually identify SWCNT bundles using electron microscopy following, for example, 9h of sonication of a 1 mg/mL sample and 24h of sonication of a 4 mg/mL sample, and subsequent 1h centrifugation of each. Such evidence would be a compelling argument in favor of their proposed mechanism.

- Fig. 1d shows that the concentration of individualized SWCNT appears to reach a maximum of approximately 1 mg/mL with dispersions of raw SWCNT > 4 mg/mL. One possible reason for this is that, upon redispersion of the suspension following the first centrifugation step, viscosity limitations are again reached due to the high remaining concentration of carbonaceous material, and SWCNT can no longer be individualized. In such cases where raw SWCNT 4 mg/mL, a third (or fourth, etc.) sonication/centrifugation procedure may be necessary to afford continual overcoming of viscosity limited dispersion. As a proof of (or demonstration against) this concept, the authors should attempt performance of additional (3+) sonication/centrifugation iterations for a sample with raw SWCNT of 8 mg/mL.

Theme 2 (pages 8-11): a demonstration of how such suspensions achieve scaled-up isolation of high-purity materials from a relatively high-cost source of unpurified SWCNT

- This section of the manuscript is relatively straightforward and serves as a powerful demonstration of the utility of the novel method described earlier in the work. However, the authors claim the isolation of "single-chirality" SWNT without specifying a quantitative threshold by which they classify a SWCNT suspension as single-chirality. Whatever quantitative threshold the authors are using should be defined early in this section, along with a brief description of the quantitation software used.

- The authors employ a method of single column SWNT purification that relies on perturbation of both surfactant concentration and temperature during adsorption to the gel and perturbation of surfactant type and concentration during elution. This is in contrast to the first-report of gel-based single chirality SWCNT purification reported by Kataura and coworkers (10.1038/ncomms1313) that relied on overloading conditions at room temperature. In the present manuscript, the authors should provide (either in the main text or the SI) a brief comparison of the rationale behind the two methods. For example, because the method employed here can accommodate adsorption/elution of SWNT at both low and high concentrations, it is expected that significant underloading conditions are used. Additionally, an SI video in the work by Kataura demonstrated near 100% elution of SWNT from a gel, while this work reports some SWNT being irreversibly adsorbed to the gel column. A more direct comparison between the two approaches would help readers contextualize this method with the broader field of gel-based SWNT purification.

- The authors utilize, both here and later in the manuscript, a series of SWCNT suspensions at various concentrations. It is unclear if these concentrations were achieved by either 1) subjecting SWCNT/SDS of varying raw material concentration to the same iterative sonication/centrifugation procedure; or 2) dilution from stock of a single, relatively high concentration sample obtained by iterative sonication/centrifugation. The authors should clarify this point, as the second method exclusively investigates SWCNT concentration effects on purification while the first is affected by both concentration and the dynamics of system viscosity during the individualization procedure. In other words, does this section purely explore concentration effects or is it a broader exploration of the iterative procedure carried out at different concentrations?

Theme 3 (pages 11-15): A demonstration of how such suspensions achieve scaled-up isolation of high-purity materials from a relatively low-cost source of unpurified SWCNT

- This section of the manuscript involves novel demonstration of separation of single-chirality SWCNT from a low cost, high impurity stock (a notable achievement). One compelling difference

between utilization of HiPCO and G-SWCNT is that the authors report high chiral purity obtained from purification of low-concentration HiPCO (Fig 2b) but relatively low chiral purity obtained of purification of low-concentration G-SWCNT (Fig 3e). The extent to which this is discussed in the manuscript is as follows: "indicating that the preparation of a high-concentration monodisperse solution is a prerequisite for the separation of single-chirality SWCNTs from G-SWCNTs." While certainly a true statement, I see this as a lost opportunity to gain valuable mechanistic insight into the relationship between SWCNT stock source, SWCNT stock concentration, and chiral purity of eluted SWCNT. Is there some impurity that is present in G-SWCNT (not present in HiPCO) that limits chiral selectivity? If so, what could this be? Is this limitation since, normalized to mass of stock, G-SWCNT contains less SWCNT than HiPCO so there is simply less SWCNT in the low concentrations of G-SWCNT? Regardless, the specific contrast between the chiral purity obtained from the two stocks should be addressed with additional manuscript text and perhaps additional experimentation to enable the authors to comment on the mechanistic underpinnings of this observation.

Theme 4 (pages 15-20): Efforts to probe mechanistic effects of SWCNT concentration on gel-based purification

- While inclusion of this section of the manuscript is appreciated, the mechanistic insight afforded by these results are unclear. Specifically, both loading amount (at constant concentration) and concentration (at constant loading amount) are explored in terms of the total purified (6,4) SWCNT obtained. It would be useful if the authors explained the logic behind these two experimental choices in terms of what scientific questions (or hypotheses led) to these experimental choices. In other words, the authors should provide a framework by which this data can be interpreted. As it stands currently, the conclusion from the data in Fig. 5 is as follows: "the rapid increase in the production of single-chirality SWCNTs caused by loading high-concentration SWCNT solution resulted from the combined effects of the increase in the concentration and loading amount of SWCNTs." While a true statement, this provides minimal scientific or mechanistic insight into why either of these factors should affect the dynamic SWNT/surfactant/gel system in the separation scheme employed. Such mechanistic insight, or at least a framework for interpreting data in Fig. 5, is more consistent with my expectations of a publication in Nature Communications than the existing analysis. Coupling this framework with a schematic image of how/why both loading amount and concentration affect purification efficiency would be useful as well.
- The analysis in Fig. 5b in terms of the "increase rate" is not presented alongside a description of why the increase rate is expected to maximize at one specific SWCNT concentration. From a perspective of system optimization, the present analysis is useful. However, I expect the readership of this journal to seek scientific and mechanistic insight beyond considerations of pure optimization.
- Data in Figs. 5b, 5c, and 5d all include error bars but there is no description of performing the experiment in triplicate and/or what statistical factor is represented in the length of the error bar.
- This section states that "with an increase in SWCNT concentration, the equilibrium between adsorption and desorption shifted to the direction of adsorption." I am skeptical of this explanation because, under isothermal conditions, if the adsorption/desorption process of SWCNT to/from gel was driven by equilibrium, it would be possible to elute adsorbed SWNT through passage of neat surfactant (which would drive the equilibrium to the desorbed state). However, performance of such does not result in eluted SWNT, rather, elution requires introduction of surfactant of higher concentration and/or cosurfactants. Given this, there is likely some other mechanism (other than equilibrium) behind the concentration effects described in Fig. 5.

Theme 5 (pages 20-22): A Life Cycle Assessment estimating the costs of using the reported novel method to produce purified SWNT at scale

- This analysis is outside of my area of expertise, so I will avoid detailed comments on its validity. However, it is nonetheless worthwhile to state that a LCA analysis fits with the scope and scale of the work and is of general interest to those pursuing more effective SWCNT purification methods.

In summary, the manuscript submitted by Liu and coworkers contains compelling results that are likely to be of significant interest to individuals working in the broad field of carbon nanotube science. As it is presently written, my assessment of this work is that it may be considered for

publication in Nature Communications after revision and further review. The overarching theme of my request for revision stems from a desire for the authors to compliment this groundbreaking finding with additional analysis (experimental or otherwise) to better describe the nanoscale mechanism of both improved dispersion concentrations and improved process efficiency of SWNT purification.

I would like to sincerely thank both the editor, for the opportunity to participate in the review process of this work, as well as the authors, for their dedication to advancing this impactful area.

Kevin Tvrdy
Associate Professor
Dept. of Chemistry & Biochemistry
University of Colorado at Colorado Springs

Reviewer #1:.....	2
Comments 1-1:.....	2
Comments 1-2:.....	11
Comments 1-3:.....	12
Reviewer #2:.....	15
Comments 2-1:.....	15
Comments 2-2:.....	25
Comments 2-3:.....	27
Comments 2-4:.....	32
Comments 2-5:.....	38
Reviewer #3:.....	41
Comments 3-1:.....	42
Comments 3-2:.....	45
Comments 3-3:.....	46
Comments 3-4:.....	47
Comments 3-5:.....	49
Comments 3-6:.....	51
Comments 3-7:.....	51
Comments 3-8:.....	54
Comments 3-9:.....	55
Comments 3-10:.....	57
Comments 3-11:.....	63
Comments 3-12:.....	63
Comments 3-13:.....	64
Comments 3-14:.....	68

Response to reviewers

Reviewer #1:

This manuscript reports large-scale purification of single chirality SWCNT species using gel chromatograph method. By increasing the concentration of initial SWCNT dispersion loaded on gels, the authors obtained increased yield for single-chirality species. This result is certainly an improvement over what they reported before. However, there are a few concerns that need to be addressed before I can recommend its publication.

Reply: Thank you for your positive comments.

Comments 1-1: What is the new scientific finding this work offers? Increasing the concentration of initial SWCNT dispersion is expected to increase the purification yield. The authors should make it clear what is new.

Reply: Thank you for your positive comments and constructive suggestions. To make the novelty and technique advancement clearer, we added additional experimental data and discussion in the revised manuscript and supplementary Information. We summarize the main novelty and technical advances as follows:

- (1) The redispersion technique has been developed. With this technique, dispersible initial concentration of SWCNTs and the resulting concentration of individualized SWCNT solution have been increased by several times, which has been demonstrated to be fundamental for improving the separation yield of single-chirality species. The achievement of high-concentration SWCNT dispersion is attributed to the removal of large SWCNT bundles and impurities through the first centrifugation which are difficult to disperse, which greatly reduces the viscosity of SWCNT solution and thus enhances the ultrasonic dispersion efficiency of SWCNTs.
- (2) The separate yield of single-chirality species has been improved by several times by employing high-concentration SWCNT solutions, and thus the milligram scale of multiples single-chirality (n , m) SWCNTs has been achieved, which is increased by more than one order of magnitude compared with previously reported large-scale separation methods.
- (3) Large-scale separation of single-chirality SWCNTs from low cost and industrially produced raw SWCNTs with a wide structural distribution has been achieved by employing high-concentration SWCNT solutions. Compared with conventional low concentration SWCNT dispersion, the separation yield of single-chirality SWCNTs is increased by more than one order of magnitude. We further explored the effect of the SWCNT concentration on the separation efficiency of Tuball SWCNTs. Similarly, the yield of the separated semiconducting SWCNTs increased dramatically by using high-concentration SWCNT dispersion. These

results indicate that increasing the separation yield of SWCNTs by increasing SWCNT concentration is universal for various raw materials.

- (4) The concentration effect of individualized SWCNT solution on the separation yield of single-chirality SWCNTs have been systematically explored. We propose that the distinct improvement in the separation yield of SWCNTs by increasing the concentration of SWCNT solution is mainly ascribed to the enhanced transfer of SWCNTs from bulk solution to the gel surface and thus their adsorption onto gel, which reduces the proportions of unadsorbed and irreversibly adsorbed SWCNTs.
- (5) We demonstrated that the gel column used could be recycled for at least 20 times without significant degradation in performance, which significantly reduced the separation cost of SWCNTs
- (6) The life-cycle assessment (LCA) and techno-economic analysis (TEA) indicate that the separation of single-chirality SWCNTs by high-concentration individualized SWCNT solution dramatically reduced the environmental impact, energy consumption and separation cost.
- (7) Given that sonication and centrifugation processes have been industrially applied, we believe that the current method can further scale up the separation of single-chirality SWCNTs, and reduce environmental impact, energy consumption and separation costs by employing industrial homogenizer, centrifugation and gel column system.

In general, we have developed a redispersion technique to achieve high-concentration individualized SWCNT solution, and thus greatly improved the separation efficiency and yield of single-chirality SWCNTs. Most importantly, the use of high-concentration SWCNT solution for separation of single-chirality SWCNTs not only increases the separation yield of single-chirality species, but also significantly reduces the environmental impact, energy consumption and separation cost. This is very important for the industrialization of single-chirality SWCNTs, which is helpful for promoting the application of SWCNTs in high-end electronics and optoelectronic fields. Clearly, our present work is a great advancement in the SWCNT field.

Introduction, Page 3 Line 62

Previous: The chirality separation of SWCNTs by gel chromatography is based on their selective adsorption onto and desorption from the gel medium^{20,23}. To improve the separation yield of single-chirality SWCNTs, the preparation of high-concentration monodisperse SWCNTs is expected to be an effective method.

Revised: The chirality separation of SWCNTs by gel chromatography is based on their selective adsorption onto and desorption from the gel medium^{20,21}. **Although the separation efficiency of single-chirality species was improved greatly by overloading¹⁷, temperature control¹⁸, mixed surfactants¹⁹ and even their**

combination²⁰⁻²², further scaling up separation yield of (n, m) is still difficult due to insufficient resolution and relatively low efficiency.¹⁷⁻²⁵ To improve the separation yield of single-chirality SWCNTs, the preparation of high-concentration individualized SWCNTs is expected to be an effective method.

Introduction, Page 5, Line 100

Previous: With this technique, the dispersible initial concentration of HiPco-SWCNTs increased from 1 to 8 mg/mL, and the corresponding concentration of the resulting monodisperse SWCNT solution increased from 0.19 to ~1.02 mg/mL.

Revised: With this technique, the dispersible initial concentration of HiPco-SWCNTs increased from 1 to 8 mg/mL, and the corresponding concentration of the resulting individualized SWCNT solution increased from 0.19 to ~1.02 mg/mL. The achievement of high-concentration SWCNT dispersion is attributed to the removal of large SWCNT bundles and impurities through the first centrifugation which are difficult to disperse, which greatly reduces the viscosity of SWCNT solution and thus enhances the ultrasonic dispersion efficiency of SWCNTs.

Introduction, Page 5 Line 119

Previous: Additionally, the diameter separation of large-diameter semiconducting SWCNTs with diameters ranging from 1.2 to 1.7 nm was also achieved on a milligram scale. These results fully demonstrate that the current dispersion technique enables the separation of single-chirality SWCNTs to be highly scalable-up.

Revised: Additionally, the diameter separation of large-diameter semiconducting SWCNTs with diameters ranging from 1.2 to 1.7 nm was also achieved on a milligram scale. The distinct improvement in the separation yield of SWCNTs by increasing the concentration of SWCNT solution is mainly ascribed to the enhanced transfer of SWCNTs from bulk solution to the gel surface and thus their adsorption onto gel, which reduces the proportions of unadsorbed and irreversibly adsorbed SWCNTs.

Introduction, Page 6, Line 128

Previous: Our present dispersion and separation strategy provides a new method for the industrial separation of single-chirality SWCNTs over a wide diameter range.

Revised: Our present dispersion and separation strategy provides a new method for the industrial separation of single-chirality SWCNTs over a wide diameter range. Given that sonication and centrifugation processes have been industrially applied⁴¹⁻⁴³, we believe that the current method can further scale up the separation of single-chirality SWCNTs, and reduce environmental impact, energy consumption and separation costs by employing industrial homogenizer, and centrifugation, and gel

column system.

Results, Page 6, Line 142

Previous: The dispersion time was set as a proportional function of the initial SWCNT concentration (see the experimental details in Supplementary Note 1).

Revised: The dispersion time was set as a proportional function of the initial SWCNT concentration (see the experimental details in Supplementary Note 1), so as to sufficiently disperse SWCNTs to increase the concentration of the resulting SWCNT dispersion (Supplementary Note 2 and Supplementary Figs. 1 and 2). To optimize dispersion time of the SWCNT solutions with different initial concentrations, we studied the variation of their viscosity and concentration with time during dispersion process (Supplementary Fig. 2). At the beginning, the viscosity of SWCNT solution increased rapidly and then reached the maximum values with increasing sonication duration. With continuously increasing sonication time, the viscosity of SWCNT solutions decreased dramatically and finally reached an approximate constant value. We can imagine that at the beginning, SWCNT powders were crushed into particles composed of large bundles and impurities under ultrasonic dispersion. Due to the low dispersity and increased number of suspended particles, the friction between these particles increased the viscosity of solutions. As the sonication time increased, SWCNTs were continuously stripped from the bundles. The sidewalls of SWCNTs exposed to the solution were readily coated with surfactant molecules, decreasing the friction due to high lubrication effect of surfactants⁴⁴. Meanwhile, denser surfactant coatings around SWCNTs provided a repulsive region and thus excellent fluidity⁴⁵. Therefore, when the viscosity of a SWCNT solution dropped to a stable value after ultrasonic dispersion, it should be sufficiently dispersed, which was evidenced by its concentration evolution over sonication time (Supplementary Note 2 and Supplementary Fig. 2).

Results, Page 10, Line 251

Previous: The detailed separation process is described in the experimental section (Supplementary Note 1 and Supplementary Fig. 4).

Revised: The detailed separation process is described in the experimental section (Supplementary Note 1 and Supplementary Fig. 4). Compared with previous works¹⁷⁻²⁵, the current separation strategy showed higher resolution on SWCNT structures and higher separation efficiency (Supplementary Note 4 and Supplementary Fig. 5).

Results, Page 13, Line 306

Previous: Notably, although the dispersion time increased with an increase in the initial concentration of dispersible SWCNTs, the increase in ultrasonic time did not significantly decrease the length of SWCNTs (Supplementary Fig. 9).

Revised: Notably, although the dispersion time increased with an increase in the initial concentration of dispersible SWCNTs, the increase in ultrasonic time did not significantly decrease the length of SWCNTs (Supplementary Fig. 9). **Another important feature of the current method is that the gel column used could be recycled for at least 20 times without significant degradation in performance, which significantly reduced the separation cost of SWCNTs (Supplementary Fig.10).**

Discussion, Page 21, Line 484

Previous: The results show that a lower concentration of the loaded SWCNT solution with a fixed loading amount led to a higher proportion of unadsorbed SWCNTs (Supplementary Fig. 15). With an increase in the SWCNT concentration, the equilibrium between adsorption and desorption shifted to the direction of adsorption^{26,27,41}, resulting in an increase in the reversibly adsorbed SWCNTs and a decrease in the proportion of unadsorbed and irreversibly adsorbed SWCNTs, which inevitably increased the separation yield of SWCNTs.

Revised: The results show that a lower concentration of the loaded SWCNT solution with a fixed loading amount led to a higher proportion of unadsorbed SWCNTs (**Supplementary Fig. 21**). **Similar to the separation of biomolecules in a chromatographic system^{32,51}, the concentration of SWCNT solution was a key parameter, which determined the mass transfer of SWCNTs from the bulk solution to the boundary layer adjacent to gel surface and thus the SWCNT concentration in the reaction volume (Supplementary Note 8 and **Supplementary Fig. 22**). Based on the adsorption kinetics of molecules^{30,31,52}, a high SWCNT concentration in the reaction volume would promote the adsorption of SWCNT onto gel surface. At low concentration, the mass transfer of SWCNTs to the gel surface was hindered due to small SWCNT concentration gradient between bulk solution and boundary layer, and thus decreasing their adsorption onto gel surfaces. Most of SWCNTs remained in the bulk solution and flew through the gel column as unabsorbed species. In contrast, with increasing SWCNT concentration, the transfer of SWCNTs to the gel surface was enhanced, providing more SWCNTs for the binding reactions and thus reducing the proportion of unadsorbed SWCNTs. As the SWCNT concentration further increased, excessive SWCNT could be transferred to the gel surface and accumulated in the reaction volume within a short time, the adsorption kinetics turned to adsorption control. Therefore, further increasing the SWCNTs concentration, the adsorption amount of SWCNTs tended to saturate and the curve slopes in Fig. 5c decreased. A detailed discussion is presented in the Supplementary Note 8.**

Discussion, Page 23, Line 530

Previous: With an increase in the loading amount, the proportion of irreversibly adsorbed SWCNTs decreased rapidly from 70% to ~20%, which originated from the quasirandom irreversible adsorption in gel chromatography⁴². A detailed discussion is presented in Supplementary Note 5.2.

Revised: With an increase in the loading amount, the proportion of irreversibly adsorbed SWCNTs decreased rapidly from 70% to ~15%, which originated from the quasirandom irreversible adsorption in gel chromatography⁵³. Compared with reversible adsorption, irreversible adsorption sites exhibited stronger affinity with SWCNTs. When SWCNTs flew through gel column, they preferentially adsorbed at the irreversible adsorption sites. Conceptually, a gel column can be considered as being composed of many thin layers of gel⁵⁴. When a small amount of SWCNT solution were loaded into a gel column, SWCNTs could be adsorbed by both reversible and irreversible adsorption sites in the upper layers of the gel column, leaving the irreversible adsorption sites in the lower layers unoccupied. During elution, the eluted SWCNTs from reversible adsorption sites may be captured again by the irreversible adsorption sites in the lower layers (Supplementary Fig. 25), leading to a significant decrease in the SWCNT concentration in the eluted solution and even no SWCNT were collected. With an increase the loading amount of SWCNTs, more and more SWCNTs could be collected in the eluted solution because of the occupation of more irreversible adsorption sites, resulting in a dramatic decrease in the proportion of irreversible adsorption and thus a rapid increase in the separation yield of SWCNTs (Fig. 5d and Supplementary Fig. 25). Notably, as the loading amount continuously increased, more reversibly adsorbed SWCNTs were eluted while the probability of irreversible adsorption of SWCNTs also increased, leading to a constant proportion of irreversible adsorption. In this way, the separation yield began to increase linearly with an increase in the loading amount (Fig. 5d). A detailed discussion was presented in Supplementary Note 9.

Discussion, Page 23, Line 576

Previous: Life cycle assessment (LCA) is very important for the industrial production of a new material.

Revised: Life cycle assessment (LCA) is very important for the industrial production of a new material. Given that large-scale production data of single-chirality SWCNTs are lacking, LCA is essential for predicting the impacts of scaling up and assessing the technological readiness of the emerging methods^{56,57}. Moreover, the LCA may provide a benchmark as well as a reference to evaluate the advantages and limitations of a separation method.

Discussion, Page 26, Line 623

Previous: The separation costs of such an amount of single-chirality species from high-concentration HiPco- and G-SWCNTs are \$0.8 and \$3.8, respectively, which are comparable to that of a 4-inch silicon wafer.

Revised: The separation costs of such an amount of single-chirality species from high-concentration HiPco- and G-SWCNTs are \$0.8 and \$3.8, respectively, which are comparable to that of a 4-inch silicon wafer. Overall, the separation efficiency using high-concentration SWCNT solution is distinctly higher compared with conventional 1-mg/mL samples, despite that the size of gel column was increased, which reduced time, energy and material consumption and thus the separation cost. Given that sonication and ultracentrifugation process has been applied the chemical synthesis industry^{41,43} and also large volumes of SWCNT dispersion⁴², we can expect that the separation efficiency of single-chirality species could be further increased while the separation cost, environmental impact and energy consumption were further reduced by scaling up with industrial dispersion, centrifugation and gel column system (as discussed in **Supplementary Note 10.3**)⁵⁸.

Supplementary Information, Page 35 Line 547

Previous: As shown in Supplementary Figure 15d, the proportion of the irreversible adsorption decreases slightly with decreasing SWCNT concentration under such a fixed loading amount.

Revised: As shown in **Supplementary Fig. 21d**, the proportion of the irreversible adsorption decreases slightly with decreasing SWCNT concentration under such a fixed loading amount, while the proportion of unadsorbed SWCNTs increased rapidly.

Similar to biomolecules¹⁰, the adsorption process of SWCNTs to gel can be described as follow:

where $SWCNT_{n,m}^{free}$ and $SWCNT_{n,m}^{ads}$ represent the number of free (n, m) species and the number of adsorbed (n, m) species in a reaction volume of gel, respectively. θ is the total number of adsorption sites in this area. $K_{n,m}^f$ represent the forward rate constants, which is determined by the interaction between a (n, m) SWCNT and an unoccupied binding site¹¹. At low concentrations, the number of each (n, m) species in the reaction volume is low, leading to a low binding rate. The change of concentration of (n, m) species with time reveal the binding rate¹¹:

$$-\frac{dC_{n,m}(t)}{dt} = K_{n,m}^f C_{n,m}(t) \left[\frac{\theta}{V} - \Sigma \left(C_{n,m}(t_0) - C_{n,m}(t) \right) \right] \quad (2)$$

At a given time, the rate of change of $C_{n,m}(t)$ nonlinearly decreases with $C_{n,m}(t)$. It

suggests that the binding of SWCNTs become harder as the concentration decreases.

As shown in supplementary Fig. 22, in a given SWCNTs/gel system, the SWCNT solution can be divided into bulk solution and boundary layer adjacent to gel surface. Due to the fast adsorption of SWCNTs, the concentration of SWCNTs in boundary layer $C_{n,m}^*$ is smaller than that in the bulk solution $C_{n,m}$, forming a concentration gradient, which drives transfer of SWCNTs from bulk solution to boundary layer.^{12, 13} The simplest expression of the relationship between the flux of a (n, m) species and the “driving force” of mass transfer is as follows¹⁰:

$$J = D_{n,m}(C_{n,m} - C_{n,m}^*)/\delta \quad (3)$$

where $D_{n,m}$ is the diffusion coefficient of the specific SWCNT/surfactant hybrid¹³, δ is the thickness of boundary layer determined by the flow rate of the solution and geometry of gel beads. At low concentration, due to the small difference in the concentrations of $C_{n,m}$ and $C_{n,m}^*$, small concentration gradient hinders the mass transfer of SWCNTs to the gel surface and thus decreasing their adsorption onto gel surfaces due to low concentration in the reaction volume adjacent to the gel surface. SWCNTs tend to remain in the bulk solution and may flow through the gel, leading to more unadsorbed SWCNTs, as shown by the yellow line in Supplementary Fig. 22.

As the concentration in the bulk solution increases, the difference between $C_{n,m}$ and $C_{n,m}^*$ increases. The transfer of SWCNTs to the gel surface has been enhanced, providing more SWCNTs for the binding reactions, as shown by the red line in Supplementary Fig. 22, which reduce the proportion of unadsorbed SWCNTs. Then, as excessive SWCNTs were transferred to the gel surface in a short time, the adsorption kinetics turn to adsorption-control. The SWCNTs adjacent to the gel surface accumulate, which increase $C_{n,m}^*$ and decreases the concentration gradient in the boundary layer. At this stage, the resistance of SWCNT binding is mainly attributed to the adsorption process (red line in Supplementary Fig. 22). Therefore, further increasing the SWCNTs concentration, the adsorption amount of SWCNTs tend to saturate. Notably, this model is very simple and does not involve interactions between different (n, m) species and SWCNT transfer inside gel beads.

Supplementary Figure 22. Schematic diagram of mass transfer from the bulk SWCNT solution to gel surface.

Supplementary Information, Page 41, Line 659

Previous: Although the adsorption condition of semiconducting SWCNTs was not fulfilled at the concentration of 5 wt% SDS, irreversible adsorption was still observed, which may relate to the inhomogeneity of gel¹⁵⁻¹⁸.

Revised: Although the adsorption condition of semiconducting SWCNTs was not fulfilled at the concentration of 5 wt% SDS, irreversible adsorption was still observed, which may relate to the inhomogeneity of gel¹⁴⁻¹⁶. Similar to equation (1), irreversible binding process regardless of the environment condition can be described as follow:

where θ_{Ir} and $K_{n,m}^{Ir}$ represent the number of binding sites of irreversible adsorption in the reaction volume and the forward rate constant of irreversible adsorption, respectively. Compared with reversible, irreversible adsorption sites exhibits stronger affinity with SWCNTs. When SWCNTs flow through gel column, they preferentially adsorb at the irreversible adsorption sites. During the elution process, eq. 1 proceeds in the reverse direction, while eq. 4 is still a forward reaction. Thus, SWCNTs eluted from reversible sites at the top of the gel column flow down and may be captured by irreversible sites at the bottom of the gel column.

Conceptually, a gel column is usually considered a pile composed of many plates, each of which composes of a very thin gel layer¹⁷. Adsorption of SWCNTs in the upper plates leads to the occupation of both reversible and irreversible adsorption sites. Then, the remained SWCNTs in solution flow down to the plates below. At low loading amounts, only plates at the top were filled. The solution that flows down contains much fewer SWCNTs, leaving the irreversible adsorption sites in the lower plates unoccupied. These unoccupied sites may trap SWCNTs that are eluted from the upper plates. As a result, nearly no SWCNTs were collected from the eluent. In contrast, when a large number of SWCNTs were loaded into the column, irreversible adsorption sites in the lower plates were filled. Thus, when the eluent flows through the column, the remained SWCNTs can be eluted from the lower plates. With an increase in loading amount, the proportion of irreversible adsorption decreases. This hypothesis coincides with the experimental results in Supplementary Fig. 10, where the amount of SWCNTs separated from a gel recycled for 5 separation runs is higher than that of the new gel, because some of the irreversible adsorption sites have been filled in the recycled gel.

Interestingly, the ratio of reversible and irreversible adsorption gradually reached a plateau with further increasing loading amount, as shown in Supplementary Fig. 24d. As mentioned above, with increasing the loading amount of SWCNTs, the irreversible adsorption sites are occupied, resulting in a gradual decrease in the proportion of irreversible adsorption. However, increasing the loading amount also increase the possibility of irreversible adsorption of SWCNTs. For example, the impurities in SWCNT solutions such as amorphous carbon and bundles also increased with loading amount and form more irreversible adsorption¹⁴. The two aspects eventually hold the proportion of irreversible adsorption at $\sim 15\%$. Notably, this mechanism is established under the condition of normal loading. Overloading is supposed to be more complicated, involving the competition between different (n, m) species and the saturation of gel. Additionally, the form of irreversible adsorption may also be attributed to the inherent properties of some SWCNTs, such as length, defects and coatings¹⁴.

Comments 1-2: What is the impact of higher dispersion concentration on gel column's life time and performance repeatability from one run to the next? These issues are important in "industrial" production.

Reply: Thank you for your constructive advice. We have added the performance repeatability of gel column by using the SWCNT solution with initial concentrations of 4 mg/mL. We found that the gel column used could be recycled for at least 20 times without significant degradation in performance. We added the corresponding experimental data in the revised manuscript and supplementary information.

Results, Page 13, Line 306

Previous: Notably, although the dispersion time increased with an increase in the initial concentration of dispersible SWCNTs, the increase in ultrasonic time did not significantly decrease the length of SWCNTs (Supplementary Fig. 9).

Revised: Notably, although the dispersion time increased with an increase in the initial concentration of dispersible SWCNTs, the increase in ultrasonic time did not significantly decrease the length of SWCNTs (Supplementary Fig. 9). **Another important feature of the current method is that the gel column used could be recycled for at least 20 times without significant degradation in performance, which significantly reduced the separation cost of SWCNTs (Supplementary Fig.10).**

Supplementary Information Page 22

Previous: None

Revised:

Supplementary Figure 10. The impact of high-concentration SWCNT solutions on gel column's life time and performance. 10 mL of SWCNT solutions with the initial concentration of 4 mg/mL were separated by columns filled with 40 mL of gel. The gel was recycled for 5, 10 and 20 separation runs. Optical absorption spectra of the separated (6, 4) SWCNTs are presented. The purities of the separated (6, 4) SWCNTs were not affected by the recycling of gel. However, the yield decreases slightly with increasing recycle times. After the column was recycled for 20 times, the mass of eluted SWCNT was decreased by approximately 30% compared with the new gel in a separation round. These results indicate that the gel column can be recycled at least 20 times.

Comments 1-3: The authors claim that they tested “a variety of raw materials”, but the manuscript reported only two: Hipco and a synthetic mixture of graphene and

SWCNTs. The latter is very interesting but not widely available. Is it similar to OCSiAl's Tuball material? Have you tested Tuball?

Reply: Thank you for your constructive suggestion. We chose G-SWCNT for separation mainly because their diameter distribution is very wide without enrichment in a specific diameter, and also, they contain a large amount of graphene flakes. It is difficult to separate single-chirality SWCNTs from this raw material by using conventional low-concentration SWCNT solution. If our current technique is applicable to this material, other materials should also be able to be separated.

To verify whether our current technique is suitable for Tuball material, we did supplementary experiments. Tuball material is enriched in the diameter range of 1.2-2.0 nm. Currently, it is difficult to separate single-chirality SWCNTs from large diameter SWCNT raw materials (>1.2 nm). Therefore, we mainly verified the separation efficiency of semiconducting SWCNTs from this material. Similarly, the separation efficiency of semiconducting SWCNTs have been improved greatly, as shown in Supplementary Fig. 19. We added the corresponding data and discussion in the revised manuscript and supplementary Information.

Results, Page 17, Line 404

Previous: These fractions are productive materials for further separation and applications.

Revised: These fractions are productive materials for further separation and applications. To further demonstrate the universality of the present method, we further applied it to another commercial large-diameter SWCNT material (OCSiAl's Tuball, 1.2-2.0 nm in diameter, Sigma-Aldrich). Since their diameter is larger than 1.2 nm, the separation of single-chirality SWCNTs cannot be achieved for the moment. We explored the separation of metallic/semiconducting SWCNTs from this material. The results also showed that with the present redispersion technique, high-concentration individualized SWCNT solution was achieved, which greatly improved the separation yield of semiconducting SWCNTs (Supplementary Fig. 19).

Supplementary Information Page 2, Line 43

Previous: The dispersion of high-concentration G-SWCNTs (GNH-1200 Beijing North Guoneng Technology Co., Ltd) was performed by the same process as HiPco-SWCNTs.

Revised: The dispersion of high-concentration G-SWCNTs (GNH-1200 Beijing North Guoneng Technology Co., Ltd) and Tuball SWCNT (1.2-2.0 nm in diameter, Sigma-Aldrich) was performed by the same process as HiPco-SWCNTs.

Supplementary Information Page 31

Previous: None

Revised:

Supplementary Figure 19. Preparation of high-concentration Tuball SWCNT solution. (a) Optical absorption spectra of as-prepared Tuball SWCNT solutions with different initial concentrations. The preparation method is similar with that of G-SWCNTs. (b) Optical absorption spectra of semiconducting SWCNTs separated from Tuball SWCNT solutions. The semiconducting SWCNTs were adsorbed at 0.25 wt% SC/ 0.5 wt% SDS at 25 °C. The absence of M_{11} peaks indicates that the separated semiconducting SWCNTs have high purity, implying a high dispersity of the as-prepared SWCNT solution. The eluted semiconducting SWCNTs were diluted to 40 mL for comparison. The difference in absorption absorbance suggests the yield of semiconducting SWCNTs increased by approximate 4 times when using Tuball SWCNT solution with an initial concentration of 4-mg/mL.

Reviewer #2:

The manuscript by Liu and co-workers describes the use of gel chromatography to purify highly concentrated solutions of single-walled carbon nanotubes into pure (single chirality) samples of numerous nanotube species. Using multiple rounds of ultrasonic dispersion, ultracentrifugation, and size-exclusion gel chromatography through Sephacryl S-200. While it is clear that the results obtained from this work are highly interesting to the community of researchers working on carbon nanotube purification, there are some concerns about the novelty and industrial applicability of the work. Some specific comments are provided below.

Reply: Thank you for your positive comments.

Comments 2-1: The authors claim that they are reporting a novel strategy for dispersing and purifying SWCNTs. However, there has been extensive literature on the purification of single-chirality SWCNTs by gel filtration through Sephacryl S-200 (references 17, 18, and 19). The authors use different elution temperatures to separate different SWCNTs, which has also been reported in reference 23. Some seminal reports of Sephacryl-based SWCNT and DWCNT purification are also missing from the references, including:

1. Flavel, B. S., Kappes, M. M., Krupke, R. & Hennrich, F. Separation of Single-Walled Carbon Nanotubes by 1-Dodecanol-Mediated Size Exclusion Chromatography. *Organic Letters* 7, 3557–3564 (2013).
2. Moore, K. E. et al. Separation of Double-Walled Carbon Nanotubes by Size Exclusion Column Chromatography. *ACS Nano* 8, 6756–6764 (2014).
3. Flavel, B. S., Moore, K. E., Pfohl, M., Kappes, M. M. & Hennrich, F. Separation of Single-Walled Carbon Nanotubes with a Gel Permeation Chromatography System. *ACS Nano* 8, 1817–1826 (2014).

Although the authors did demonstrate that this separation method can be scaled up significantly relative to prior reports, the main difference here is an additional sonication and centrifugation step. This does not seem to be at the level of novelty required to warrant publication of the work in this journal.

Reply: Thank you for your comments. According to your suggestion, we added the missing references. Industrial production of single-chirality SWCNTs with identical properties has long been a goal in the field of carbon nanomaterials. However, so far, various liquid phase separation techniques, including gel chromatography, have not achieved the industrial preparation of single chirality SWCNTs, indicating that it is extremely difficult to realize such a goal. Gel chromatography has been widely studied for the separation of single-chirality SWCNTs due to the advantages of the simplicity, high efficiency and low cost. In the present work, by developing redispersion technique to increasing the concentration of SWCNT solution, we achieved the milligram scale separation of single-chirality SWCNTs, which is increased by more than one order of magnitude compared with previously reported

large-scale separation. More importantly, the use of high-concentration SWCNT solution for separation of single-chirality SWCNTs not only increases the separation yield of single-chirality species, but also significantly reduces the environmental impact, energy consumption and separation cost. This is very important for the industrialization of single-chirality SWCNTs, which is helpful for promoting the application of SWCNTs in high-end electronics and optoelectronic fields. Clearly, our present work is a great advancement in the SWCNT field. We believe that our present work will stimulate great interest to researchers working in the diverse and multi-disciplinary areas including materials, chemistry, electronics, optics, optoelectronic integration and bioimaging, etc. and is suitable for publication in Nature Communications.

The main novelty and technical advances of our present work are listed as follows:

- 1) The redispersion technique has been developed. With this technique, dispersible initial concentration of SWCNTs and the resulting concentration of individualized SWCNT solution have been increased by several times, which has been demonstrated to be fundamental for improving the separation yield of single-chirality species. The achievement of high-concentration SWCNT dispersion is attributed to the removal of large SWCNT bundles and impurities through the first centrifugation which are difficult to disperse, which greatly reduces the viscosity of SWCNT solution and thus enhances the ultrasonic dispersion efficiency of SWCNTs.
- 2) The separate yield of single-chirality species has been improved by several times by employing high-concentration SWCNT solutions, and thus the milligram scale of multiples single-chirality (n , m) SWCNTs has been achieved, which is increased by more than one order of magnitude compared with previously reported large-scale separation methods.
- 3) Large-scale separation of single-chirality SWCNTs from low cost and industrially produced raw SWCNTs with a wide structural distribution has been achieved by employing high-concentration SWCNT solutions. Compared with conventional low concentration SWCNT dispersion, the separation yield of single-chirality SWCNTs is increased by more than one order of magnitude. We further explored the effect of the SWCNT concentration on the separation efficiency of Tuball SWCNTs. Similarly, the yield of the separated semiconducting SWCNTs increased dramatically by using high-concentration SWCNT dispersion. These results indicate that increasing the separation yield of SWCNTs by increasing SWCNT concentration is universal for various raw materials.
- 4) The concentration effect of individualized SWCNT solution on the separation yield of single-chirality SWCNTs have been systematically explored. We propose that the distinct improvement in the separation yield of SWCNTs by increasing the concentration of SWCNT solution is mainly ascribed to the enhanced transfer of SWCNTs from bulk solution to the gel surface and thus their adsorption onto gel, which reduces the proportions of unadsorbed and irreversibly adsorbed

SWCNTs.

- 5) We demonstrated that the gel column used could be recycled for at least 20 times without significant degradation in performance, which significantly reduced the separation cost of SWCNTs
- 6) The life-cycle assessment (LCA) and techno-economic analysis (TEA) indicate that the separation of single-chirality SWCNTs by high-concentration individualized SWCNT solution dramatically reduced the environmental impact, energy consumption and separation cost.
- 7) Given that sonication and centrifugation processes have been industrially applied, we believe that the current method can further scale up the separation of single-chirality SWCNTs, and reduce environmental impact, energy consumption and separation costs by employing industrial homogenizer, centrifugation and gel column system.

To make the novelty and technique advancement clearer, we added additional experimental data and discussion in the revised manuscript and supplementary Information.

Introduction, Page 3 Line 62

Previous: The chirality separation of SWCNTs by gel chromatography is based on their selective adsorption onto and desorption from the gel medium^{20,23}. To improve the separation yield of single-chirality SWCNTs, the preparation of high-concentration monodisperse SWCNTs is expected to be an effective method.

Revised: The chirality separation of SWCNTs by gel chromatography is based on their selective adsorption onto and desorption from the gel medium^{20,21}. **Although the separation efficiency of single-chirality species was improved greatly by overloading¹⁷, temperature control¹⁸, mixed surfactants¹⁹ and even their combination²⁰⁻²², further scaling up separation yield of (n, m) is still difficult due to insufficient resolution and relatively low efficiency.¹⁷⁻²⁵** To improve the separation yield of single-chirality SWCNTs, the preparation of high-concentration **individualized** SWCNTs is expected to be an effective method.

Introduction, Page 5, Line 100

Previous: With this technique, the dispersible initial concentration of HiPco-SWCNTs increased from 1 to 8 mg/mL, and the corresponding concentration of the resulting monodisperse SWCNT solution increased from 0.19 to ~1.02 mg/mL.

Revised: With this technique, the dispersible initial concentration of HiPco-SWCNTs increased from 1 to 8 mg/mL, and the corresponding concentration of the resulting **individualized** SWCNT solution increased from 0.19 to ~1.02 mg/mL. **The**

achievement of high-concentration SWCNT dispersion is attributed to the removal of large SWCNT bundles and impurities through the first centrifugation which are difficult to disperse, which greatly reduces the viscosity of SWCNT solution and thus enhances the ultrasonic dispersion efficiency of SWCNTs.

Introduction, Page 5 Line 119

Previous: Additionally, the diameter separation of large-diameter semiconducting SWCNTs with diameters ranging from 1.2 to 1.7 nm was also achieved on a milligram scale. These results fully demonstrate that the current dispersion technique enables the separation of single-chirality SWCNTs to be highly scalable-up.

Revised: Additionally, the diameter separation of large-diameter semiconducting SWCNTs with diameters ranging from 1.2 to 1.7 nm was also achieved on a milligram scale. The distinct improvement in the separation yield of SWCNTs by increasing the concentration of SWCNT solution is mainly ascribed to the enhanced transfer of SWCNTs from bulk solution to the gel surface and thus their adsorption onto gel, which reduces the proportions of unadsorbed and irreversibly adsorbed SWCNTs.

Introduction, Page 6, Line 128

Previous: Our present dispersion and separation strategy provides a new method for the industrial separation of single-chirality SWCNTs over a wide diameter range.

Revised: Our present dispersion and separation strategy provides a new method for the industrial separation of single-chirality SWCNTs over a wide diameter range. Given that sonication and centrifugation processes have been industrially applied⁴¹⁻⁴³, we believe that the current method can further scale up the separation of single-chirality SWCNTs, and reduce environmental impact, energy consumption and separation costs by employing industrial homogenizer, and centrifugation, and gel column system.

Results, Page 6, Line 142

Previous: The dispersion time was set as a proportional function of the initial SWCNT concentration (see the experimental details in Supplementary Note 1).

Revised: The dispersion time was set as a proportional function of the initial SWCNT concentration (see the experimental details in Supplementary Note 1), so as to sufficiently disperse SWCNTs to increase the concentration of the resulting SWCNT dispersion (Supplementary Note 2 and Supplementary Figs. 1 and 2). To optimize dispersion time of the SWCNT solutions with different initial concentrations, we studied the variation of their viscosity and concentration with time during dispersion

process (Supplementary Fig. 2). At the beginning, the viscosity of SWCNT solution increased rapidly and then reached the maximum values with increasing sonication duration. With continuously increasing sonication time, the viscosity of SWCNT solutions decreased dramatically and finally reached an approximate constant value. We can imagine that at the beginning, SWCNT powders were crushed into particles composed of large bundles and impurities under ultrasonic dispersion. Due to the low dispersity and increased number of suspended particles, the friction between these particles increased the viscosity of solutions. As the sonication time increased, SWCNTs were continuously stripped from the bundles. The sidewalls of SWCNTs exposed to the solution were readily coated with surfactant molecules, decreasing the friction due to high lubrication effect of surfactants⁴⁴. Meanwhile, denser surfactant coatings around SWCNTs provided a repulsive region and thus excellent fluidity⁴⁵. Therefore, when the viscosity of a SWCNT solution dropped to a stable value after ultrasonic dispersion, it should be sufficiently dispersed, which was evidenced by its concentration evolution over sonication time (Supplementary Note 2 and Supplementary Fig. 2).

Results, Page 10, Line 251

Previous: The detailed separation process is described in the experimental section (Supplementary Note 1 and Supplementary Fig. 4).

Revised: The detailed separation process is described in the experimental section (Supplementary Note 1 and Supplementary Fig. 4). **Compared with previous works¹⁷⁻²⁵, the current separation strategy showed higher resolution on SWCNT structures and higher separation efficiency (Supplementary Note 4 and Supplementary Fig. 5).**

Results, Page 13, Line 306

Previous: Notably, although the dispersion time increased with an increase in the initial concentration of dispersible SWCNTs, the increase in ultrasonic time did not significantly decrease the length of SWCNTs (Supplementary Fig. 9).

Revised: Notably, although the dispersion time increased with an increase in the initial concentration of dispersible SWCNTs, the increase in ultrasonic time did not significantly decrease the length of SWCNTs (Supplementary Fig. 9). **Another important feature of the current method is that the gel column used could be recycled for at least 20 times without significant degradation in performance, which significantly reduced the separation cost of SWCNTs (Supplementary Fig.10).**

Discussion, Page 21, Line 484

Previous: The results show that a lower concentration of the loaded SWCNT solution with a fixed loading amount led to a higher proportion of unadsorbed SWCNTs (Supplementary Fig. 15). With an increase in the SWCNT concentration, the equilibrium between adsorption and desorption shifted to the direction of adsorption^{26,27,41}, resulting in an increase in the reversibly adsorbed SWCNTs and a decrease in the proportion of unadsorbed and irreversibly adsorbed SWCNTs, which inevitably increased the separation yield of SWCNTs.

Revised: The results show that a lower concentration of the loaded SWCNT solution with a fixed loading amount led to a higher proportion of unadsorbed SWCNTs (Supplementary Fig. 21). Similar to the separation of biomolecules in a chromatographic system^{32,51}, the concentration of SWCNT solution was a key parameter, which determined the mass transfer of SWCNTs from the bulk solution to the boundary layer adjacent to gel surface and thus the SWCNT concentration in the reaction volume (Supplementary Note 8 and Supplementary Fig. 22). Based on the adsorption kinetics of molecules^{30,31,52}, a high SWCNT concentration in the reaction volume would promote the adsorption of SWCNT onto gel surface. At low concentration, the mass transfer of SWCNTs to the gel surface was hindered due to small SWCNT concentration gradient between bulk solution and boundary layer, and thus decreasing their adsorption onto gel surfaces. Most of SWCNTs remained in the bulk solution and flew through the gel column as unadsorbed species. In contrast, with increasing SWCNT concentration, the transfer of SWCNTs to the gel surface was enhanced, providing more SWCNTs for the binding reactions and thus reducing the proportion of unadsorbed SWCNTs. As the SWCNT concentration further increased, excessive SWCNT could be transferred to the gel surface and accumulated in the reaction volume within a short time, the adsorption kinetics turned to adsorption control. Therefore, further increasing the SWCNTs concentration, the adsorption amount of SWCNTs tended to saturate and the curve slopes in Fig. 5c decreased. A detailed discussion is presented in the Supplementary Note 8.

Discussion, Page 23, Line 530

Previous: With an increase in the loading amount, the proportion of irreversibly adsorbed SWCNTs decreased rapidly from 70% to ~20%, which originated from the quasirandom irreversible adsorption in gel chromatography⁴². A detailed discussion is presented in Supplementary Note 5.2.

Revised: With an increase in the loading amount, the proportion of irreversibly adsorbed SWCNTs decreased rapidly from 70% to ~15%, which originated from the quasirandom irreversible adsorption in gel chromatography⁵³. Compared with reversible adsorption, irreversible adsorption sites exhibited stronger affinity with SWCNTs. When SWCNTs flew through gel column, they preferentially adsorbed at

the irreversible adsorption sites. Conceptually, a gel column can be considered as being composed of many thin layers of gel⁵⁴. When a small amount of SWCNT solution were loaded into a gel column, SWCNTs could be adsorbed by both reversible and irreversible adsorption sites in the upper layers of the gel column, leaving the irreversible adsorption sites in the lower layers unoccupied. During elution, the eluted SWCNTs from reversible adsorption sites may be captured again by the irreversible adsorption sites in the lower layers (Supplementary Fig. 25), leading to a significant decrease in the SWCNT concentration in the eluted solution and even no SWCNT were collected. With an increase the loading amount of SWCNTs, more and more SWCNTs could be collected in the eluted solution because of the occupation of more irreversible adsorption sites, resulting in a dramatic decrease in the proportion of irreversible adsorption and thus a rapid increase in the separation yield of SWCNTs (Fig. 5d and Supplementary Fig. 25). Notably, as the loading amount continuously increased, more reversibly adsorbed SWCNTs were eluted while the probability of irreversible adsorption of SWCNTs also increased, leading to a constant proportion of irreversible adsorption. In this way, the separation yield began to increase linearly with an increase in the loading amount (Fig. 5d). A detailed discussion was presented in Supplementary Note 9.

Discussion, Page 23, Line 576

Previous: Life cycle assessment (LCA) is very important for the industrial production of a new material.

Revised: Life cycle assessment (LCA) is very important for the industrial production of a new material. Given that large-scale production data of single-chirality SWCNTs are lacking, LCA is essential for predicting the impacts of scaling up and assessing the technological readiness of the emerging methods^{56,57}. Moreover, the LCA may provide a benchmark as well as a reference to evaluate the advantages and limitations of a separation method.

Discussion, Page 26, Line 623

Previous: The separation costs of such an amount of single-chirality species from high-concentration HiPco- and G-SWCNTs are \$0.8 and \$3.8, respectively, which are comparable to that of a 4-inch silicon wafer.

Revised: The separation costs of such an amount of single-chirality species from high-concentration HiPco- and G-SWCNTs are \$0.8 and \$3.8, respectively, which are comparable to that of a 4-inch silicon wafer. Overall, the separation efficiency using high-concentration SWCNT solution is distinctly higher compared with conventional 1-mg/mL samples, despite that the size of gel column was increased, which reduced time, energy and material consumption and thus the separation cost. Given that sonication and ultracentrifugation process has been applied the chemical synthesis

industry^{41,43} and also large volumes of SWCNT dispersion⁴², we can expect that the separation efficiency of single-chirality species could be further increased while the separation cost, environmental impact and energy consumption were further reduced by scaling up with industrial dispersion, centrifugation and gel column system (as discussed in **Supplementary Note 10.3**)⁵⁸.

Supplementary Information, Page 35 Line 547

Previous: As shown in Supplementary Figure 15d, the proportion of the irreversible adsorption decreases slightly with decreasing SWCNT concentration under such a fixed loading amount.

Revised: As shown in **Supplementary Fig. 21d**, the proportion of the irreversible adsorption decreases slightly with decreasing SWCNT concentration under such a fixed loading amount, **while the proportion of unadsorbed SWCNTs increased rapidly.**

Similar to biomolecules¹⁰, the adsorption process of SWCNTs to gel can be described as follow:

where $SWCNT_{n,m}^{free}$ and $SWCNT_{n,m}^{ads}$ represent the number of free (n, m) species and the number of adsorbed (n, m) species in a reaction volume of gel, respectively. θ is the total number of adsorption sites in this area. $K_{n,m}^f$ represent the forward rate constants, which is determined by the interaction between a (n, m) SWCNT and an unoccupied binding site¹¹. At low concentrations, the number of each (n, m) species in the reaction volume is low, leading to a low binding rate. The change of concentration of (n, m) species with time reveal the binding rate¹¹:

$$-\frac{dC_{n,m}(t)}{dt} = K_{n,m}^f C_{n,m}(t) \left[\frac{\theta}{V} - \Sigma (C_{n,m}(t_0) - C_{n,m}(t)) \right] \quad (2)$$

At a given time, the rate of change of $C_{n,m}(t)$ nonlinearly decreases with $C_{n,m}(t)$. It suggests that the binding of SWCNTs become harder as the concentration decreases.

As shown in supplementary Fig. 22, in a given SWCNTs/gel system, the SWCNT solution can be divided into bulk solution and boundary layer adjacent to gel surface. Due to the fast adsorption of SWCNTs, the concentration of SWCNTs in boundary layer $C_{n,m}^*$ is smaller than that in the bulk solution $C_{n,m}$, forming a concentration gradient, which drives transfer of SWCNTs from bulk solution to boundary layer.^{12, 13} The simplest expression of the relationship between the flux of a (n, m) species and the “driving force” of mass transfer is as follows¹⁰:

$$J = D_{n,m}(C_{n,m} - C_{n,m}^*)/\delta \quad (3)$$

where $D_{n,m}$ is the diffusion coefficient of the specific SWCNT/surfactant hybrid¹³, δ is the thickness of boundary layer determined by the flow rate of the solution and geometry of gel beads. At low concentration, due to the small difference in the concentrations of $C_{n,m}$ and $C_{n,m}^*$, small concentration gradient hinders the mass

transfer of SWCNTs to the gel surface and thus decreasing their adsorption onto gel surfaces due to low concentration in the reaction volume adjacent to the gel surface. SWCNTs tend to remain in the bulk solution and may flow through the gel, leading to more unadsorbed SWCNTs, as shown by the yellow line in Supplementary Fig. 22.

As the concentration in the bulk solution increases, the difference between $C_{n,m}$ and $C_{n,m}^*$ increases. The transfer of SWCNTs to the gel surface has been enhanced, providing more SWCNTs for the binding reactions, as shown by the red line in Supplementary Fig. 22, which reduce the proportion of unadsorbed SWCNTs. Then, as excessive SWCNTs were transferred to the gel surface in a short time, the adsorption kinetics turn to adsorption-control. The SWCNTs adjacent to the gel surface accumulate, which increase $C_{n,m}^*$ and decreases the concentration gradient in the boundary layer. At this stage, the resistance of SWCNT binding is mainly attributed to the adsorption process (red line in Supplementary Fig. 22). Therefore, further increasing the SWCNTs concentration, the adsorption amount of SWCNTs tend to saturate. Notably, this model is very simple and does not involve interactions between different (n, m) species and SWCNT transfer inside gel beads.

Supplementary Figure 22. Schematic diagram of mass transfer from the bulk SWCNT solution to gel surface.

Supplementary Information, Page 41, Line 659

Previous: Although the adsorption condition of semiconducting SWCNTs was not fulfilled at the concentration of 5 wt% SDS, irreversible adsorption was still observed, which may relate to the inhomogeneity of gel¹⁵⁻¹⁸.

Revised: Although the adsorption condition of semiconducting SWCNTs was not fulfilled at the concentration of 5 wt% SDS, irreversible adsorption was still observed, which may relate to the inhomogeneity of gel¹⁴⁻¹⁶. Similar to equation (1), irreversible binding process regardless of the environment condition can be described as follow:

where θ_{Ir} and $K_{n,m}^{Ir}$ represent the number of binding sites of irreversible adsorption in the reaction volume and the forward rate constant of irreversible adsorption, respectively. Compared with reversible, irreversible adsorption sites exhibits stronger affinity with SWCNTs. When SWCNTs flow through gel column, they preferentially adsorb at the irreversible adsorption sites. During the elution process, eq. 1 proceeds in the reverse direction, while eq. 4 is still a forward reaction. Thus, SWCNTs eluted from reversible sites at the top of the gel column flow down and may be captured by irreversible sites at the bottom of the gel column. Conceptually, a gel column is usually considered a pile composed of many plates, each of which composes of a very thin gel layer¹⁷. Adsorption of SWCNTs in the upper plates leads to the occupation of both reversible and irreversible adsorption sites. Then, the remained SWCNTs in solution flow down to the plates below. At low loading amounts, only plates at the top were filled. The solution that flows down contains much fewer SWCNTs, leaving the irreversible adsorption sites in the lower plates unoccupied. These unoccupied sites may trap SWCNTs that are eluted from the upper plates. As a result, nearly no SWCNTs were collected from the eluent. In contrast, when a large number of SWCNTs were loaded into the column, irreversible adsorption sites in the lower plates were filled. Thus, when the eluent flows through the column, the remained SWCNTs can be eluted from the lower plates. With an increase in loading amount, the proportion of irreversible adsorption decreases. This hypothesis coincides with the experimental results in Supplementary Fig. 10, where the amount of SWCNTs separated from a gel recycled for 5 separation runs is higher than that of the new gel, because some of the irreversible adsorption sites have been filled in the recycled gel.

Interestingly, the ratio of reversible and irreversible adsorption gradually reached a plateau with further increasing loading amount, as shown in Supplementary Fig. 24d. As mentioned above, with increasing the loading amount of SWCNTs, the irreversible adsorption sites are occupied, resulting in a gradual decrease in the proportion of irreversible adsorption. However, increasing the loading amount also increase the possibility of irreversible adsorption of SWCNTs. For example, the impurities in SWCNT solutions such as amorphous carbon and bundles also increased with loading amount and form more irreversible adsorption¹⁴. The two aspects eventually hold the proportion of irreversible adsorption at $\sim 15\%$. Notably, this mechanism is established under the condition of normal loading. Overloading is supposed to be more complicated, involving the competition between different (n , m) species and the saturation of gel. Additionally, the form of irreversible adsorption may also be attributed to the inherent properties of some SWCNTs, such as length, defects and

coatings¹⁴.

References:

Revised:

24. Flavel, B. S., Kappes, M. M., Krupke, R. & Hennrich, F. Separation of single-Walled carbon nanotubes by 1-dodecanol-mediated size exclusion Chromatography. *ACS Nano* 7, 3557–3564 (2013).
25. Flavel, B. S., Moore, K. E., Pfohl, M., Kappes, M. M. & Hennrich, F. Separation of single-walled carbon nanotubes with a gel permeation chromatography system. *ACS Nano* 8, 1817–1826 (2014).
28. Moore, K. E. *et al.* Separation of double-walled carbon nanotubes by size exclusion column chromatography. *ACS Nano* 8, 6756–6764 (2014).
29. Han, L. *et al.* Inner- and outer-wall sorting of double-walled carbon nanotubes. *Nature Nanotech.* **12**, 1176–1182 (2017).

Comments 2-2: The authors claim that their approach “provides a new method for the industrial separation of single-chirality SWCNTs”. However, it is not clear that the extensive sonication and ultracentrifugation required to produce the dispersions, which are still on milligram scale, will be industrially relevant. In addition, the chromatographic gel used for this work is extremely expensive, limiting the feasibility of industrial scale-up.

Reply: Thank you for your comments. Extensive sonication and ultracentrifugation processes are required for all solution-based SWCNT separation methods, because the preparation of individualized SWCNT dispersion is the premise of structure separation. Actually, sonication dispersion has been applied in the food industry (<https://doi.org/10.1016/B978-0-12-818717-3.00007-X>), chemical synthesis industry (<https://doi.org/10.1039/B503848K>) and wastewater treatment industry ([https://doi.org/10.1016/S1093-0191\(01\)00067-3](https://doi.org/10.1016/S1093-0191(01)00067-3)). Moreover, the ultrasonic dispersion of large volume of carbon nanotube solution has also been reported (<https://doi.org/10.1016/j.carbon.2019.01.026>). For the ultracentrifugation process, currently, commercial centrifugal devices can perform centrifugation of liter-scale solution (<https://www.eppendorf.com/gb-en/eShop-Products/Centrifugation/Ultracentrifuges/Ultracentrifuge-CP-NX-Series-p-5720110012>). Compared with the separation process, the centrifugation process is much shorter. Therefore, ultrasonic dispersion and centrifugation processes are not steps that limit the industrial separation of SWCNTs.

The gel medium is expensive, but it can be recycled over and over again during separation of SWCNTs. To illustrate the recyclability of gel, we added the experimental data on the impact of high-concentration SWCNT solutions on gel column's life time. Even after 20 cycles, the gel is still able to separate high-purity single-chirality SWCNTs without significantly degrading the performance, as shown in Supplementary Figure 10. In our present work, gel can be recycled at least 20 times,

which distinctly decreases the cost of gel. Actually, the techno-economic analysis in Supplementary Note 10.3 and Supplementary Fig. 27 suggests that gel cost is even lower than that of surfactants.

Thus, we believe the present method is scalable and has a great potential in industrial production of multiple single-chirality (n, m) species. Although only milligram scale of multiples single-chirality (n, m) SWCNTs has been achieved for the moment, it is increased by more than an order of magnitude compared with previously reported large-scale separation methods. The milligram-scale yield of single-chirality species is sufficient for the preparation of thousands of 4-inch wafers of dense SWCNT films (the calculation is presented in Supplementary Note 9). We added the corresponding data and discussion in the revised manuscript and supplementary Information.

Introduction, Page 6, Line 128

Previous: Our present dispersion and separation strategy provides a new method for the industrial separation of single-chirality SWCNTs over a wide diameter range.

Revised: Our present dispersion and separation strategy provides a new method for the industrial separation of single-chirality SWCNTs over a wide diameter range. Given that sonication and centrifugation processes have been industrially applied⁴¹⁻⁴³, we believe that the current method can further scale up the separation of single-chirality SWCNTs, and reduce environmental impact, energy consumption and separation costs by employing industrial homogenizer, and centrifugation, and gel column system.

Results, Page 13, Line 304

Previous: Notably, although the dispersion time increased with an increase in the initial concentration of dispersible SWCNTs, the increase in ultrasonic time did not significantly decrease the length of SWCNTs (Supplementary Fig. 9).

Revised: Notably, although the dispersion time increased with an increase in the initial concentration of dispersible SWCNTs, the increase in ultrasonic time did not significantly decrease the length of SWCNTs (Supplementary Fig. 9). Another important feature of the current method is that the gel column used could be recycled for at least 20 times without significant degradation in performance, which significantly reduced the separation cost of SWCNTs (Supplementary Fig.10).

Supplementary Information Page 22

Previous: None

Revised:

Supplementary Figure 10. The impact of high-concentration SWCNT solutions on gel column's life time and performance. 10 mL of SWCNT solutions with the initial concentration of 4 mg/mL were separated by columns filled with 40 mL of gel. The gel was recycled for 5, 10 and 20 separation runs. Optical absorption spectra of the separated (6, 4) SWCNTs are presented. The purities of the separated (6, 4) SWCNTs were not affected by the recycling of gel. However, the yield decreases slightly with increasing recycle times. After the column was recycled for 20 times, the mass of eluted SWCNT was decreased by approximate 30% compared with the new gel in a separation round. These results indicate that the gel column can be recycled at least 20 times..

Comments 2-3: The authors state that the yield of a particular SWCNT species (e.g., the (6,4) nanotubes) increases linearly with the concentration of the starting dispersion, which was loaded onto a 40 mL gel column. However, this must imply that the column does not get saturated by the SWCNTs, as one would expect a plateau to be reached within the plot in Figure 2c. While one can appreciate that higher nanotube concentrations in the starting dispersion are difficult to achieve, it would have been helpful if the authors first showed that, with a smaller column volume, a plateau in nanotube separation efficiency is reached.

Reply: Thank you for your suggestions. We added the additional experiments on the separation of single-chirality (6, 4) SWCNTs by loading SWCNT dispersion into a 10-mL gel column. As shown in the revised Figure 2c, the yield of separated (6, 4) gradually reaches a plateau as the concentration increases. We added the experimental data and the corresponding discussion in the revised manuscript and Supplementary Information.

Results, Page 11 Line 267

Previous: None

Revised: Notably, to improve the separation yield of single-chirality species by increasing the concentration of SWCNT dispersion, the size of gel column is also important. The capacity of small gel columns to carry SWCNTs is limited. With increasing SWCNT concentration, the gel column would be saturated by SWCNTs, limiting an increase in the separation yield of SWCNTs. To confirm this, we employed a 10-mL gel column for the separation of SWCNTs. Similarly, the volume of the loaded SWCNT solution was fixed at 10 mL. The optical absorption spectra of the separated (6, 4) SWCNTs under different SWCNT concentration are presented in Supplementary Fig. 6. The relationship between the yield of (6, 4) SWCNTs and the concentration of the loaded SWCNT solution is described in Fig. 2c. It can be clearly seen that with an increase in concentration, the separation yield of (6, 4) SWCNTs gradually reaches a plateau.

Results, Page 12, Figure 2

Previous:

Figure 2. Milligram-scale separation of single-chirality SWCNTs from a high-concentration individualized SWCNT solution. (a) Schematic diagram of the separation of single-chirality SWCNTs. (b) Optical absorption spectra of single-chirality (6, 4) SWCNTs from SWCNT solutions with different concentrations. The separated (6, 4) solutions were diluted to 20 mL for comparison. (c) The yield of single-chirality (6, 4) as a function of the concentration of individualized SWCNT solution. (d) Optical absorption spectra of separated (n, m) species on the milligram scale. (e) Photograph of the solution of single-chirality species separated from high-concentration SWCNT solution.

Revised:

Figure 2. Milligram-scale separation of single-chirality SWCNTs from a high-concentration individualized SWCNT solution. (a) Schematic diagram of the separation of single-chirality SWCNTs. (b) Optical absorption spectra of single-chirality (6, 4) SWCNTs from SWCNT solutions with different concentrations

using a 40-mL gel. The separated (6, 4) solutions were diluted to 20 mL for comparison. (c) The yield of single-chirality (6, 4) that separated by large (40 mL) and small (10 mL) gel columns as a function of the concentration of individualized SWCNT solution. (d) Optical absorption spectra of separated (n, m) species on the milligram scale. (e) Photograph of the solution of single-chirality species separated from high-concentration SWCNT solution.

Supplementary Information Page 17

Previous:

Supplementary Figure 3. Optical absorption spectra of the separated (6, 4) SWCNTs from HiPco-SWCNT mono-dispersions with concentrations ranging from 0.19 to 1.02 mg/ml. Notably, the separation of (6, 4) SWCNTs was performed by selective adsorption at 18 °C and selective desorption at 25 °C, as described in the method section. Specifically, 10 mL of the as-prepared SWCNT dispersion was loaded into a 40-mL gel column.

Revised:

Supplementary Figure 6. Relationship between the yield of (6, 4) and the concentration of SWCNT solutions. Optical absorption spectra of the separated (6, 4) SWCNTs from HiPco-SWCNT solutions with concentrations ranging from 0.19 to 1.02 mg/ml using a) 40-mL gel columns and b) 10-mL gel columns. Notably, the separation of (6, 4) SWCNTs was performed by selective adsorption at 18 °C and selective desorption at 25 °C, as described in the method section. Specifically, 10 mL of the as-prepared SWCNT dispersion was loaded into a 40-mL gel column.

Comments 2-4: On page 14, the authors state that with a starting concentration of G-SWCNTs of 4 mg/mL, they achieved good separation of single chirality species, but when the initial concentration was 1 mg/mL, single-chirality samples were not achieved. It is not clear why decreasing the concentration would decrease separation efficiency. One would expect the opposite. It would be helpful if the authors would propose an explanation for this result.

Reply 2-4: Thank you for your questions and suggestions. We apologize for not specifying the separation results of G-SWCNTs of 1 mg/mL. We separated several single-chirality (n, m) species from the 1-mg/mL G-SWCNTs, regardless of the very low yield. Their purities were similar to those separated from 4-mg/mL. The purity analysis on the (n, m) species achieved from 1-mg/mL G-SWCNTs has been added in Supplementary Figure S14. However, some (n, m) species such as (6, 5), (7, 5), (7, 6) and (8, 4) were not achieved from 1-mg/mL G-SWCNTs because none of SWCNTs were detected in the eluents. These species are consumed as irreversible adsorption and unadsorbed parts, as evidenced in Figure 5 and Supplementary Figs 21-24 due to low content. We added the corresponding discussion in the revised manuscript and Supplementary Notes 8 and 9.

Results, Page 16 Line 379

Previous: Notably, in the case of the initial concentration of 1 mg/mL, single-chirality (6, 5), (7, 5), (7, 6) and (8, 4) SWCNTs were not achieved, indicating that the preparation of a high-concentration and high-dispersity solution is a prerequisite for the separation of single-chirality SWCNTs from G-SWCNTs.

Revised: Notably, in the case of the initial concentration of 1 mg/mL, although the purities of the separated (n , m) species were higher than 80% or even 90% (Supplementary Fig. 13), the yield was distinctly low, and even single-chirality (6, 5), (7, 5), (7, 6) and (8, 4) SWCNTs were not achieved, most likely due to their low concentration and low absolute content, and a large portion of them was lost as irreversible or unadsorbed species. A more detailed discussion was presented later. These results indicate that the preparation of a high-concentration and high-dispersity solution is a prerequisite for the separation of single-chirality SWCNTs from G-SWCNTs.

Discussion, Page 21, Line 484

Previous: The results show that a lower concentration of the loaded SWCNT solution with a fixed loading amount led to a higher proportion of unadsorbed SWCNTs (Supplementary Fig. 15). With an increase in the SWCNT concentration, the equilibrium between adsorption and desorption shifted to the direction of adsorption^{26,27,41}, resulting in an increase in the reversibly adsorbed SWCNTs and a decrease in the proportion of unadsorbed and irreversibly adsorbed SWCNTs, which inevitably increased the separation yield of SWCNTs.

Revised: The results show that a lower concentration of the loaded SWCNT solution with a fixed loading amount led to a higher proportion of unadsorbed SWCNTs (Supplementary Fig. 21). Similar to the separation of biomolecules in a chromatographic system^{32,51}, the concentration of SWCNT solution was a key parameter, which determined the mass transfer of SWCNTs from the bulk solution to the boundary layer adjacent to gel surface and thus the SWCNT concentration in the reaction volume (Supplementary Note 8 and Supplementary Fig. 22). Based on the adsorption kinetics of molecules^{30,31,52}, a high SWCNT concentration in the reaction volume would promote the adsorption of SWCNT onto gel surface. At low concentration, the mass transfer of SWCNTs to the gel surface was hindered due to small SWCNT concentration gradient between bulk solution and boundary layer, and thus decreasing their adsorption onto gel surfaces. Most of SWCNTs remained in the bulk solution and flew through the gel column as unabsorbed species. In contrast, with increasing SWCNT concentration, the transfer of SWCNTs to the gel surface was enhanced, providing more SWCNTs for the binding reactions and thus reducing the proportion of unadsorbed SWCNTs. As the SWCNT concentration further increased, excessive SWCNT could be transferred to the gel surface and accumulated in the reaction volume within a short time, the adsorption kinetics turned to adsorption control. Therefore, further increasing the SWCNTs concentration, the adsorption amount of SWCNTs tended to saturate and the curve slopes in Fig. 5c decreased. A detailed discussion is presented in the Supplementary Note 8.

Discussion, Page 23, Line 530

Previous: With an increase in the loading amount, the proportion of irreversibly adsorbed SWCNTs decreased rapidly from 70% to ~20%, which originated from the quasirandom irreversible adsorption in gel chromatography⁴². A detailed discussion is presented in Supplementary Note 5.2.

Revised: With an increase in the loading amount, the proportion of irreversibly adsorbed SWCNTs decreased rapidly from 70% to ~15%, which originated from the quasirandom irreversible adsorption in gel chromatography⁵³. Compared with reversible adsorption, irreversible adsorption sites exhibited stronger affinity with SWCNTs. When SWCNTs flew through gel column, they preferentially adsorbed at the irreversible adsorption sites. Conceptually, a gel column can be considered as being composed of many thin layers of gel⁵⁴. When a small amount of SWCNT solution were loaded into a gel column, SWCNTs could be adsorbed by both reversible and irreversible adsorption sites in the upper layers of the gel column, leaving the irreversible adsorption sites in the lower layers unoccupied. During elution, the eluted SWCNTs from reversible adsorption sites may be captured again by the irreversible adsorption sites in the lower layers (Supplementary Fig. 25), leading to a significant decrease in the SWCNT concentration in the eluted solution and even no SWCNT were collected. With an increase the loading amount of SWCNTs, more and more SWCNTs could be collected in the eluted solution because of the occupation of more irreversible adsorption sites, resulting in a dramatic decrease in the proportion of irreversible adsorption and thus a rapid increase in the separation yield of SWCNTs (Fig. 5d and Supplementary Fig. 25). Notably, as the loading amount continuously increased, more reversibly adsorbed SWCNTs were eluted while the probability of irreversible adsorption of SWCNTs also increased, leading to a constant proportion of irreversible adsorption. In this way, the separation yield began to increase linearly with an increase in the loading amount (Fig. 5d). A detailed discussion was presented in Supplementary Note 9.

Supplementary Information, Page 25

Previous: None

Supplementary Figure 13. Purity evaluation of the (n, m) species separated from G-SWCNTs with the initial concentration of 1 mg/mL.

Supplementary Information, Page 35 Line 547

Previous: As shown in Supplementary Figure 15d, the proportion of the irreversible adsorption decreases slightly with decreasing SWCNT concentration under such a fixed loading amount.

Revised: As shown in **Supplementary Fig. 21d**, the proportion of the irreversible adsorption decreases slightly with decreasing SWCNT concentration under such a fixed loading amount, **while the proportion of unadsorbed SWCNTs increased rapidly.**

Similar to biomolecules¹⁰, the adsorption process of SWCNTs to gel can be described as follow:

where $SWCNT_{n,m}^{free}$ and $SWCNT_{n,m}^{ads}$ represent the number of free (n, m) species and the number of adsorbed (n, m) species in a reaction volume of gel, respectively. θ is the total number of adsorption sites in this area. $K_{n,m}^f$ represent the forward rate constants, which is determined by the interaction between a (n, m) SWCNT and an unoccupied binding site¹¹. At low concentrations, the number of each (n, m) species in the reaction volume is low, leading to a low binding rate. The change of concentration of (n, m) species with time reveal the binding rate¹¹:

$$-\frac{dC_{n,m}(t)}{dt} = K_{n,m}^f C_{n,m}(t) \left[\frac{\theta}{V} - \Sigma (C_{n,m}(t_0) - C_{n,m}(t)) \right] \quad (2)$$

At a given time, the rate of change of $C_{n,m}(t)$ nonlinearly decreases with $C_{n,m}(t)$. It suggests that the binding of SWCNTs become harder as the concentration decreases.

As shown in supplementary Fig. 22, in a given SWCNTs/gel system, the SWCNT solution can be divided into bulk solution and boundary layer adjacent to gel surface. Due to the fast adsorption of SWCNTs, the concentration of SWCNTs in boundary layer $C_{n,m}^*$ is smaller than that in the bulk solution $C_{n,m}$, forming a concentration gradient, which drives transfer of SWCNTs from bulk solution to boundary layer.^{12, 13} The simplest expression of the relationship between the flux of a (n, m) species and the “driving force” of mass transfer is as follows¹⁰:

$$J = D_{n,m}(C_{n,m} - C_{n,m}^*)/\delta \quad (3)$$

where $D_{n,m}$ is the diffusion coefficient of the specific SWCNT/surfactant hybrid¹³, δ is the thickness of boundary layer determined by the flow rate of the solution and geometry of gel beads. At low concentration, due to the small difference in the concentrations of $C_{n,m}$ and $C_{n,m}^*$, small concentration gradient hinders the mass transfer of SWCNTs to the gel surface and thus decreasing their adsorption onto gel surfaces due to low concentration in the reaction volume adjacent to the gel surface. SWCNTs tend to remain in the bulk solution and may flow through the gel, leading to more unadsorbed SWCNTs, as shown by the yellow line in Supplementary Fig. 22.

As the concentration in the bulk solution increases, the difference between $C_{n,m}$ and $C_{n,m}^*$ increases. The transfer of SWCNTs to the gel surface has been enhanced, providing more SWCNTs for the binding reactions, as shown by the red line in Supplementary Fig. 22, which reduce the proportion of unadsorbed SWCNTs. Then, as excessive SWCNTs were transferred to the gel surface in a short time, the adsorption kinetics turn to adsorption-control. The SWCNTs adjacent to the gel surface accumulate, which increase $C_{n,m}^*$ and decreases the concentration gradient in the boundary layer. At this stage, the resistance of SWCNT binding is mainly attributed to the adsorption process (red line in Supplementary Fig. 22). Therefore, further increasing the SWCNTs concentration, the adsorption amount of SWCNTs tend to saturate. Notably, this model is very simple and does not involve interactions between different (n, m) species and SWCNT transfer inside gel beads.

Supplementary Figure 22. Schematic diagram of mass transfer from the bulk SWCNT solution to gel surface.

Supplementary Information, Page 41, Line 659

Previous: Although the adsorption condition of semiconducting SWCNTs was not fulfilled at the concentration of 5 wt% SDS, irreversible adsorption was still observed, which may relate to the inhomogeneity of gel¹⁵⁻¹⁸.

Revised: Although the adsorption condition of semiconducting SWCNTs was not fulfilled at the concentration of 5 wt% SDS, irreversible adsorption was still observed, which may relate to the inhomogeneity of gel¹⁴⁻¹⁶. **Similar to equation (1), irreversible binding process regardless of the environment condition can be described as follow:**

where θ_{Ir} and $K_{n,m}^{Ir}$ represent the number of binding sites of irreversible adsorption in the reaction volume and the forward rate constant of irreversible adsorption, respectively. Compared with reversible, irreversible adsorption sites exhibits stronger affinity with SWCNTs. When SWCNTs flow through gel column, they preferentially adsorb at the irreversible adsorption sites. During the elution process, eq. 1 proceeds in the reverse direction, while eq. 4 is still a forward reaction. Thus, SWCNTs eluted from reversible sites at the top of the gel column flow down and may be captured by irreversible sites at the bottom of the gel column. Conceptually, a gel column is usually considered a pile composed of many plates, each of which composes of a very

thin gel layer¹⁷. Adsorption of SWCNTs in the upper plates leads to the occupation of both reversible and irreversible adsorption sites. Then, the remained SWCNTs in solution flow down to the plates below. At low loading amounts, only plates at the top were filled. The solution that flows down contains much fewer SWCNTs, leaving the irreversible adsorption sites in the lower plates unoccupied. These unoccupied sites may trap SWCNTs that are eluted from the upper plates. As a result, nearly no SWCNTs were collected from the eluent. In contrast, when a large number of SWCNTs were loaded into the column, irreversible adsorption sites in the lower plates were filled. Thus, when the eluent flows through the column, the remained SWCNTs can be eluted from the lower plates. With an increase in loading amount, the proportion of irreversible adsorption decreases. This hypothesis coincides with the experimental results in Supplementary Fig. 10, where the amount of SWCNTs separated from a gel recycled for 5 separation runs is higher than that of the new gel, because some of the irreversible adsorption sites have been filled in the recycled gel.

Interestingly, the ratio of reversible and irreversible adsorption gradually reached a plateau with further increasing loading amount, as shown in Supplementary Fig. 24d. As mentioned above, with increasing the loading amount of SWCNTs, the irreversible adsorption sites are occupied, resulting in a gradual decrease in the proportion of irreversible adsorption. However, increasing the loading amount also increase the possibility of irreversible adsorption of SWCNTs. For example, the impurities in SWCNT solutions such as amorphous carbon and bundles also increased with loading amount and form more irreversible adsorption¹⁴. The two aspects eventually hold the proportion of irreversible adsorption at $\sim 15\%$. Notably, this mechanism is established under the condition of normal loading. Overloading is supposed to be more complicated, involving the competition between different (n, m) species and the saturation of gel. Additionally, the form of irreversible adsorption may also be attributed to the inherent properties of some SWCNTs, such as length, defects and coatings¹⁴.

Comments 2-5: Overall, the authors show that as the concentration and volume of the starting nanotube dispersion increases, the size of the separation column must also increase. Additionally, due to some small amount of irreversible adsorption of nanotubes on the stationary phase, low initial nanotube concentrations will not lead to as high a recovery yield as high initial nanotube concentrations. Ultimately, many of the significant results are not surprising, and thus the novelty of the work is questionable. It does not seem to meet the standards required for this journal.

Reply: Thank you for your comments. Industrial production of single-chirality SWCNTs has been a long-term pursuit in the field of carbon nanomaterials. For this, various methods including ATPE, gel chromatography and polymer wrapping, have been developed. However, to our knowledge, none of these methods achieved milligram-scale separation of multiple single-chirality (n, m) species from high-quality raw materials, indicating that it is extremely difficult to realize such a goal. Let alone large-scale separation of single-chirality species from low-cost raw

materials such as a mixture of graphene and SWCNTs without structure enrichment.

Although increasing the separation yield is supposed to be possible by simply increasing the capacity of the separation system, the time and energy consumption will also grow, leading to a low efficiency and high fabrication cost, as we discussed in Supplementary Notes 8-10.

However, in our present work, the use of high-concentration SWCNT solution for the separation of single-chirality SWCNTs not only increases the separation yield of single-chirality species, but also significantly reduces the environmental impact, energy consumption and separation cost. This is very important for industrialization of single-chirality SWCNTs, which is helpful for promoting the application of SWCNTs in high-end electronics and optoelectronic fields. At the same time, we also proposed a scientific explanation on the dispersion mechanism of high-concentration SWCNT solution and thus the mechanism on the increased separation yield of single-chirality SWCNTs. Clearly, our present work is a great advancement in the SWCNT field. We believe that our present work will stimulate great interest to researchers working in the diverse and multi-disciplinary areas including materials, chemistry, electronics, optics, optoelectronic integration and bioimaging, etc. and is suitable for publication in Nature Communications

Discussion, Page 24, Line 576

Previous: Life cycle assessment (LCA) is very important for the industrial production of a new material.

Revised: Life cycle assessment (LCA) is very important for the industrial production of a new material. **Given that large-scale production data of single-chirality SWCNTs are lacking, LCA is essential for predicting the impacts of scaling up and assessing the technological readiness of the emerging methods^{56,57}. Moreover, the LCA may provide a benchmark as well as a reference to evaluate the advantages and limitations of a separation method.**

Discussion, Page 26 Line 620

Previous: The separation costs of such an amount of single-chirality species from high-concentration HiPco- and G-SWCNTs are \$0.8 and \$3.8, respectively, which are comparable to that of a 4-inch silicon wafer. We can expect that the separation cost of single-chirality species could be further reduced by scaling up (as discussed in **Supplementary Note 7.3**)⁵¹. These results demonstrate a high technology readiness level of our present technique, exhibiting a great promising of industrial preparation and commercial application of single-chirality SWCNTs.

Revised: The separation costs of such an amount of single-chirality species from high-concentration HiPco- and G-SWCNTs are \$0.8 and \$3.8, respectively, which are comparable to that of a 4-inch silicon wafer. **Overall, the separation efficiency using**

high-concentration SWCNT solution is distinctly higher compared with conventional 1-mg/mL samples, despite that the size of gel column was increased, which reduced time, energy and material consumption and thus the separation cost. Given that sonication and ultracentrifugation process has been applied the chemical synthesis industry^{41,43} and also large volumes of SWCNT dispersion⁴², we can expect that the separation efficiency of single-chirality species could be further increased while the separation cost, environmental impact and energy consumption were further reduced by scaling up with industrial dispersion, centrifugation and gel column system (as discussed in Supplementary Note 10.3)⁵⁸. These results demonstrate a high technology readiness level of our present technique, exhibiting a great promising of industrial preparation and commercial application of single-chirality SWCNTs.

Reviewer #3:

This communication describes new achievements in gel-based purification of single walled carbon nanotubes (SWCNTs). The achievement of low-cost procurement of highly-purified “single-chirality” SWCNTs is a longstanding challenge of the carbon nanoscience community. Significant advancements in this area are highly likely to have a substantial immediate positive impact within a variety of areas that 1) study fundamental SWCNT properties; 2) presently employ SWCNT within devices/schemes but stand to benefit from the availability of higher purity materials; and 3) are limited in scale due to the high cost of purified SWCNT. Because of this potential impact coupled with the many scientific resources that have been invested in solving the problem of producing low-cost high purity SWCNT, the scope and scale of this work is appropriate for consideration in Nature Communications.

As an academic researcher who is choosing to sign my review letter, I would like to briefly highlight both my expertise and my naivete in the areas of this manuscript. Specifically, I have approximately 11 years of experience in SWCNT purification, nearly all of which has utilized gel-based methods. More recently, my research team at the University of Colorado at Colorado Springs has focused on understanding mechanistically the unique relationship between hydrogel surface, SWCNT structure/chirality, and surfactant nature/density that affords gel-based single chirality purification. Several of our recent publications have focused on the role of the gel medium in this dynamic system. Given this, I am intimately familiar with the challenges of creating high concentration SWNT suspensions and the general use of gel-based techniques to afford isolation of single chirality species (although my research team does not employ the temperature gradient method used in this specific work). In contrast, I am markedly less familiar with the Life Cycle Assessment (LCA) performed in this work, although I appreciate the value of such given the direct connection made by the authors between novel methodology and industrial scalability. All this is to say that I accepted the task of reviewing this work with both enthusiasm to learn about new findings in this area and with confidence that I am able to provide comments that are both informed and without conflicts of interest either in favor of or against acceptance for publication.

The manuscript reports data and analysis that appear to fit within five distinct themes (in order of appearance in the manuscript): 1) the novel method of iterative sonication/centrifugation to achieve high concentration suspensions of individualized SWCNT; 2) a demonstration of how such suspensions achieve scaled-up isolation of high-purity materials from a relatively high-cost source of unpurified SWCNT; 3) a demonstration of how such suspensions achieve scaled-up isolation of high-purity materials from a relatively low-cost source of unpurified SWCNT; 4) efforts to probe mechanistic effects of SWCNT concentration on gel-based purification; and 5) a Life Cycle Assessment estimating the costs of using the reported novel method to produce purified SWNT at scale. In general, the manuscript is well written and well connected with the data presented and the experimental methods are clear. Those method employed are of appropriate scope and scale for this project and align well with

standards in the field. The pure quantity of data collectively presented within the manuscript and the Supplementary Information (SI) demonstrates the investment that this team has made in this project, certainly a commendable achievement. It is my expectation that most SWNT purification researchers, upon reading this work, will be inspired to quickly test the effectiveness of relatively high-concentration SWCNT suspensions within their SWCNT purification schemes/workflows.

That said, there are some shortcomings of this manuscript that I would like to see addressed by the authors before this work is considered for publication. My concerns primarily fall into two categories: 1) a lack of control experiments to better demonstrate the novel effects of the iterative sonication/centrifugation method; and 2) a lack of a clear mechanism for concentration effects on both purification efficiency and efficacy (% purity of single-chirality species). Other relatively minor concerns are also listed below, as organized by the 5 themes of the manuscript listed above.

Reply: Thank you very much for your positive comments, valuable recommendation and constructive suggestions. According to your suggestions, we added extra experiments and discussion in the revised manuscript and supplementary Information.

Theme 1 (pages 6-8): The novel method of iterative sonication/centrifugation to achieve high concentration suspensions of individualized SWCNT

Comments 3-1: This section of the manuscript, specifically the method of iterative sonication/centrifugation, represents the foundation of the work and affords/justifies the remainder of the study. The authors report that “the dispersion time was set as a proportional function of the initial SWCNT concentration.” I would expect that most of the energy dissipation from a tip-horn sonicator is transferred thermally (and not chemically), so to me it is unclear why scaling sonication duration with SWCNT concentration is a logical choice. Further, this choice convolutes initial SWNT concentration and duration of the first (of two) sonication periods. For example, if suspensions initially at 1, 4 and 8 mg/mL are subjected to identical sonication/centrifugation duration, how does the final SWNT concentration (after the second centrifugation) differ from a procedure that is scaled to initial SWCNT concentration? It would be useful for the authors to conduct and report this experiment, as it would provide insight into the necessity of prolonged sonication for high concentration samples as well as the linear scaling choice of sonication time vs. SWCNT concentration.

Reply: Thank you for your good suggestions. Given that with increasing the initial concentration of SWCNTs, more bundles and impurities are introduced into the solution and thus the viscosity increases significantly. As we mentioned in the manuscript, increasing viscosity and concentration of SWCNTs negatively affect the ultrasound propagation, thus degrading the dispersion efficiency. Sufficiently dispersing higher concentration SWCNT solution should require a longer dispersion time. To verify the necessity of dispersion time with increasing concentration, we added additional experiments. As you suggested, suspensions initially at 1, 4 and 8

mg/mL are subjected to identical sonication/centrifugation duration.

Results, Page 6 Line 142

Previous: The dispersion time was set as a proportional function of the initial SWCNT concentration (see the experimental details in **Supplementary Note 1**),

Revised: The dispersion time was set as a proportional function of the initial SWCNT concentration (see the experimental details in **Supplementary Note 1**), so as to sufficiently disperse SWCNTs to increase the concentration of the resulting SWCNT dispersion (Supplementary Note 2 and Supplementary Figs. 1 and 2). To optimize dispersion time of the SWCNT solutions with different initial concentrations, we studied the variation of their viscosity and concentration with time during dispersion process (Supplementary Fig. 2). At the beginning, the viscosity of SWCNT solution increased rapidly and then reached the maximum values with increasing sonication duration. With continuously increasing sonication time, the viscosity of SWCNT solutions decreased dramatically and finally reached an approximate constant value. We can imagine that at the beginning, SWCNT powders were crushed into particles composed of large bundles and impurities under ultrasonic dispersion. Due to the low dispersity and increased number of suspended particles, the friction between these particles increased the viscosity of solutions. As the sonication time increased, SWCNTs were continuously stripped from the bundles. The sidewalls of SWCNTs exposed to the solution were readily coated with surfactant molecules, decreasing the friction due to high lubrication effect of surfactants⁴⁴. Meanwhile, denser surfactant coatings around SWCNTs provided a repulsive region and thus excellent fluidity⁴⁵. Therefore, when the viscosity of a SWCNT solution dropped to a stable value after ultrasonic dispersion, it should be sufficiently dispersed, which was evidenced by its concentration evolution over sonication time (Supplementary Note 2 and Supplementary Fig. 2).

Supplementary Information, Page 7

Previous: None

Revised:

Supplementary Note 2: Effect of sonication time on the preparation of high-concentration SWCNT solution.

During dispersion, scaling sonication duration with SWCNT concentration is very important. To confirm this, we have done the following experiments. 20-mL dispersions with initial concentrations of 1, 4 and 8 mg/mL were prepared by sonicating for identical 2 hours and subsequently centrifugating for 1 hour at 210,000 g. For comparison, a control group of samples with initial concentrations of 4 and 8 mg/mL were sonicated for 6 and 11 hours, respectively. Redispersion was not

performed for these two groups. After the identical sonication and ultracentrifugation, the concentrations of the as-prepared dispersions increase with an increase in initial SWCNT concentrations, as shown in supplementary **Figure 1**. More importantly, for SWCNT solutions with initial concentrations of 4 and 8 mg /mL, the concentrations of the resulted dispersion solution increase significantly with the extension of ultrasonic time to 4 and 8 h, respectively. Therefore, scaling sonication duration with SWCNT concentration is very important to fully disperse them to increase the concentration of the individualized SWCNT solution and thus improve their separation efficiency.

Supplementary Figure 1. Optical absorption spectra of dispersions with different initial concentrations subjected to different sonication times.

To understand suspension evolution and dynamics during tip horn sonication, we explored the viscosity variation of SWCNT dispersions with initial concentrations of 1 and 4 mg/mL over time. As shown in supplementary Fig. 2a, the viscosity of two dispersions exhibits similar trend over time. At the beginning, the viscosity increases rapidly and then reach the maximum values with increasing sonication duration. With continuously increasing sonication time, the viscosity of SWCNT solutions decreases dramatically and finally reaches an approximate constant value.

Based on the viscosity variation of SWCNTs over sonication time, we proposed the dispersion process of SWCNTs over time. At the initial stage, SWCNT powders floated on the surface of the solution or form precipitate at the bottom. The viscosity is merely attributed to the water and surfactants. Subsequently, SWCNT powders were crushed into particles composed of large bundles and impurities. Due to the low dispersity and increased number of suspended particles, the friction between these particles increases the viscosity of solutions. As the sonication time increased, SWCNTs were continuously stripped from the bundles. The sidewalls of SWCNTs exposed to the solution were readily coated with surfactant molecules, decreasing the friction due to high lubrication effect of surfactants³. Meanwhile, denser surfactant coatings around SWCNTs provided a repulsive region and thus excellent fluidity⁴.

During the dispersion process, the viscosity of the SWCNT solution with an initial concentration of 4 mg/mL was much higher than that of the SWCNT solution with a concentration of 1 mg/mL, especially for the maximum viscosity. Moreover,

the viscosity growth stage lasted for 40 minutes, which is also much longer. These results indicate that viscosity has a significant impact on the dispersion of SWCNTs for higher initial concentration. As shown in Supplementary Figs. 2b-d, the concentration of the remained SWCNTs in the supernatant decrease rapidly with a decrease in sonication time, indicating that most of the SWCNTs were still present in bundles and precipitated during centrifugation. These results confirmed that scaling sonication duration with SWCNT concentration is very important to fully disperse SWCNTs. As shown in Figure 1b in the main manuscript, the resulting viscosity differed slightly over the wide concentration range of 1-8 mg/mL after first centrifugation. Besides, we observed little variation in SWCNT concentration and viscosity after the second centrifugation, which evidenced the relatively high efficiency dispersion during the second sonication due to the removal of large bundles and impurities via the first centrifugation.

Supplementary Figure 2. Time-dependent variation of solution viscosity and corresponding SWCNT concentration following centrifugation. a) Relationship between sonication time and viscosity of SWCNT solutions with the initial concentration of 1 mg/mL and 4 mg/mL. b)-c) Optical adsorption spectra of SWCNT solutions after first b) and second c) ultracentrifugation. The durations of the first sonication period were shown in the legend. The time of the second sonication was fixed at 15 minutes for all samples. d) Variation of the SWCNT concentration of solutions after centrifugation with respect to duration of the first sonication period.

Comments 3-2: The authors state that prolonged sonication does not achieve high concentrations of individualized SWCNT due to a limitation in viscosity. It would be interesting (and certainly related to the foundational achievement of this work) to know how SWCNT suspension viscosity increases over time for the two different SWCNT source materials used in this study. This question would be addressed in a

singular plot showing sonication time on the horizontal axis and containing three vertical axes of 1) pre centrifugation viscosity; 2) total SWCNT dispersed following first centrifugation; and 3) total SWCNT dispersed following second sonication/centrifugation. This data at, say 1 and 4 mg/mL loading of raw SWNT, would clearly demonstrate the role of initial sonication time in eventual procurement of high concentration suspensions. Further, it would be a compelling addition to this work if the temporal increase in viscosity during sonication could be correlated with a model of system viscosity driven by a combination of rods (individualized SWNT), large diameter rods (bundled SWNT), and plates (graphitic impurities). The findings presented in this work clearly demonstrate that an improved understanding of suspension evolution and dynamics during tip horn sonication could greatly benefit the efficacy and efficiency of SWCNT purification, so any additional physical/mechanistic insights would be greatly appreciated by myself and would likely be appreciated by others in the SWCNT purification community.

Reply 3-2: Thank you for your constructive suggestions. According to your suggestions, we added the corresponding experiments and discussion in the revised manuscript and Supplementary Information.

The corresponding revision is the same to Comment 3-1

Comments 3-3: The authors claim that their two-step sonication/centrifugation method creates “monodisperse” SWCNT suspensions (I will use the word “individualized” here as “monodisperse” elicits, at least in my mind, a narrow colloidal size distribution that is necessarily larger than the smallest fundamental unit of the colloid). In the methods section of the SI, they further state that while centrifuging for 1h in their two-step method leaves behind bundled SWCNT that can be individualized by further sonication, centrifuging for 2h removes all bundles from the suspension. It is expected that any SWCNT suspension that has been subjected to sonication and centrifugation in SDS contains some fraction of both individualized and bundled (of varying bundle sizes) SWCNT. While the authors have discovered a novel way to manipulate these fractions, it should be acknowledged somewhere in the manuscript that a truly “bundle free” sample is not likely attained, but rather, SWCNT bundles are minimized (not eliminated) by shortening the first centrifugation step and subjecting to further sonication/centrifugation.

Reply: Thank you for your constructive suggestions. We have revised the expression “monodisperse” into “individualized” throughout the manuscript and Supplementary Information. We also gave a statement that SWCNT bundles were minimized other than eliminated after dispersion and centrifugation in the revised manuscript and Supplementary Information.

Results, Page 8 Line 186

Previous: Before separation, the remaining impurities, such as carbon impurities, and the metallic particles generated from the ultrasonic tips were removed by a second centrifugation at $210000 \times g$ for 15 min.

Revised: Before separation, the remaining impurities, such as carbon impurities, and the metallic particles generated from the ultrasonic tips were removed by a second centrifugation at $210000 \times g$ for 15 min. **It should be noted that although high dispersity of SWCNT solution was verified by the separation of high-purity single-chirality SWCNTs (Supplementary Note 3 and Supplementary Fig. 3), it was not possible for the two-step dispersion and centrifugation method to achieve bundle-free solution, but to minimize bundles in solution.**

Comments 3-4: The authors claim that a 1h centrifugation following their initial sonication leaves some bundles in the suspension that can be individualized by further sonication. If this mechanism is accurate, the authors should be able to visually identify SWCNT bundles using electron microscopy following, for example, 9h of sonication of a 1 mg/mL sample and 24h of sonication of a 4 mg/mL sample, and subsequent 1h centrifugation of each. Such evidence would be a compelling argument in favor of their proposed mechanism.

Reply: Thank you for your constructive suggestions. Given that the purity of single-chirality SWCNTs is very sensitive to the dispersity of SWCNT solutions, we verified the dispersity through the separation of single-chirality (6, 4) SWCNTs, as shown in the Supplementary Note 3. As you suggested, we further characterized SWCNT bundles using atomic force microscopy (AFM). We added the corresponding experimental results and discussion in the revised manuscript and Supplementary Information.

Results, Page 8, Line 183

Previous: Thus, the remained small bundles were dispersed into individual SWCNTs, which was demonstrated by the separation of single-chirality (6, 4) species

Revised: Thus, the remained small bundles were dispersed into individual SWCNTs, which was demonstrated by the separation of single-chirality (6, 4) species **and their atomic force microscope images (Supplementary Note 3 and Supplementary Figure 3).**

Supplementary Information, Page 10 Line 215

Previous: As shown in Supplementary Figure 3a, the separation of (6, 4) single-chirality species cannot be achieved from the SWCNT solution with an initial concentration of 4 mg/mL dispersed by traditional method even if the dispersion time was increased from 16 hours to 32 hours, indicating that the as-prepared SWCNT solution was not highly dispersed. To obtain the individualized SWCNT solution, the

alternative method is to increase the centrifugation time. For HiPco-SWCNT solution with an initial concentration of 4 mg/mL, the centrifugation time increased from 1 to 2 hours after dispersing for 32 hours. Although the separation of (6, 4) single-chirality SWCNTs was achieved, the SWCNT concentration of the as-prepared solution decreased significantly (as shown in Supplementary Figure 1b). Compared with the individualized SWCNT solution with an initial concentration of 1 mg/mL, the SWCNT concentration did not increase clearly, resulting in the low yield of (6, 4) SWCNTs (Supplementary Figure 1c).

Revised: As shown in **Supplementary Fig. 3a**, the separation of (6, 4) single-chirality species cannot be achieved from the SWCNT solution with an initial concentration of 4 mg/mL dispersed by traditional method even if the dispersion time was increased from 16 hours to 32 hours followed by centrifugation for 1 hour, indicating that the as-prepared SWCNT solution was not highly dispersed. The reason why single-chirality (6, 4) SWCNTs cannot be obtained is that the dispersion solution is not sufficiently dispersed. (6, 4) SWCNTs are still mixed with other (n, m) SWCNTs in the form of bundles. After loading into gel column, (6, 4) nanotubes that bundled with other (n, m) species were adsorbed onto the gel, leading to a low purity of separated (6, 4). Therefore, the separation of (6,4) SWCNTs can be used to probe the SWCNT dispersity. Given that the structure distribution of the separated SWCNTs decrease with increasing sonication time, as shown in Supplementary Fig. 3a, we assumed that the size of bundles remained in the solutions decreased. And, it is difficult to remove these small bundles by 1-h ultracentrifugation. To obtain the individualized SWCNT solution, the alternative method is to increase the centrifugation time. For HiPco-SWCNT solution with an initial concentration of 4 mg/mL, the centrifugation time increased from 1 to 2 hours after dispersing for 32 hours. After a 2-hour ultracentrifugation, the separation purity distinctly increased accompanied by a decrease in the SWCNT concentration (as shown in **Supplementary Fig. 3b**). Compared with the individualized SWCNT solution with an initial concentration of 1 mg/mL, the SWCNT concentration did not increase clearly, resulting in the low yield of (6, 4) SWCNTs (**Supplementary Fig. 3c**). To further confirm whether bundles were present in the solutions with initial concentration of 4 mg/mL prepared by the traditional method, the SWCNTs with and without a redispersion were deposited on a Si/SiO₂ and characterized by AFM. More bundles were observed in the AFM image of SWCNTs prepared by sonication for 24 h followed by ultracentrifugation for 1 h (Supplementary Fig. 3d). In contrast, Supplementary Fig. 3e shows the well dispersed SWCNTs which were prepared by the redispersion method, in which the SWCNT samples were redispersed for another 1 h and centrifugated for 15 min. These results fully indicate that it is not possible to prepare high-concentration and high-dispersity SWCNT solution using the traditional preparation method.

Supplementary Figure 3. (a) Optical adsorption spectra of the SWCNTs separated from high-concentration SWCNT solution prepared by conventional methods; The eluted SWCNT solutions are diluted to 20 mL. (b) Optical adsorption spectra of the as-prepared SWCNT solutions with the initial concentration of 4 mg/ml using the re-dispersion method and the traditional method followed by centrifugation of 2 hours. (c) Optical absorption spectra of the separated (6, 4) SWCNTs from the **individualized** SWCNT solutions prepared by re-dispersion method and traditional method with 2-h centrifugation. The eluted SWCNT solutions are diluted to 20 mL. (d) **AFM images of the SWCNTs after the first period of sonication (24 h) and ultracentrifugation (1 h) and (e) after the second period of sonication (1 h) and ultracentrifugation (15 min).** The height variations along the white dash line in d) and e) were exhibited below the AFM images. Multiple small bundles with the diameter larger than 1.5 nm were observed in d). While most of the nanotubes shown in e) were less than 1.5 nm in diameter.

Comments 3-5: Fig. 1d shows that the concentration of individualized SWCNT appears to reach a maximum of approximately 1 mg/mL with dispersions of raw SWCNT > 4 mg/mL. One possible reason for this is that, upon redispersion of the suspension following the first centrifugation step, viscosity limitations are again

reached due to the high remaining concentration of carbonaceous material, and SWCNT can no longer be individualized. In such cases where raw SWCNT 4 mg/mL, a third (or fourth, etc.) sonication/centrifugation procedure may be necessary to afford continual overcoming of viscosity limited dispersion. As a proof of (or demonstration against) this concept, the authors should attempt performance of additional (3+) sonication/centrifugation iterations for a sample with raw SWCNT of 8 mg/mL.

Reply: Thank you for your good suggestions. According to your suggestions, we tried to disperse the SWCNT solution with initial concentration of 8 mg/mL by three iterative dispersions. However, after the first ultrasonic dispersion for 11 hours and the subsequent centrifugation for 1 hour, the concentration of supernatant obtained was only 1.17 mg /mL, which was close to the concentration of SWCNTs obtained by re-dispersion, as shown in Figure 1c and d and Supplementary Figure 1. Therefore, even after the second and third dispersions, the highest SWCNT concentration obtained would not exceed 1.17 mg/mL. Due to the high initial concentration, it is difficult for SWCNTs to disperse sufficiently, resulting in a significant decrease in their concentration after the first centrifugation. Meanwhile, the viscosity was also reduced to a low level after the first centrifugation, as shown in Figure 1b. Therefore, currently it is difficult to further increase the separation of SWCNT dispersion by the third sonication/centrifugation iterations. We added the corresponding experimental data and discussion into the revised manuscript and supplementary information.

Results, Page 9 Line 222

Previous: This result was consistent with the dramatic decrease in the viscosity of the higher initial concentration SWCNT solution after the first centrifugation (**Figure 1b**). The original concentration of 4 mg/mL could be the optimal concentration from the perspective of the full utilization of raw materials and subsequent separation efficiency.

Revised: This result was consistent with the dramatic decrease in the viscosity of the higher initial concentration SWCNT solution after the first centrifugation (**Figure 1b**). To improve the dispersion efficiency of high-concentration SWCNT solution, we tried to disperse the SWCNT solution with initial concentration of 8 mg/mL by three iterative dispersions. However, after the first ultrasonic dispersion for 11 hours and subsequent centrifugation for 1 hour, the concentration of supernatant obtained was only 1.17mg /mL (Supplementary Fig. 1), which was close to the concentration of SWCNTs obtained by re-dispersion (Fig. 1c and d). Therefore, even after the second and third dispersions, the highest SWCNT concentration obtained would not exceed 1.17 mg/mL. Due to the high initial concentration, it was difficult for SWCNTs to disperse sufficiently in a short time, resulting in a significant decrease in their concentration after the first centrifugation. The original concentration of 4 mg/mL could be the optimal concentration from the perspective of the full utilization of raw materials and subsequent separation efficiency.

Theme 2 (pages 8-11): a demonstration of how such suspensions achieve scaled-up isolation of high-purity materials from a relatively high-cost source of unpurified SWCNT

Comments 3-6: This section of the manuscript is relatively straightforward and serves as a powerful demonstration of the utility of the novel method described earlier in the work. However, the authors claim the isolation of “single-chirality” SWNT without specifying a quantitative threshold by which they classify a SWCNT suspension as single-chirality. Whatever quantitative threshold the authors are using should be defined early in this section, along with a brief description of the quantitation software used.

Reply: Thank you for your constructive suggestions. We specified a quantitative threshold of “single-chirality” SWCNTs in the revised manuscript.

Results, Page 11 Line 263

Previous: The yields of other (n, m) SWCNTs exhibited a similar increasing trend.

Revised: The yields of other (n, m) SWCNTs exhibited a similar increasing trend. Mass separation of 14 types of (n, m) species was achieved (Supplementary Fig. 7). The lowest chiral purity of them was evaluated to be higher than 70% (Supplementary Fig. 8), which was referred as “single-chirality SWCNTs” due to high enrichment in chiral structure distribution.

Comments 3-7: The authors employ a method of single column SWNT purification that relies on perturbation of both surfactant concentration and temperature during adsorption to the gel and perturbation of surfactant type and concentration during elution. This is in contrast to the first-report of gel-based single chirality SWCNT purification reported by Kataura and coworkers (10.1038/ncomms1313) that relied on overloading conditions at room temperature. In the present manuscript, the authors should provide (either in the main text or the SI) a brief comparison of the rationale behind the two methods. For example, because the method employed here can accommodate adsorption/elution of SWNT at both low and high concentrations, it is expected that significant underloading conditions are used. Additionally, an SI video in the work by Kataura demonstrated near 100% elution of SWNT from a gel, while this work reports some SWNT being irreversibly adsorbed to the gel column. A more direct comparison between the two approaches would help readers contextualize this method with the broader field of gel-based SWNT purification.

Reply: Thank you for your constructive suggestions. We added a brief comparison of the characteristics of the two methods in the Supplementary Information. And, an investigation on the irreversible adsorption of overloading separation was also performed. The comparison suggests:

- 1) The separation of single-chirality SWCNTs at room temperature by selective adsorption under overloading requires repeated separation. This process is complex, time-consuming, and low efficiency. It is difficult to realize mass separation of multiple single-chirality SWCNTs.
- 2) The irreversible adsorption under overloading is also observed, which is even higher than current technique.

We added the comparison in the revised manuscript and supplementary information.

Results, Page 10 Line 251

Previous: The detailed separation process is described in the experimental section (Supplementary Note 1 and Supplementary Fig. 4). The optical absorption spectra indicated that the yield of the separated (6, 4) SWCNTs increased substantially with an increase in the concentration of **individualized** SWCNTs (Figure 2b and Supplementary Fig. 6).

Revised: The detailed separation process is described in the experimental section (Supplementary Note 1 and Supplementary Fig. 4). **Compared with previous works¹⁷⁻²⁵, the current separation strategy showed higher resolution on SWCNT structures and higher separation efficiency (Supplementary Note 4 and Supplementary Fig. 5).** The optical absorption spectra indicated that the yield of the separated (6, 4) SWCNTs increased substantially with an increase in the concentration of **individualized** SWCNTs (Figure 2b and Supplementary Fig. 6).

Supplementary Information, Page 14

Previous: None

Revised:

Supplementary Note 4: Comparison of the previous reported and the current separation strategy

In the previous works^{5,6}, overloading was employed to achieve selective adsorption of specific (n, m) species at room temperature, in which excessive amount of SWCNTs were loaded to a gel column to promote the competition between different (n, m) species in the adsorption onto gel. In this case, the loaded amount of SWCNTs must be much larger than the capacity of the gel column. Therefore, this method is suitable for small gel columns. Overloading has low resolution on the chiral structure of SWCNTs. To separate single-chirality species, multiple iterative overloading or stepwise elution has to be performed. Meanwhile, many SWCNTs that are capable of being adsorbed flowed through the column and remained in the unsorted solutions, which decreases the utilization rate of raw materials. These characteristics decrease the separation efficiency of SWCNTs and thus hindering the

industrial production of single-chirality species.

To further clarifying the characteristics of the overloading method, 10 mL of 4-mg/mL HiPco-SWCNTs dispersed in 2 wt% SDS was loaded to a 10-mL gel column at 25 °C. The adsorbed SWCNTs were eluted by 5 wt% SDS solution and characterized by optical absorption spectra. The unadsorbed fractions were successively loaded to the next columns. As shown in Supplementary Fig. 5a, although most of the (6, 4) and (6, 5) SWCNTs were adsorbed in the first column (Col.1), there was still a considerable amount of them left in the unadsorbed fractions, as evidenced by the adsorption peaks of (6, 4) and (6, 5) in Cols. 2-4. To separate the (6, 4) and (6, 5) in the flow-through fractions, additional separation rounds were required, which significantly degrade the separation efficiency. Besides, because the (6, 4) and (6, 5) that adsorbed in the following columns were relatively low in purity, their separation efficiency decreased dramatically.

The irreversible adsorption under overloading was investigated. 30 mL of (6, 4)-enriched sample was loaded to a 1-mL gel equilibrated by 2 wt% SDS. Then, the column was rinsed by 2 wt% and 5 wt% SDS. 35 mL of unadsorbed and 5 mL of eluted SWCNT solutions were collected and characterized by optical absorption spectra, as shown in supplementary Fig. 5b. Overloading was evidenced by an increase in relative intensity of the (6, 4) absorption peak compared with that of (6, 5) in the eluted solution. The irreversible adsorption of (6, 4) was estimated by $R_{Ir} = \frac{A_{total} * V_{total} - A_{un} * V_{un} - A_{eluted} * V_{eluted}}{A_{total} * V_{total}}$, where A_{un} , A_{eluted} and A_{total} are the area of

S_{11} peaks of (6, 4) in optical absorption spectra of the unadsorbed fraction, the eluted fraction and the loaded SWCNTs, V_{un} , V_{eluted} and V_{total} are the volumes of corresponding solutions. Under the overloading condition, the irreversible adsorption is approximately 28%, which is higher than that of the normal loading condition (~ 15% as shown in Supplementary Fig. 24d in Supplementary Note 9). Collectively, although higher purity and even enantiomer separation could be achieved through iterative separations under overloading⁶, the separation efficiency is much lower compared with our current method.

Supplementary Figure 5. Comparison of overloading and temperature-controlled selective adsorption. (a) Optical absorption spectra of eluted SWCNTs in repeated columns under the condition of overloading. The volume of each eluted solution was tuned to 20 mL. (b) Optical absorption spectra of loaded, unadsorbed and eluted SWCNTs. (c) Optical absorption spectra of eluted SWCNTs

under normal loading conditions at 18 °C. Notably, the eluted SWCNT solutions were diluted to 40 mL due to the absorbance intensity exceeding the test range.

For comparison, selective adsorption of SWCNTs was performed by temperature control. 10 mL of HiPco-SWCNTs with an initial concentration of 4-mg/mL was loaded to a 40-mL gel column at 18 °C. After eluting the unadsorbed SWCNTs, an aqueous solution of 5 %wt SDS was loaded to elute the adsorbed SWCNTs. At a specific temperature, the target (n , m) SWCNTs can be selectively adsorbed under normal loading and are fully extracted due to high resolution, and the adsorption of undesired (n , m) species was prevented and flow through gel columns. The amount of adsorbed SWCNTs under a normal loading condition is certainly larger than that under overloading, leading to an increase in separation efficiency. As exhibited in Supplementary Fig. 5c, nearly all (6, 4) and (6, 5) were adsorbed in Col.1 at 18 °C, as evidenced by the negligible adsorption in Col. 2. These adsorbed (6, 4) and (6, 5) were subsequently separated by stepwise elution. Clearly, temperature control method exhibits higher separation efficiency.

Moreover, compared with the overloading method, the current separation strategy has high resolution. In the previous report⁵, the chirality purities of many of the separated SWCNTs such as (8, 3), (8, 4), (9, 4) and (10, 2) were lower than 70%. And near Zigzag and Zigzag SWCNTs cannot be achieved. Recently, by temperature control method, SWCNTs were separated successively by diameter and by chiral angle using a two-step method^{7, 8}. On the basis of these works, the current method was optimized by combining the selective adsorption and selective desorption to produce single-chirality SWCNTs in a single-step process. Most of them show chiral purities higher than 90% including near zigzag (9, 1) SWCNTs.

Comments 3-8: The authors utilize, both here and later in the manuscript, a series of SWCNT suspensions at various concentrations. It is unclear if these concentrations were achieved by either 1) subjecting SWCNT/SDS of varying raw material concentration to the same iterative sonication/centrifugation procedure; or 2) dilution from stock of a single, relatively high concentration sample obtained by iterative sonication/centrifugation. The authors should clarify this point, as the second method exclusively investigates SWCNT concentration effects on purification while the first is affected by both concentration and the dynamics of system viscosity during the individualization procedure. In other words, does this section purely explore concentration effects or is it a broader exploration of the iterative procedure carried out at different concentrations?

Reply: Thank you for your constructive comments. The SWCNT solutions with different concentrations mentioned in this section were prepared by subjecting SWCNT with different initial concentrations to the sonication-centrifugation-redispersion procedure. As mentioned in the manuscript, the dispersity of the as-prepared SWCNT solution was verified by the separation of (6, 4) SWCNTs. The results indicate that the separated (6, 4) SWCNTs show similar chirality purities and the yield increases linearly with concentrations, indicating that the viscosity

of SWCNTs with different initial concentration during dispersion has a weak influence on the separation of high-purity single-chirality SWCNTs.

Results, Page 10 Line 237

Previous: To explore the effect of SWCNT concentration on the separation yield of single-chirality species, 10 mL of as-prepared HiPco-SWCNT solutions with different concentrations were loaded into a 40-mL gel column for the separation of single-chirality SWCNTs by selective adsorption into and subsequent desorption of SWCNTs from the gel column, as shown in Figure 2a.

Revised: To explore the effect of SWCNT concentration on the separation yield of single-chirality species, 10 mL of as-prepared SWCNT dispersions with different concentrations, **which were prepared by different initial concentrations of HiPco-SWCNT solution**, were loaded into a 40-mL gel column for the separation of single-chirality SWCNTs by selective adsorption into and subsequent desorption of SWCNTs from the gel column, as shown in **Figure 2a**.

Theme 3 (pages 11-15): A demonstration of how such suspensions achieve scaled-up isolation of high-purity materials from a relatively low-cost source of unpurified SWCNT

Comments 3-9: This section of the manuscript involves novel demonstration of separation of single-chirality SWCNT from a low cost, high impurity stock (a notable achievement). One compelling difference between utilization of HiPCO and G-SWCNT is that the authors report high chiral purity obtained from purification of low-concentration HiPCO (Fig 2b) but relatively low chiral purity obtained of purification of low-concentration G-SWCNT (Fig 3e). The extent to which this is discussed in the manuscript is as follows: “indicating that the preparation of a high concentration monodisperse solution is a prerequisite for the separation of single-chirality SWCNTs from G-SWCNTs.” While certainly a true statement, I see this as a lost opportunity to gain valuable mechanistic insight into the relationship between SWCNT stock source, SWCNT stock concentration, and chiral purity of eluted SWCNT. Is there some impurity that is present in G-SWCNT (not present in HiPCO) that limits chiral selectivity? If so, what could this be? Is this limitation since, normalized to mass of stock, G-SWCNT contains less SWCNT than HiPCO so there is simply less SWCNT in the low concentrations of G-SWCNT? Regardless, the specific contrast between the chiral purity obtained from the two stocks should be addressed with additional manuscript text and perhaps additional experimentation to enable the authors to comment on the mechanistic underpinnings of this observation.

Reply: We thank you for your constructive suggestions. Actually, the purity of the separated single-chirality species from 1-mg/mL G-SWCNTs is not low, as shown in the revised supplementary Figure 13, which is comparable with those separated from HiPco-SWCNTs (Figure 4b). As mentioned in the manuscript, G-SWCNTs contain a large number of graphite sheets, but they cannot be adsorbed in gel columns

(Supplementary Figure 17). Therefore, graphite sheets have no effect on the selective adsorption and desorption of SWCNTs. The separation of single-chirality (6, 5), (7, 5), (7, 6) and (8, 4) SWCNTs were not achieved from G-SWCNT with initial concentration of 1 mg/mL possibly because their absolute contents are lower (Supplementary Figure 15), and a large proportion of them were irreversibly adsorbed or flowed through the column as unadsorbed, or both. The comparison of the purities of separated (n, m) species from G-SWCNTs and HiPco-SWCNTs was performed and exhibited in Figure 4 in the manuscript. Most of the separated (n, m) species are comparable. Some of them such as (6,5), (10, 2) SWCNTs exhibit a little lower in purities, which should be ascribed to their different enrichment in different raw materials (Supplementary Figure 15). We added the corresponding discussion on these points in the revised manuscript and supplementary information.

Results, Page 16 Line 379

Previous: Notably, in the case of the initial concentration of 1 mg/mL, single-chirality (6, 5), (7, 5), (7, 6) and (8, 4) SWCNTs were not achieved, indicating that the preparation of a high-concentration and high-dispersity solution is a prerequisite for the separation of single-chirality SWCNTs from G-SWCNTs.

Revised: Notably, in the case of the initial concentration of 1 mg/mL, **although the purities of the separated (n, m) species were higher than 80% or even 90% (Supplementary Fig. 13), the yield was distinctly low, and even single-chirality (6, 5), (7, 5), (7, 6) and (8, 4) SWCNTs were not achieved, most likely due to their low concentration and low absolute content, and a large portion of them was lost as irreversible or unadsorbed species. A more detailed discussion was presented later. These results** indicate that the preparation of a high-concentration **and high-dispersity** solution is a prerequisite for the separation of single-chirality SWCNTs from G-SWCNTs.

Supplementary Information, Page 25

Previous: None

Supplementary Figure 13. Purity evaluation of the (n, m) species separated from G-SWCNTs with the initial concentration of 1 mg/mL.

Theme 4 (pages 15-20): Efforts to probe mechanistic effects of SWCNT concentration on gel-based purification

Comments 3-10: While inclusion of this section of the manuscript is appreciated, the mechanistic insight afforded by these results are unclear. Specifically, both loading amount (at constant concentration) and concentration (at constant loading amount) are explored in terms of the total purified (6,4) SWCNT obtained. It would be useful if the authors explained the logic behind these two experimental choices in terms of what scientific questions (or hypotheses led) to these experimental choices. In other words, the authors should provide a framework by which this data can be interpreted. As it stands currently, the conclusion from the data in Fig. 5 is as follows: “the rapid increase in the production of single-chirality SWCNTs caused by loading high-concentration SWCNT solution resulted from the combined effects of the increase in the concentration and loading amount of SWCNTs.” While a true statement, this provides minimal scientific or mechanistic insight into why either of these factors should affect the dynamic SWNT/surfactant/gel system in the separation scheme employed. Such mechanistic insight, or at least a framework for interpreting data in Fig. 5, is more consistent with my expectations of a publication in Nature Communications than the existing analysis. Coupling this framework with a schematic image of how/why both loading amount and concentration affect purification efficiency would be useful as well.

Reply: Thank you for your constructive suggestions. According to your suggestions,

we added a discussion on the mechanism of adsorption kinetics of SWCNTs in revised manuscript and supplementary information to explain how the concentration and loading amount affect the dynamic system of SWCNT/surfactant/gel.

Results, Page 19, Line 446

Discussion: Notably, increasing the concentration of the loaded SWCNT solution with a fixed volume increased both the concentration and the loading amount of SWCNTs.

Revised: Notably, increasing the concentration of the loaded SWCNT solution with a fixed volume increased both the concentration and the loading amount of SWCNTs. To understand the mechanism of high-concentration SWCNT solution improving the separation efficiency of single-chirality SWCNTs, it was necessary to study the influence mechanism of the concentration and the loading amount of SWCNT solution on the separation yield of single-chirality species, respectively.

Discussion, Page 21, Line 484

Previous: The results show that a lower concentration of the loaded SWCNT solution with a fixed loading amount led to a higher proportion of unadsorbed SWCNTs (Supplementary Fig. 21).

Revised: The results show that a lower concentration of the loaded SWCNT solution with a fixed loading amount led to a higher proportion of unadsorbed SWCNTs (Supplementary Fig. 21). Similar to the separation of biomolecules in a chromatographic system^{32,51}, the concentration of SWCNT solution was a key parameter, which determined the mass transfer of SWCNTs from the bulk solution to the boundary layer adjacent to gel surface and thus the SWCNT concentration in the reaction volume (Supplementary Note 8 and Supplementary Fig. 22). Based on the adsorption kinetics of molecules^{30,31,52}, a high SWCNT concentration in the reaction volume would promote the adsorption of SWCNT onto gel surface. At low concentration, the mass transfer of SWCNTs to the gel surface was hindered due to small SWCNT concentration gradient between bulk solution and boundary layer, and thus decreasing their adsorption onto gel surfaces. Most of SWCNTs remained in the bulk solution and flew through the gel column as unabsorbed species. In contrast, with increasing SWCNT concentration, the transfer of SWCNTs to the gel surface was enhanced, providing more SWCNTs for the binding reactions and thus reducing the proportion of unadsorbed SWCNTs. As the SWCNT concentration further increased, excessive SWCNT could be transferred to the gel surface and accumulated in the reaction volume within a short time, the adsorption kinetics turned to adsorption control. Therefore, further increasing the SWCNTs concentration, the adsorption amount of SWCNTs tended to saturate and the curve slopes in Fig. 5c decreased. A

detailed discussion is presented in the **Supplementary Note 8**.

Discussion, Page 23, Line 530

Previous: With an increase in the loading amount, the proportion of irreversibly adsorbed SWCNTs decreased rapidly from 70% to ~15%, which originated from the quasirandom irreversible adsorption in gel chromatography⁵³.

Revised: With an increase in the loading amount, the proportion of irreversibly adsorbed SWCNTs decreased rapidly from 70% to ~15%, which originated from the quasirandom irreversible adsorption in gel chromatography⁵³. Compared with reversible adsorption, irreversible adsorption sites exhibited stronger affinity with SWCNTs. When SWCNTs flew through gel column, they preferentially adsorbed at the irreversible adsorption sites. Conceptually, a gel column can be considered as being composed of many thin layers of gel⁵⁴. When a small amount of SWCNT solution were loaded into a gel column, SWCNTs could be adsorbed by both reversible and irreversible adsorption sites in the upper layers of the gel column, leaving the irreversible adsorption sites in the lower layers unoccupied. During elution, the eluted SWCNTs from reversible adsorption sites may be captured again by the irreversible adsorption sites in the lower layers (Supplementary Fig. 25), leading to a significant decrease in the SWCNT concentration in the eluted solution and even no SWCNT were collected. With an increase the loading amount of SWCNTs, more and more SWCNTs could be collected in the eluted solution because of the occupation of more irreversible adsorption sites, resulting in a dramatic decrease in the proportion of irreversible adsorption and thus a rapid increase in the separation yield of SWCNTs (Fig. 5d and Supplementary Fig. 25). Notably, as the loading amount continuously increased, more reversibly adsorbed SWCNTs were eluted while the probability of irreversible adsorption of SWCNTs also increased, leading to a constant proportion of irreversible adsorption. In this way, the separation yield began to increase linearly with an increase in the loading amount (Fig. 5d). A detailed discussion was presented in Supplementary Note 9.

Discussion, Page 23, Line 553

Previous: These results sufficiently indicate that the rapid increase in the production of single-chirality SWCNTs caused by loading high-concentration SWCNT solution resulted from the combined effects of the increase in the concentration and loading amount of SWCNTs. In particular, for low concentrations and low loadings, the yield of single-chirality SWCNTs increased more obviously.

Revised: These results sufficiently indicate that the rapid increase in the production of single-chirality SWCNTs caused by loading high-concentration SWCNT solution resulted from the combined effects of the increase in the concentration and loading

amount of SWCNTs. In particular, for low concentrations and low loadings, the yield of single-chirality SWCNTs increased more obviously. At this stage, as the concentration and loading amount increased, the mass transfer and binding rate of SWCNTs were enhanced, and the proportion of reversible adsorption also increased, resulting in a rapid increase in the separation yield (Fig. 5b and Supplementary Notes 8 and 9). With further increasing the concentration and amount of the loaded SWCNTs, the increase rate of the adsorption of SWCNTs gradually slowed down to a constant value (Fig. 5b) because the adsorption of SWCNTs turned to adsorption control instead of mass transfer and the proportion of the irreversible adsorption reduced to a low constant.

Supplementary Information, Page 35 Line 542

Previous: As shown in Supplementary Figure 15d, the proportion of the irreversible adsorption decreases slightly with decreasing SWCNT concentration under such a fixed loading amount.

Revised: As shown in Supplementary Figure 15d, the proportion of the irreversible adsorption decreases slightly with decreasing SWCNT concentration under such a fixed loading amount, while the proportion of unadsorbed SWCNTs increased rapidly.

Similar to biomolecules¹⁰, the adsorption process of SWCNTs to gel can be described as follow:

where $SWCNT_{n,m}^{free}$ and $SWCNT_{n,m}^{ads}$ represent the number of free (n, m) species and the number of adsorbed (n, m) species in a reaction volume of gel, respectively. θ is the total number of adsorption sites in this area. $K_{n,m}^f$ represent the forward rate constants, which is determined by the interaction between a (n, m) SWCNT and an unoccupied binding site¹¹. At low concentrations, the number of each (n, m) species in the reaction volume is low, leading to a low binding rate. The change of concentration of (n, m) species with time reveal the binding rate¹¹:

$$-\frac{dC_{n,m}(t)}{dt} = K_{n,m}^f C_{n,m}(t) \left[\frac{\theta}{V} - \Sigma (C_{n,m}(t_0) - C_{n,m}(t)) \right] \quad (2)$$

At a given time, the rate of change of $C_{n,m}(t)$ nonlinearly decreases with $C_{n,m}(t)$. It suggests that the binding of SWCNTs become harder as the concentration decreases.

As shown in supplementary Fig. 22, in a given SWCNTs/gel system, the SWCNT solution can be divided into bulk solution and boundary layer adjacent to gel surface. Due to the fast adsorption of SWCNTs, the concentration of SWCNTs in boundary layer $C_{n,m}^*$ is smaller than that in the bulk solution $C_{n,m}$, forming a concentration gradient, which drives transfer of SWCNTs from bulk solution to boundary layer.^{12, 13} The simplest expression of the relationship between the flux of a (n, m) species and the “driving force” of mass transfer is as follows¹⁰:

$$J = D_{n,m}(C_{n,m} - C_{n,m}^*)/\delta \quad (3)$$

where $D_{n,m}$ is the diffusion coefficient of the specific SWCNT/surfactant hybrid¹³, δ is the thickness of boundary layer determined by the flow rate of the solution and geometry of gel beads. At low concentration, due to the small difference in the concentrations of $C_{n,m}$ and $C_{n,m}^*$, small concentration gradient hinders the mass transfer of SWCNTs to the gel surface and thus decreasing their adsorption onto gel surfaces due to low concentration in the reaction volume adjacent to the gel surface. SWCNTs tend to remain in the bulk solution and may flow through the gel, leading to more unadsorbed SWCNTs, as shown by the yellow line in Supplementary Fig. 22.

As the concentration in the bulk solution increases, the difference between $C_{n,m}$ and $C_{n,m}^*$ increases. The transfer of SWCNTs to the gel surface has been enhanced, providing more SWCNTs for the binding reactions, as shown by the red line in Supplementary Fig. 22, which reduce the proportion of unadsorbed SWCNTs. Then, as excessive SWCNTs were transferred to the gel surface in a short time, the adsorption kinetics turn to adsorption-control. The SWCNTs adjacent to the gel surface accumulate, which increase $C_{n,m}^*$ and decreases the concentration gradient in the boundary layer. At this stage, the resistance of SWCNT binding is mainly attributed to the adsorption process (red line in Supplementary Fig. 22). Therefore, further increasing the SWCNTs concentration, the adsorption amount of SWCNTs tend to saturate. Notably, this model is very simple and does not involve interactions between different (n, m) species and SWCNT transfer inside gel beads.

Supplementary Figure 22. Schematic diagram of mass transfer from the bulk SWCNT solution to gel surface.

Previous: Although the adsorption condition of semiconducting SWCNTs was not fulfilled at the concentration of 5 wt% SDS, irreversible adsorption was still observed, which may relate to the inhomogeneity of gel¹⁵⁻¹⁸.

Revised: Although the adsorption condition of semiconducting SWCNTs was not fulfilled at the concentration of 5 wt% SDS, irreversible adsorption was still observed, which may relate to the inhomogeneity of gel¹⁴⁻¹⁶. **Similar to equation (1), irreversible binding process regardless of the environment condition can be described as follow:**

where θ_{Ir} and $K_{n,m}^{Ir}$ represent the number of binding sites of irreversible adsorption in the reaction volume and the forward rate constant of irreversible adsorption, respectively. Compared with reversible, irreversible adsorption sites exhibits stronger affinity with SWCNTs. When SWCNTs flow through gel column, they preferentially adsorb at the irreversible adsorption sites. During the elution process, eq. 1 proceeds in the reverse direction, while eq. 4 is still a forward reaction. Thus, SWCNTs eluted from reversible sites at the top of the gel column flow down and may be captured by irreversible sites at the bottom of the gel column. Conceptually, a gel column is usually considered a pile composed of many plates, each of which composes of a very thin gel layer¹⁷. Adsorption of SWCNTs in the upper plates leads to the occupation of both reversible and irreversible adsorption sites. Then, the remained SWCNTs in solution flow down to the plates below. At low loading amounts, only plates at the top were filled. The solution that flows down contains much fewer SWCNTs, leaving the irreversible adsorption sites in the lower plates unoccupied. These unoccupied sites may trap SWCNTs that are eluted from the upper plates. As a result, nearly no SWCNTs were collected from the eluent. In contrast, when a large number of SWCNTs were loaded into the column, irreversible adsorption sites in the lower plates were filled. Thus, when the eluent flows through the column, the remained SWCNTs can be eluted from the lower plates. With an increase in loading amount, the proportion of irreversible adsorption decreases. This hypothesis coincides with the experimental results in Supplementary Fig. 10, where the amount of SWCNTs separated from a gel recycled for 5 separation runs is higher than that of the new gel, because some of the irreversible adsorption sites have been filled in the recycled gel.

Interestingly, the ratio of reversible and irreversible adsorption gradually reached a plateau with further increasing loading amount, as shown in Supplementary Fig. 24d. As mentioned above, with increasing the loading amount of SWCNTs, the irreversible adsorption sites are occupied, resulting in a gradual decrease in the proportion of irreversible adsorption. However, increasing the loading amount also increase the possibility of irreversible adsorption of SWCNTs. For example, the impurities in SWCNT solutions such as amorphous carbon and bundles also increased with loading

amount and form more irreversible adsorption¹⁴. The two aspects eventually hold the proportion of irreversible adsorption at $\sim 15\%$. Notably, this mechanism is established under the condition of normal loading. Overloading is supposed to be more complicated, involving the competition between different (n, m) species and the saturation of gel. Additionally, the form of irreversible adsorption may also be attributed to the inherent properties of some SWCNTs, such as length, defects and coatings¹⁴.

Comments 3-11: The analysis in Fig. 5b in terms of the “increase rate” is not presented alongside a description of why the increase rate is expected to maximize at one specific SWCNT concentration. From the perspective of system optimization, the present analysis is useful. However, I expect the readership of this journal to seek scientific and mechanistic insight beyond considerations of pure optimization.

Reply: Thank you for your good suggestions. We added a discussion on the dynamic adsorption of SWCNTs with increasing concentration and loading amount in the revised manuscript and supplementary information. Please refer to the reply to Comments 3-10.

Discussion, Page 23, Line 553:

Previous: These results sufficiently indicate that the rapid increase in the production of single-chirality SWCNTs caused by loading high-concentration SWCNT solution resulted from the combined effects of the increase in the concentration and loading amount of SWCNTs. In particular, for low concentrations and low loadings, the yield of single-chirality SWCNTs increased more obviously.

Revised: These results sufficiently indicate that the rapid increase in the production of single-chirality SWCNTs caused by loading high-concentration SWCNT solution resulted from the combined effects of the increase in the concentration and loading amount of SWCNTs. In particular, for low concentrations and low loadings, the yield of single-chirality SWCNTs increased more obviously. *At this stage, as the concentration and loading amount increased, the mass transfer and binding rate of SWCNTs were enhanced, and the proportion of reversible adsorption also increased, resulting in a rapid increase in the separation yield (Fig. 5b and Supplementary Notes 8 and 9). With further increasing the concentration and amount of the loaded SWCNTs, the increase rate of the adsorption of SWCNTs gradually slowed down to a constant value (Fig. 5b) because the adsorption of SWCNTs turned to adsorption control instead of mass transfer and the proportion of the irreversible adsorption reduced to a low constant.*

Comments 3-12: Data in Figs. 5b 5c, and 5d all include error bars but there is no

description of performing the experiment in triplicate and/or what statistical factor is represented in the length of the error bar.

Reply: Thank you for your suggestions. We apologize for missing the description of the error bars. The experiments were performed repeatedly for three times. The error bars exhibit the standard deviation of the results of the three repetitions. We added the description of error bars in the caption of Figure 5 and Supplementary Figs. 21 and 24.

Discussion, Caption of Figure 5, Page 20 Line 461

Previous: None

Revised: Each experiment was repeated for three times. The error bars exhibit the standard deviation.

Supplementary Information, Caption of Supplementary Fig. 21, Page 35 Line 559

Previous: None

Revised: Experimental error was evaluated by repeating the experiment for three times. The standard deviation was exhibited through error bars.

Supplementary Information, Caption of Supplementary Fig. 24, Page 40 Line 631

Previous: None

Revised: Experimental error was evaluated by repeating the experiment for three times. The standard deviation was exhibited through error bars.

Comments 3-13: This section states that “with an increase in SWCNT concentration, the equilibrium between adsorption and desorption shifted to the direction of adsorption.” I am skeptical of this explanation because, under isothermal conditions,

if the adsorption/desorption process of SWCNT to/from gel was driven by equilibrium, it would be possible to elute adsorbed SWNT through passage of neat surfactant (which would drive the equilibrium to the desorbed state). However, performance of such does not result in eluted SWNT, rather, elution requires introduction of surfactant of higher concentration and/or cosurfactants. Given this, there is likely some other mechanism (other than equilibrium) behind the concentration effects described in Fig. 5.

Reply: Thank you very much for your thought-provoking question. We agree with you. We revised the corresponding description in the revised manuscript and supplementary information. As described in Reply to Comments 3-10, our revised explanation on the mechanism is based on a forward binding reaction other than equilibrium.

Introduction, Page 4, Line 86

Previous: Based on molecular adsorption kinetics^{26,27}, an increase in the SWCNT concentration could shift the adsorption equilibrium of SWCNTs and result in an increase in the adsorption of the absolute amount of each (n, m) semiconducting species in solution, thus improving the separation efficiency and yield of single-chirality SWCNTs.

Revised: Based on molecular adsorption kinetics³⁰⁻³², **increasing the SWCNT concentration could promote their mass transfer and binding rate to the gel surface** and result in an increase in the adsorption of the absolute amount of each (n, m) semiconducting species in solution, thus improving the separation efficiency and yield of single-chirality SWCNTs.

Discussion, Page 21, Line 484

Previous: The results show that a lower concentration of the loaded SWCNT solution with a fixed loading amount led to a higher proportion of unadsorbed SWCNTs (Supplementary Fig. 15). With an increase in the SWCNT concentration, the equilibrium between adsorption and desorption shifted to the direction of adsorption^{26,27,41}, resulting in an increase in the reversibly adsorbed SWCNTs and a decrease in the proportion of unadsorbed and irreversibly adsorbed SWCNTs, which inevitably increased the separation yield of SWCNTs.

Revised: The results show that a lower concentration of the loaded SWCNT solution with a fixed loading amount led to a higher proportion of unadsorbed SWCNTs (Supplementary Fig. 21). **Similar to the separation of biomolecules in a chromatographic system^{32,51}, the concentration of SWCNT solution was a key parameter, which determined the mass transfer of SWCNTs from the bulk solution to the boundary layer adjacent to gel surface and thus the SWCNT concentration in the**

reaction volume (Supplementary Note 8 and Supplementary Fig. 22). Based on the adsorption kinetics of molecules^{30,31,52}, a high SWCNT concentration in the reaction volume would promote the adsorption of SWCNT onto gel surface. At low concentration, the mass transfer of SWCNTs to the gel surface was hindered due to small SWCNT concentration gradient between bulk solution and boundary layer, and thus decreasing their adsorption onto gel surfaces. Most of SWCNTs remained in the bulk solution and flew through the gel column as unadsorbed species. In contrast, with increasing SWCNT concentration, the transfer of SWCNTs to the gel surface was enhanced, providing more SWCNTs for the binding reactions and thus reducing the proportion of unadsorbed SWCNTs. As the SWCNT concentration further increased, excessive SWCNT could be transferred to the gel surface and accumulated in the reaction volume within a short time, the adsorption kinetics turned to adsorption control. Therefore, further increasing the SWCNTs concentration, the adsorption amount of SWCNTs tended to saturate and the curve slopes in Fig. 5c decreased. A detailed discussion is presented in the Supplementary Note 8.

Supplementary Information, Page 23 Line 373

Previous: In contrast, the proportion of the unadsorbed SWCNTs increased as the SWCNT concentration decreasing, indicating that the adsorption equilibrium between adsorption and desorption shifts towards the desorption¹¹⁻¹⁴: $SWCNT_{free} + \theta \xrightleftharpoons{K} SWCNT_{abs}$, where $SWCNT_{free}$ and $SWCNT_{abs}$ represent the number of free SWCNT and the number of adsorbed SWCNT in a reaction volume of gel, respectively. θ is the total number of adsorption site in this area. K represents the equilibrium constant.

Revised: Similar to biomolecules¹⁰, the adsorption process of SWCNTs to gel can be described as follow:

where $SWCNT_{n,m}^{free}$ and $SWCNT_{n,m}^{ads}$ represent the number of free (n, m) species and the number of adsorbed (n, m) species in a reaction volume of gel, respectively. θ is the total number of adsorption sites in this area. $K_{n,m}^f$ represent the forward rate constants, which is determined by the interaction between a (n, m) SWCNT and an unoccupied binding site¹¹. At low concentrations, the number of each (n, m) species in the reaction volume is low, leading to a low binding rate. The change of concentration of (n, m) species with time reveal the binding rate¹¹:

$$-\frac{dC_{n,m}(t)}{dt} = K_{n,m}^f C_{n,m}(t) \left[\frac{\theta}{V} - \Sigma \left(C_{n,m}(t_0) - C_{n,m}(t) \right) \right] \quad (2)$$

At a given time, the rate of change of $C_{n,m}(t)$ nonlinearly decreases with $C_{n,m}(t)$. It

suggests that the binding of SWCNTs become harder as the concentration decreases.

As shown in supplementary Fig. 22, in a given SWCNTs/gel system, the SWCNT solution can be divided into bulk solution and boundary layer adjacent to gel surface. Due to the fast adsorption of SWCNTs, the concentration of SWCNTs in boundary layer $C_{n,m}^*$ is smaller than that in the bulk solution $C_{n,m}$, forming a concentration gradient, which drives transfer of SWCNTs from bulk solution to boundary layer.^{12, 13} The simplest expression of the relationship between the flux of a (n, m) species and the “driving force” of mass transfer is as follows¹⁰:

$$J = D_{n,m}(C_{n,m} - C_{n,m}^*)/\delta \quad (3)$$

where $D_{n,m}$ is the diffusion coefficient of the specific SWCNT/surfactant hybrid¹³, δ is the thickness of boundary layer determined by the flow rate of the solution and geometry of gel beads. At low concentration, due to the small difference in the concentrations of $C_{n,m}$ and $C_{n,m}^*$, small concentration gradient hinders the mass transfer of SWCNTs to the gel surface and thus decreasing their adsorption onto gel surfaces due to low concentration in the reaction volume adjacent to the gel surface. SWCNTs tend to remain in the bulk solution and may flow through the gel, leading to more unadsorbed SWCNTs, as shown by the yellow line in Supplementary Fig. 22.

As the concentration in the bulk solution increases, the difference between $C_{n,m}$ and $C_{n,m}^*$ increases. The transfer of SWCNTs to the gel surface has been enhanced, providing more SWCNTs for the binding reactions, as shown by the red line in Supplementary Fig. 22, which reduce the proportion of unadsorbed SWCNTs. Then, as excessive SWCNTs were transferred to the gel surface in a short time, the adsorption kinetics turn to adsorption-control. The SWCNTs adjacent to the gel surface accumulate, which increase $C_{n,m}^*$ and decreases the concentration gradient in the boundary layer. At this stage, the resistance of SWCNT binding is mainly attributed to the adsorption process (red line in Supplementary Fig. 22). Therefore, further increasing the SWCNTs concentration, the adsorption amount of SWCNTs tend to saturate. Notably, this model is very simple and does not involve interactions between different (n, m) species and SWCNT transfer inside gel beads.

Supplementary Figure 22. Schematic diagram of mass transfer from the bulk SWCNT solution to gel surface.

Theme 5 (pages 20-22): A Life Cycle Assessment estimating the costs of using the reported novel method to produce purified SWNT at scale

Comments 3-14: This analysis is outside of my area of expertise, so I will avoid detailed comments on its validity. However, it is nonetheless worthwhile to state that a LCA analysis fits with the scope and scale of the work and is of general interest to those pursuing more effective SWCNT purification methods.

Reply: Thank you for your constructive suggestions.

In summary, the manuscript submitted by Liu and coworkers contains compelling results that are likely to be of significant interest to individuals working in the broad field of carbon nanotube science. As it is presently written, my assessment of this work is that it may be considered for publication in Nature Communications after revision and further review. The overarching theme of my request for revision stems from a desire for the authors to compliment this groundbreaking finding with additional analysis (experimental or otherwise) to better describe the nanoscale mechanism of both improved dispersion concentrations and improved process efficiency of SWNT purification. I would like to sincerely to thank both the editor, for the opportunity to participate in the review process of this work, as well as the authors, for their dedication to advancing this impactful area.

Reply: Thank you again for your positive comments, constructive suggestions, valuable recommendation. Your suggestion has greatly improved our manuscript.

REVIEWERS' COMMENTS

Reviewer #1 (Remarks to the Author):

The revision has addressed all the concerns I raised. I recommend publication of this work.

Reviewer #2 (Remarks to the Author):

The manuscript by Liu and co-workers, which describes the use of gel chromatography to purify highly concentrated solutions of single-walled carbon nanotubes, was re-reviewed after extensive revisions. The authors are commended on the extent to which they revised this manuscript in their attempt to address reviewer comments. While the authors addressed many of the reviewer concerns, this reviewer feels that there are still several outstanding issues that render this manuscript unsuitable for publication in this journal. For example, the authors claim that the gel can be reused at least 20 times, but the data shows that the mass eluted after 20 cycles is decreased by 30%, which is not consistent with the claims. If the authors were to plot the mass eluted as a function of cycle number, it would show a clear and relatively rapid decline in performance. In addition, the authors extensively discuss materials that are irreversibly adsorbed to the gel. If a portion of the material is irreversibly adsorbed to the gel, this necessarily precludes the gel from performing in a consistent manner from one batch to the next, making it non-reusable. Furthermore, the ultimate purity of the single-chirality nanotubes is relatively low for most chiralities, especially when starting with the commercially available inexpensive nanotubes (as depicted in Figure 4b). Real-world device applications require much higher purities. Finally, the manuscript is extremely long, making it inappropriate as a communication. It should be submitted to another journal as a full paper. Ultimately, the authors do a good job of showing that gel chromatography on concentrated samples leads to higher yields of isolated nanotube species, but this result remains unsurprising. The use of gel chromatography to isolate samples of specific chirality has been known for some time. This report certainly adds to the body of literature on this topic, and should be published in a more specialized journal, but is not the sort of ground breaking work that should appear in Nature Communications.

Reviewer #3 (Remarks to the Author):

Following careful consideration of the comments made by myself and two other reviewers, along with the detailed response to those comments by the authors, I recommend in favor of publication of this manuscript in Nature Communications. I look forward to following the impact of this work within the SWCNT purification community.

Response to Reviewers:

Reviewer #1 (Remarks to the Author):

The revision has addressed all the concerns I raised. I recommend publication of this work.

Reply: Thank you for your positive comments and valuable recommendation.

Reviewer #2 (Remarks to the Author):

The manuscript by Liu and co-workers, which describes the use of gel chromatography to purify highly concentrated solutions of single-walled carbon nanotubes, was re-reviewed after extensive revisions. The authors are commended on the extent to which they revised this manuscript in their attempt to address reviewer comments. While the authors addressed many of the reviewer concerns, this reviewer feels that there are still several outstanding issues that render this manuscript unsuitable for publication in this journal. For example, the authors claim that the gel can be reused at least 20 times, but the data shows that the mass eluted after 20 cycles is decreased by 30%, which is not consistent with the claims. If the authors were to plot the mass eluted as a function of cycle number, it would show a clear and relatively rapid decline in performance. In addition, the authors extensively discuss materials that are irreversibly adsorbed to the gel. If a portion of the material is irreversibly adsorbed to the gel, this necessarily precludes the gel from performing in a consistent manner from one batch to the next, making it non-reusable. Furthermore, the ultimate purity of the single-chirality nanotubes is relatively low for most chiralities, especially when starting with the commercially available inexpensive nanotubes (as depicted in Figure 4b). Real-world device applications require much higher purities. Finally, the manuscript is extremely long, making it inappropriate as a communication. It should be submitted to another journal as a full paper. Ultimately, the authors do a good job of showing that gel chromatography on concentrated samples leads to higher yields of isolated nanotube species, but this result remains unsurprising. The use of gel chromatography to isolate samples of specific chirality has been known for some time. This report certainly adds to the body of literature on this topic, and should be published in a more specialized journal, but is not the sort of ground breaking work that should appear in Nature Communications.

Reply: Thank you for your comments. Irreversible adsorption is a common problem. In our present work, it should be noted that the separation yield of single-chirality species increased by 15% within the first 5 cycles of the gel column, although it

decreased by approximately 30% after 20 cycles of the gel column due to irreversible adsorption. On the average, the separation yield of single-chirality species did not decrease significantly. Meanwhile, the purities of the separated single-chirality SWCNTs was not affected by the recycled gel column (Supplementary Fig. 10a). As you suggested, we plotted the mass eluted as a function of cycle number, which was added into the revised supplementary information (Supplementary Fig. 10b). We also added a detailed experimental data and discussion in the revised supplementary Note 6 and the revised manuscript.

As shown in Figure 4b, the purities of most of the separated single-chirality species are higher than 80%. Multiple species even exhibit the purity of more than 90%, such as (6,4), (9,1), (6,5), (7,3), (7,5) (10, 2), (9,4), which is sufficient for the application in optics, electronics and photo-electronics. We believe that the purity of the separated single-chirality SWCNTs can be further increased by optimizing the separation parameters.

Based on important comments and suggestions from reviewers, we have added many critical experimental details and mechanism discussions to the revised manuscript, greatly improving its quality. But it also increased its length. We believe the quality should be more important.

Results, Page 13 Line 306

Previous: Another important feature of the current method is that the gel column used could be recycled for at least 20 times without significant degradation in performance, which significantly reduced the separation cost of SWCNTs (Supplementary Fig.10).

Revised: Another important feature of the current method is that the gel columns could be recycled for 20 times, which reduced the separation cost of SWCNTs. Specifically, the separation yield of single-chirality SWCNTs increased by 8-15% gradually with increasing cycle number within the first 5 cycles. Beyond 5 cycles, the separation yield started to decline, possibly because the growing SWCNT bundles and carbon impurities trapped in gel columns degraded the adsorption capacity of the gel column. After 20 cycles, the separation yield decreased by approximately 30% compared with the new gel. Thus, on the average, the separation yield of single-chirality species did not decrease significantly. The detailed experimental process and discussion was presented in supplementary Note 6 and supplementary Fig.10.

Supplementary Information, Page 22

Previous:

Supplementary Figure 10. The impact of high-concentration SWCNT solutions on gel column's life time and performance. 10 mL of SWCNT solutions with the initial concentration of 4 mg/mL were separated by columns filled with 40 mL of gel. The gel was recycled for 5, 10 and 20 separation runs. Optical absorption spectra of the separated (6, 4) SWCNTs are presented. The purities of the separated (6, 4) SWCNTs were not affected by the recycling of gel. However, the yield decreases slightly with increasing recycle times. After the column was recycled for 20 times, the mass of eluted SWCNT was decreased by approximate 30% compared with the new gel in a separation round. These results indicate that the gel column can be recycled at least 20 times.

Revised:

Supplementary Note 6: The impact of high-concentration SWCNT solutions on gel column's life time and performance.

Ten-mL SWCNT solutions with the initial concentration of 4 mg/mL were separated by a column filled with 40 mL of gel using the method described in supplementary Note 1. The gel column was recycled for 20 separation runs. The optical absorption spectra of the separated (6, 4) SWCNTs are presented in supplementary Fig. 10a. Clearly, the purities of the separated (6, 4) SWCNTs were not affected by the recycling of gel, but the varied absorbance indicated that the yield of (6, 4) SWCNTs changed with cycle number. The relationship between the (6, 4) yield and the number of cycles of gel column were plotted in supplementary Fig. 10b, where the yield of (6, 4) SWCNTs was represented by the S_{11} peak area. Interestingly, the separation yield slightly increased by 8-15% within the first 5 cycles. However, the yield decreased upon 5 cycles. After 20 cycles, the yield decreased by approximate 30% compared with the new gel in a separation round. This result was confirmed by three independent experiments. The yield variation of (6, 4) SWCNTs should be strongly related to the irreversible adsorption of SWCNTs in the gel column. Due to the presence of a large number of irreversible adsorption sites in the new gel, the loaded

SWCNTs suffered from loss due to irreversible adsorption. With increasing the number of cycles, the amount of irreversible adsorption possibly decreased due to the occupation of irreversible adsorption sites as discussed in supplementary Note 10. Therefore, within the first five cycles, the production of (6, 4) SWCNTs increased. With continuously increasing the number of cycles, the reversible adsorption sites may begin to turn into irreversible adsorption sites, weakening the adsorption capacity of the gel column and thus a decrease in the separation yield. Additionally, the impurities such as amorphous carbon are also likely to occupy adsorption sites and form irreversible adsorption, thus decreasing the gel adsorbability to SWCNTs⁹. The degradation of gel may vary with the type of surfactant, the structural distribution of SWCNTs, separation temperature and the dispersity of SWCNT solutions.

Supplementary Figure 10. The variation of gel performance with respect to cycles. (a) Optical absorption spectra of the separated (6, 4) using recycled gel. (b) Relationship between the (6, 4) yield and the number of cycles of gel column. The amount of collected (6, 4) was evaluated by the S₁₁ peak area of (6, 4) in the range from 800 to 940 nm. The error bars exhibit the standard deviation of three repeated experiments.

Reviewer #3 (Remarks to the Author):

Following careful consideration of the comments made by myself and two other reviewers, along with the detailed response to those comments by the authors, I recommend in favor of publication of this manuscript in Nature Communications. I look forward to following the impact of this work within the SWCNT purification community.

Reply: Thank you for your positive comments and valuable recommendation.